# Between global risk reduction goals, scientific-technical capabilities and local realities: a modular approach for user-centric multi-risk assessment

Elisabeth Schoepfer[1], Jörn Lauterjung[2], Torsten Riedlinger[1], Harald Spahn[3], Juan Camilo Gómez Zapata[2,4], Christian León[5], Hugo Rosero-Velásquez[6], Sven Harig[7], Michael Langbein[1], Nils Brinckmann[2], Günter Strunz[1], Christian Geiß[1,8] and Hannes Taubenböck[1,9]

[1]German Aerospace Center (DLR), German Remote Sensing Data Center (DFD), Wessling, 82234, Germany
[2]Helmholtz Centre Potsdam GFZ German Research Centre for Geosciences, Potsdam, 14473, Germany
[3]Independent Consultant, Apen, 26689, Germany
[4]University of Potsdam, Institute for Geosciences, Potsdam, 14476, Germany
[5]DIALOGIK gGmbH, Stuttgart, 70176, Germany
[6]Technical University of Munich, Engineering Risk Analysis Group, Munich, 80333, Germany
[7]Alfred Wegener Institute, Helmholtz Centre for Polar and Marine Research (AWI), Bremerhaven, 27570, Germany
[8]University of Bonn, Department of Geography, Bonn, 53115, Germany
[9]University of Würzburg, Institute of Geography and Geology, Würzburg, 97074, Germany

*Correspondence to*: Elisabeth Schoepfer (elisabeth.schoepfer@dlr.de)

**Abstract.** We live in a rapidly changing and globalized society. The increasing interdependence and interconnection of our economic, social and technical systems, growing urbanization and increasing vulnerability to natural hazards (including climate change) are leading to ever more complex risk situations. This paper presents a conceptual approach for user-centred multi-risk assessment aimed to support potential users like disaster risk managers, urban planners or critical infrastructure operators. Based on the latest scientific and technical capabilities, we developed a method that enables the simulation and visualization of a range of scenarios with different intensities. It is based on a modular and decentralized system architecture using distributed web services that are published online, including a user-friendly interface. The approach is demonstrated using the example of earthquakes and tsunamis for the Lima Metropolitan area (Peru), a megacity exposed to various cascading natural hazards. The development involved a wider group of Peruvian stakeholders from research and practice in a structured, iterative and participative feedback process over a period of 2.5-years to capture the needs and requirements from the user perspective. Results from the feedback process, including 94 responses to 5 questionnaires, confirmed the potential of the demonstrator as a complementary analysis and visualization tool. Together with the visualisation of cascading processes, the ability to simulate and compare scenarios of varying severity was considered relevant and useful for improving the understanding and preparedness for complex multi-risk situations, in the practical application, especially at the local level.

# 1 Introduction

In this article, we provide a brief introduction on the paradigm shift from managing disasters to managing risks, followed by single-hazard to multi-risk assessment. We highlight four global strategies that address disaster risk reduction and call for action. In these introductory sections, we note the need to bridge the gap between these global goals and specific tools for implementation.

## 1.1 From managing disasters to managing risks

A disaster as defined by the UNISDR is "*a serious disruption of the functioning of a community or a society involving widespread human, material, economic or environmental losses and impacts, which exceed the ability of the affected community or society to cope using its own resources*" (UNISDR, 2009, p. 9). Among all disasters those caused by natural hazards claim the greatest number of victims per year. Especially climate induced hazards have increased in frequency and intensity of events (EEA, 2021; WMO, 2021). In 2021 global losses from disasters induced by the interplay of natural hazards and vulnerabilities added up to US$ 280bn (Munich RE, 2022). Population growth, rapid urbanization and concentration of people, assets and economic activities in hazardous areas raised during the past decades (Pesaresi et al., 2017). Inadequate or unplanned socioeconomic development in places exposed to a variety of hazards increases the vulnerability of societies (UNEP, 2016).

Beyond immediate crisis management and rapid response, disaster preparedness is growing in importance (Strunz et al., 2022). The transition from disaster management to risk management is emphasized in the Sendai Framework for Disaster Risk Reduction 2015-2030 adopted at the Third UN World Conference in Sendai, Japan on 18 March 2015 (UNISDR, 2015a). The Sendai Framework for Action's priority one is "*understanding disaster risk*" (UNISDR, 2015a). This priority describes that "*policies and practices for disaster risk management should be based on an understanding of disaster risk in all its dimensions of vulnerability, capacity, exposure of persons and assets, hazard characteristics and the environment*" (UNISDR, 2015a, p. 14). The demand to consider and to understand all dimensions of risk is a basic requirement. At the same time, it remains an enormous challenge taking into consideration multi-hazard and multi-risk situations with all the interdependencies involved in such events. The paradigm shift was taken up by UNDRR in the Global Assessment Report on Disaster Risk Reduction (GAR) in 2019. In the report it is stated that this shift is "*seeking to redress practice that has for many years seen ex ante action articulating the complex risk drivers from which disasters materialize eclipsed by action responding to the manifestation of disasters*" (UNDRR, 2019, chapter 15, p. 403). In 2022 UNDRR stresses that "*scientific risk assessments by experts are essential in designing strategies for reducing risk and future losses from extreme events.*" (UNDRR, 2022a, chapter 8, p. 111). It is further stated that scientific results are in demand to assist key decision-makers, and that information needs to be communicated clearly and transparently.

## 1.2 From single-hazard to multi-risk assessment

More people globally face natural hazards, especially in poorly planned urban areas where effective prevention and risk management could save lives and minimize losses (Pesaresi et al., 2017; Hossain et al., 2017; UNDRR, 2023). However,

meaningful risk management strategies are complex, since hazards effects are multi-dimensional and beyond this rarely isolated. An earthquake can trigger a tsunami, soil liquefaction and/or landslides. The hazard interactions are manifold (Gill and Malamud, 2014) and become even more complex as they can further affect critical infrastructures (Barquet et al., 2023). UNDRR (2019) calls for "*information on the nature and extent of hazards, vulnerabilities, and the magnitude and likelihood of potential damage and loss needs to expand from single-hazard to multi-risk assessments to capture the range of intersecting threats.*" (UNDRR, 2019, chapter 12.3.3, p. 346).

First studies on *multi-hazards* and *multi-hazard risk* are documented by the mid-1980s (e.g., Fitz Simons, 1986; Chiu and Chock, 1998; Granger et al., 1999). Fitz Simons (1986) discusses different hazard forces and agents to which buildings, and in particular historic architectures and museums, are exposed. However, the interdependencies of hazards were not studied. Chiu and Chock (1998) presented a proposal for "*multi-hazard performance-based building design criteria*" addressing wind and earthquake hazards. They conclude that due to limited desktop computational power handling of multi-hazard design criteria was not possible in previous years. Granger et al. (1999) performed a provisional *multi-hazard risk assessment* of Cairns, Australia. Five hazards types, i.e. earthquakes, landslides, floods, destructive winds and storm tides, were analysed. The exposure to the hazard and related vulnerability was considered while cascading effects and interdependencies were not considered in detail.

While single-hazard oriented research dominated the past, studies on multi-hazards risks and multi-risk analysis came more into the focus shortly after the turn of the millennium. Marzocchi et al. (2009) define the purpose of multi-risk analyses "*to establish a ranking of the different types of risk taking into account possible cascade effects i.e. the situation for which an adverse event triggers one or more sequential events (synergistic event).*" A detailed review is not provided in our article, but refers to selected publications as follows. Kappes et al. (2012) focused in a review on the challenges of analysing multi-hazard risks whereas Komendantova et al. (2014) analysed the feedback from civil protection stakeholders on two multi-hazard and multi-risk decision support tools which reveal that interest is high, but hampered due to the underlying complexity. Gallina et al. (2016) published a review of multi-risk methodologies for natural hazards concluding that most of the approaches rely on the analysis of static vulnerability. Zschau (2017) provided an overview of the multi-hazards and multi-risk assessment including a terminology from single hazard to multi-risk. Pittore et al. (2017) discussed the challenge of implementing an exposure model suitable for different hazards. A comprehensive review of multi-hazards research and risk assessment was published by Ciurean et al. (2018). The authors provide various observations among others that methodologies in real case study examples was (at the time of publication) still limited. Gill et al. (2020) analysed seven regional multi-hazard interaction frameworks (Tarvainen et al., 2006; De Pippo et al., 2008; Kappes et al., 2010; van Westen et al., 2014a; Neri et al., 2008, Neri et al., 2013; Liu et al., 2016) and presented a scalable interaction framework approach with different resolutions of information using Guatemala as an example. Further approaches for identifying and characterizing hazard interactions are for example laid out in Taubenböck et al., (2009; 2013), Mignan et al. (2014), Gill and Malamud (2014; 2016; 2017), Liu et al. (2015), Tilloy et al. (2019; 2022), De Angeli et al. (2022), and Sköld Gustafsson et al. (2023). Ward et al. (2020) provide a review of global risk studies across different hazards. They list similarities and differences between the approaches taken within and across the different hazards. Lópex-Saavedra and Martí (2023) assessed the application of the multi-hazard concept in existing risk management systems. The need to model multiple

hazards is addressed in the review published by Cremen et al. (2022). This is supported by the systematic and scientometric review on multi-hazard risk assessment performed by Owolabi and Sajjad (2023) where they conclude, among others, on emphasizing on cascading and interrelated relationships among multiple hazards. Most recently, Hochrainer-Stigler et al. (2023)
proposed a framework to guide the analysis of multi- and systemic risk which however has not yet been applied in real case studies. Goda and De Risi (2023) discussed future perspectives of earthquake-tsunami catastrophe modelling.

The aim is not to stop at theory and research, but to offer practical solutions via tools and applications. There are databases, applications and platforms existing which support or directly target to model risks, as also recently outlined by Negulescu et al. (2023). Among them, we mention a few, like the initiatives PAGER (Wald et al., 2011) and ShakeCast (Wald and Lin, 2007),
and further focusing on multiple hazards, e.g., HAZUS-MH (FEMA, 2004); CAPRA (Cardona et al., 2012); RiskCity and WebRiskCity (Frigerio and Westen, 2010); PREVIEW (Giuliani and Peduzzi, 2011); RiskChanges (van Westen et al., 2014b; 2022); WESR (UNEP, 2021, 2023); DRMKC (Marin Ferrer et al., 2019; Joint Research Centre (European Commission) et al., 2020); RiskScape (Paulik et al., 2022); CLIMADA (Kropf et al., 2022); IN-CORE (van de Lindt et al. 2023); VIGIRISKS (Negulescu et al., 2023).
Designing information systems or tools to analyse multi-hazard risks and dynamically update impacts from cascading hazard effects presents significant challenging (cf., Cremen, et al., 2022; Paulik et al., 2022).

### 1.3 From global risk reduction goals to local solutions

In addition to the scientific work, the importance of risk assessment and its challenges are addressed in global strategies. Table 1 shows four selected global strategies ranging from the 2030 Agenda for Sustainable Development (UNISDR, 2015) to the New
Urban Agenda (United Nations, 2017). As part of these strategies, we see the need for a better understanding of disaster risk and the need to consider the requirements of different categories of users.

**Table 1.** Overview of key global strategies calling among others for reducing risks and damage from disasters.

| Global strategies | Excerpts and statements | References |
|---|---|---|
| 2030 Agenda for Sustainable Development; Sustainable Development Goals (SDGs) | - 17 goals for improving human society, ecological sustainability and the quality of life are aiming to contribute to the global risk reduction agenda<br>- 25 targets related to disaster risk reduction in 10 of the 17 SDGs<br>- Among others, the objective of reducing the in number of deaths and people affected as well as decrease of economic losses caused by disasters is addressed in goal 11: "*Make cities and human settlements inclusive, safe, resilient and sustainable*" | UNISDR, 2015a<br><br>UNISDR, 2015b, p. 2<br>UNISDR, 2015a, p. 24 |
| Sendai Framework for Disaster Risk Reduction 2015-2030 | - Outlines 7 targets and 4 priorities for action to prevent new and reduce existing disaster risks<br>- Priority 1: Understanding of disaster risk<br>"*Policies and practices for disaster risk management should be based on an understanding of disaster risk in all its dimensions of vulnerability, capacity, exposure of persons and assets, hazard characteristics and the environment*" | UNISDR, 2015a, p. 14<br><br>UNISDR, 2015a, p. 14 |

| | | |
|---|---|---|
| | - Calls "*to promote the collection, analysis, management and use of relevant data and practical information and ensure its dissemination, taking into account the needs of different categories of users*" | UNISDR, 2015a, p. 16 |
| | - Advocates "*to support the development of local, national, regional and global user-friendly systems and services*" | UNISDR, 2015a, p. 16 |
| Paris Agreement | - International treaty on climate change | United Nations, 2015b |
| | - Calls for "*reducing vulnerability to climate change*" in article 7.1 | United Nations, 2015b, p. 9 |
| | - Calls for the "*importance of averting, minimizing and addressing loss and damage associated with the adverse effects of climate change [...]*" in article 8.1 | United Nations, 2015b, p. 12 |
| New Urban Agenda | - Addresses various field of action and calls for strengthening resilience in the event of disasters | United Nations, 2017 |
| | - Envisages cities and human settlements that "*adopt and implement disaster risk reduction and management, reduce vulnerability, build resilience and responsiveness to natural and human-made hazards and foster mitigation of and adaptation to climate change*" | United Nations, 2017, p. 7 |

Global goals are set to mitigate risks and damage from disasters. With this demand and in line with the objective to provide user-friendly systems, we seek to address this challenge by presenting a conceptual approach that allows users to analyse the impact of various natural hazards. This is supported by a study recently performed by Šakić Trogrlić et al. (2022). Among other questions, they asked in a survey how the natural hazard community could support the implementation of the SDGs. Enhanced stakeholder engagement, communication and knowledge transfer was rated at first place (39 % from 350 replies), followed by increased

management and reduction of disaster risks (34 %), and enhanced interdisciplinary research and its translation to policy and practice (29 %) on the third place.

Following this introduction, Sect. 2 presents the conceptual approach to developing a scenario-based multi-risk assessment tool. With the aim of developing a demonstrator (and not a fully operational system), we focused on analysing the physical vulnerability (e.g., Fuchs et al., 2018) of buildings (i.e., the likelihood that assets will be damaged or destroyed when exposed to a hazard

event), and the systemic vulnerability (e.g., Pascale et al., 2010; Hernandez-Fajardo and Dueñas-Osorio, 2013) of electrical power networks (i.e., probability of failure of interconnected systems given hazard intensities). Sect. 3 describes the results and steps taken, including findings from the user perspective. The discussions and conclusions are outlined in Sect. 4.

## 2 Conceptual approach

Considering the above-mentioned guidelines and strategies in the context of disaster risk reduction (DRR) and disaster risk

management (DRM), as well as the outlined research needs, we present a conceptual approach developed within the research projects RIESGOS and its successor RIESGOS 2.0 (Schoepfer et al., 2018; Schoepfer et al., 2024). The projects focused on the development of innovative scientific methods for the *assessment of multi-risk situations* with the aim of designing an approach that meets the needs of users at the local level. In addition to the German team coming from various disciplines, the project collaborated with a variety of research institutions and public authorities in Chile, Peru and Ecuador. This collaboration, both

with potential users and stakeholders across different levels, frames the novelty of the approach towards its practical applicability.

The starting point of our conceptual approach is the finding that local risk situations and the challenges for decision-makers to pursue global risk reduction goals in practice can vary across the globe. Thus, there is a gap between scientific and technical possibilities (i.e., the knowledge created by them and concrete fact-based decisions in the planning or political field). The conceptualization of this overall approach is visualized in Fig. 1.

- First, we conducted a context and stakeholder analysis to understand the organizational environment and underlying structures of the disaster risk governance and to identify stakeholders to engage (Sect. 2.1).

- A concept for a scenario-based multi-risk information system was developed (Sect. 2.2). We selected a story-based scenario concept that allows the description of a specific multi-risk situation and its representation through multiple scenarios (Sect. 2.2.1).

- As input for the demonstrator tool, the elements of risk (hazard, exposure, and vulnerability) and their impacts on critical infrastructure were considered in terms of their potential implementation (Sect. 2.2.2). In the process, we devoted efforts to the study of interactions at the physical and systemic vulnerability levels of cascading hazards, addressing cumulative damage and loss.

- During the development of the demonstrator for a multi-risk information system, we involved potential users from the
beginning to ensure that the designed tool meets their requirements and needs (Sect. 2.2.3). For the demonstrator we chose a decentralized system architecture approach built on distributed web services, with a graphical user interface as the frontend (Sect. 2.2.4).

During the project individual results have already been published and are cited accordingly. In this paper, we aim to present the overall approach, with focus on the feedback process from the user perspective showing the practical relevance of the designed
tool. We are convinced that such a user-oriented approach for exploring, describing and quantifying different *What-if scenarios* can constitute a valuable tool for understanding complex multi-risk situations and to prepare for such situations.

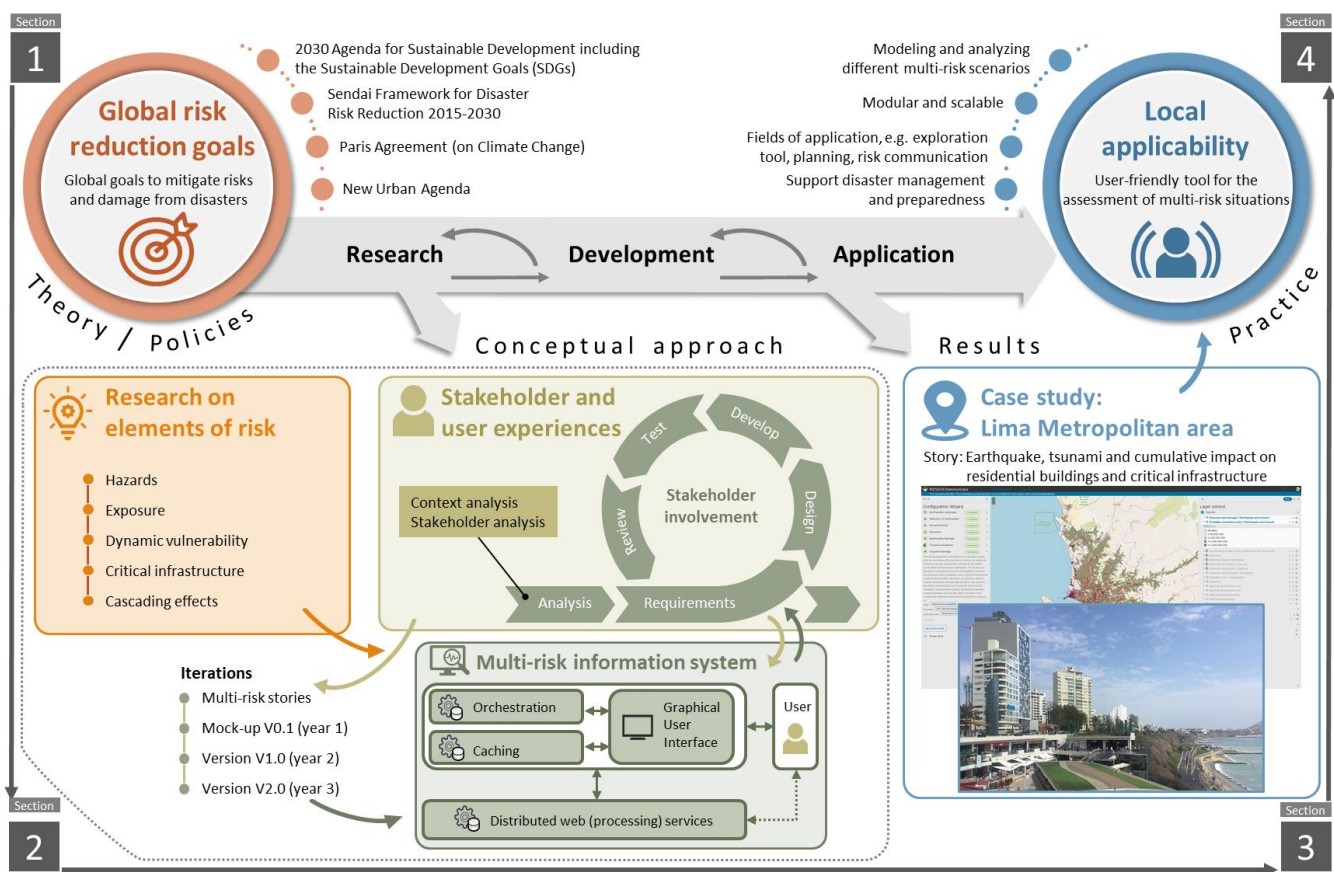

**Figure 1.** Conceptualization and workflow of the development of an analysis tool aiming at the implementation of global risk reduction goals (from theory / policies) on local level (to practice). The numbers in the boxes indicate the corresponding sections of the paper (Sect. 1-4). Photograph taken by Elisabeth Schoepfer (2019); screenshot of the tool (map data © OpenStreetMap contributors and available from https://www.openstreetmap.org).

## 2.1 Context and stakeholder analysis

Before starting the design of a tool or system, it is crucial to understand the context in which it is aimed to be used. A *context analysis* aims to understand the environment the work is placed in (Meaux and Osofisan, 2016).

To do so, we first defined the thematic context, i.e. here the disaster risk reduction (DRR) and disaster risk management (DRM) domains and the assessment of the risk profiles of the location or country. We started with the identification of disasters that have occurred in the country and their ranking according to frequency and impact by consulting existing and open geo-data sets (e.g., World Bank Open Knowledge Repository (World Bank Group, 2021); DRMKC INFORM (European Commission, 2023)). In doing so, we collected the information on historical disaster events with the aim of providing deeper insights on the dynamics of possible hazard scenarios. We focused on complex situations where hazardous events were observed to have interacted and caused cascading effects in the past, and that due to the increasingly exposed people and infrastructure can cause more damage and losses if they occurred again (Sect. 2.2.1). Secondly, a detailed analysis of the DRM policies, structures, strategies, and plans was

conducted. This included the documentation of frameworks and regulations in the DRM domain as well as the respective mechanisms for coordination and cooperation in the corresponding country. Next to the country-specific instruments also
activities in international cooperation were considered.

A *stakeholder analysis* has been done to identify relevant actors involved in the DRM context, describing their roles, responsibilities, relationships, interests, and relative influence / power. Naturally, the stakeholders belong to different sectors, i.e.:

     (1) research community,

     (2) institutions operating hazard information and monitoring systems,

(3) institutions operating DRM information systems,

     (4) institutions working on local and regional level in DRM contexts, and

     (5) institutions working on national planning level.

Key stakeholders per group were identified and described in detail on different levels ranging from national and regional to local level covering their specific objectives and tasks in the working contexts (Tab. 2).


**Table 2.** Stakeholder groups and stakeholders involved in the feedback process from the user perspective in Peru.

| Stakeholder group | Peruvian stakeholders involved in the feedback process |
|---|---|
| Research community | - Universities<br>- Research institutes |
| Institutions operating hazard information and monitoring systems | - Geological institute<br>- Geographical institute<br>- Geophysical institute<br>- Oceanographic institute |
| Institutions operating DRM information systems | - National institutions for risk analysis, risk reduction and risk mitigation<br>- National civil protection agency |
| Institutions working on local and regional level in DRM contexts | - Disaster management authorities<br>- Municipalities<br>- Non-governmental organisations (NGOs) |
| Institutions working on national planning level | - National Center for Strategic Planning<br>- Ministry of Housing, Construction and Sanitation<br>- Ministry of Transport and Communications |

## 2.2 Framework to design a multi-risk information system

As briefly outlined in Sect. 1, although there are several approaches on how to address multi-hazard risk situations, they are often
insufficient for a practical application. Suggestions for future research include developing of support tools adapted to different stakeholders, considering multiple hazards and dynamic aspects, and adequate communication of results (e.g., Curt, 2021; Cremen et al., 2022). These tools should enable the analysis of escalation effects and multi-level scenarios. After reviewing the current research landscape and following the recommendations, our overall objective was to develop such a multi-risk approach that considers the treatment of cascading effects. We set out to build a tool which allows the user to simulate and analyse complex
multi-risk situations from the perspective of *What-if scenarios* on a local level. With this, we aim to provide users the possibility

to explore various scenarios, and not only focusing on one fixed scenario (often referred to as reference scenario). Following this deterministic approach, we decided against a probabilistic assessment where all possible scenarios are combined (OECD, 2012). During the design of the tool we involved various stakeholder groups to ensure that the tool is geared towards the needs of potential users and its practicality. Our guiding questions in the design process were:

(1) "*How can natural hazards (e.g., earthquake and tsunami) that occur in close temporal succession or that trigger each other (cascading or consecutive) be described and represented considering their combined impacts?*",

    (2) "*What is the cumulative impact of such multi-hazard events and how is the impact amplified compared to single hazards, e.g., damage on residential buildings and/or critical infrastructure?*".

### 2.2.1 Story-based concept planning

With these objectives in mind, we followed the concept of story and scenarios to understand and describe possible multi-risk situations (e.g., Jarke et al., 1998; Sutcliffe, 2003).

With the term *story* we refer to a "*narrative description of a situation, defining the specific involved hazards, cascading effects and impacts, looking at a specific area of interest*". These stories represent realistic multi-risk situations with cascading effects. We ensure physically sound settings of the multi-hazard situation by performing all calculations with identical fault parameters

for both earthquake and tsunami simulations. However, although a story is based on physical drivers – i.e., natural hazards, – it is not limited to their description alone. Instead, a story should also incorporate the aspects of damage and losses in a realistic way as well as the impact, e.g., on critical infrastructures.

The term *scenario* represents for us a single (numerical) realisation or expression within a story (e.g., Li et al., 2016). Scenarios represent different intensities of the triggering natural hazards and their effects. For each chosen story, multiple scenarios are

available to describe different intensities of the triggering natural hazards and their effects. The quantitative models in the individual scenarios do not necessarily represent the entire complexity of a story. To which degree a story agrees with realistic circumstances depends on the modelling capabilities as well as on the availability of (geo-)data. Limited reproducibility should not, however, diminish the importance of qualified stories.

In summary, the selection of the multi-risk story is of crucial importance. It is the basis of our designed multi-risk approach.

### 2.2.2 Research on elements of risk

The story descriptions need to be matched with the scientific-technical potential of research. Research on multi-risk requires a thorough understanding of the three risk components, *hazard*, *exposure* and *vulnerability,* but mostly important, their interrelations (cf., Gill and Malamud, 2017).

*Hazard* is defined as "*a process, phenomenon or human activity that may cause loss of life, injury or other health impacts,*

*property damage, social and economic disruption or environmental degradation.*" (UNDRR, 2022b, p. 7). UNDRR (2022b) further differentiates between natural, anthropogenic or socio-natural hazards. In the scope of our project, we focused on *natural hazards*, and geophysical hazards in particular. For example, in seismic hazard assessment, future earthquake risks require access

to existing earthquake catalogues (e.g., Nievas et al., 2020). They contain few parameters that allows to simulate the geometry and intensity of similar future earthquake ruptures. Their spatially distributed intensities (i.e., seismic ground motion fields) are typically simulated through statistical or numerical models that are constantly updated thanks to the current instrumentation initiatives (e.g. Weatherill et al., 2023). Moreover, since some earthquakes can trigger a tsunami, numerical tsunami models are typically used to determine the wave propagation and to estimate the flow depth in the inundated coastal areas (e.g., Rakowsky et al., 2013). This multi-hazard interaction of an earthquake triggering a tsunami is one example of many possible hazard interactions (cf., Gill and Malamud, 2016, Fig. 4, p. 672). Other interactions are for example volcanic activities (e.g., Plank et al., 2018), that, depending on the geographical and climatic framework, can also cause landslides, lahars and/or floods (e.g., Frimberger et al., 2021).

On the other hand, *exposure* describes all elements that can be subject to loss or damage in a hazard zone, such as people, property or critical infrastructure (UNISDR, 2009, p. 15; Geiß and Taubenböck, 2013). Often, exposure data are outdated, spatially aggregated and discontinuous, or are simply non-existent in many regions of the world. In addition, it is crucial to deal with the dynamic change processes of settlement areas induced by, for example, rapid population growth and increasing urbanization (cf., Taubenböck et al., 2012; Geiß et al., 2019). To overcome this bottleneck, approaches have been developed in the past to combine relevant information from spatial data such as earth observation and (geo-)statistics to create detailed exposure information (e.g., Wieland et al., 2012; Geiß et al., 2014, 2015, 2017, 2022). First, the physical-structural and non-structural characteristics of buildings are identified (e.g., Geiß et al., 2017; Aravena Pelizari et al., 2021) and in the following their *vulnerabilities* are estimated (e.g., physical vulnerability, Gómez Zapata et al., 2021b, 2022a). Particularly, when working with multi-hazard events *dynamically changing vulnerabilities* considering cumulative damages, e.g., on buildings caused by earthquakes and tsunamis, need to be assessed (Gómez Zapata et al., 2022b, 2023).

The negative effects of disasters may cause a failure or disruption of critical infrastructures, and they are not geographically limited to the area directly affected by the disaster. For example, in power networks a failure of one of its components can trigger a cascade of failures in other components (cf., UCTE 2004; FIUBA, 2020). Furthermore, the dependence to other critical infrastructures, such as water supply pumps, traffic signals, SCADA systems (acronym for Supervisory Control And Data Acquisition), can extend the negative effects of the disaster in unforeseeable ways (Rinaldi et al., 2001). For studying the *impacts on critical infrastructure,* different approaches analyse the fragility of infrastructure components (cf., FEMA, 2003; Pitilakis et al., 2014), simulate cascading failures (Crucitti et al., 2004; Hernandez-Fajardo and Dueñas-Osorio, 2013), assess the criticality of infrastructures (e.g., Greiving et al., 2021) and propose frameworks for probabilistic risk analysis (cf., Ferrario et al., 2022; Rosero-Velásquez and Straub, 2022).

In addition, such negative effects persist over the time until the failures and disruptions are repaired. The longer it takes to resume the normal operation of critical infrastructure, the larger the impact to the economic activity becomes. Therefore, risk is also determined by the recovery of critical infrastructure after the disaster. The study of the *resilience of critical infrastructure* combines models for simulating the impact of natural hazards and the recovery process thereafter (FEMA, 2003; Ouyang et al.,

2012; Sedzro et al., 2018). It also supports a risk analysis considering not only the direct impact to the infrastructure, but also the indirect consequences to the society (Bruneau et al., 2003).

### 2.2.3 Feedback process from the user perspective

For the development of the presented information system for multi-risk assessment, we chose an agile software development approach (cf., Kent et al., 2001). This methodology is based on an iterative development approach where user requirements can be updated and considered for the system development through close interaction. This allows the co-creation of a system where the role of the user shifts from consumers of information to informants for needs on how systems are developed (cf., Gomillion, 2013).

The call for user involvement is not new (Kling, 1977; Norman, 1986) and has been suggested to be treated as one of several means for information systems development projects to be more successful (e.g., He and King, 2008; Bano and Zowghi, 2014). Kujala (2003) conducted a review of benefits and challenges of user involvement finding that interaction with users has various positive effects, especially on user satisfaction. This was confirmed by Bano et al. (2017) who performed an empirical exploration of user involvement in software development and concluded that user satisfaction and the resulting system are mutually constituted. Being aware that user requirements can sometimes be contradictory, it was considered valuable for the development process to evaluate the necessity and consequences of each requirement. Accordingly, we geared our approach to the needs of potential users and its practicality (cf., user-centred design; Gould and Lewis 1985; Karat, 1997) throughout the design and development process (Fig. 2).

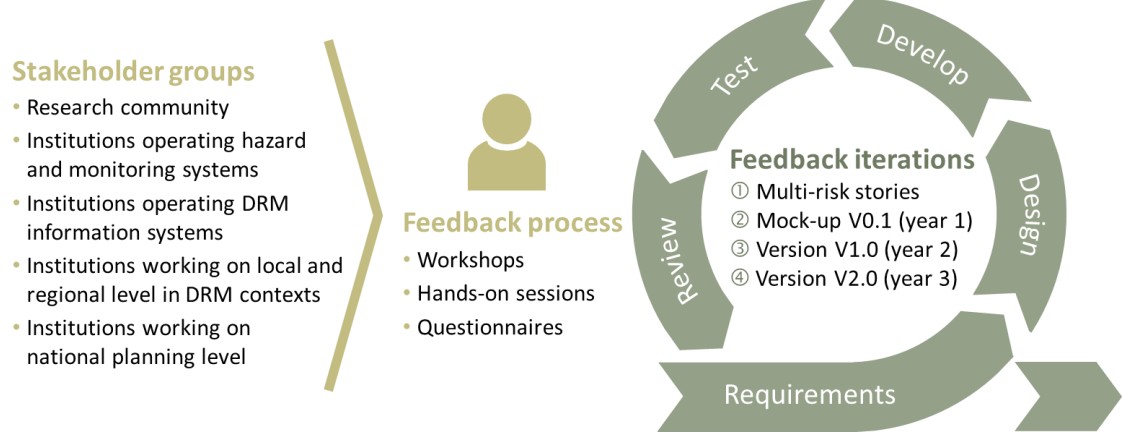

**Figure 2.** Detailed graphical representation of the stakeholders and user involvement in the design and development process (compare Fig. 1). The five different stakeholder groups were involved in the four iterations, with requirements and feedback gathered mainly through workshops and hands-on sessions, as well as questionnaires.

The development of our multi-risk assessment tool was based on a structured and systematic feedback process from the user perspective. Thereby the goal was to target various representatives from different stakeholder groups. The iterative design and

development process can be broken down into four iterations, each of which was accompanied by a feedback mechanism to define further requirements and to reassess and adjust existing ones.

(1) Starting point for the approach is the definition of *multi-risk stories* with the potential users (Sect. 2.2.1). The joint discussion with the different stakeholder groups was intended to ensure the realism and relevance of the stories, thus elaborating a common starting point that will allow structured discussions throughout the design and development process of the tool to capture the requirements from the user's point of view. Researchers must ensure that the processes of the multi-risk story are described in a scientifically sound (and possibly abstracted) way. These serve as input, definition and enhancement of the tool and its functionalities.

(2) As a second step, a *mock-up* (version V0.1) was used to visualize the envisaged tool. Therefore, we designed a graphical user interface representation and visualized the possible functionalities and outputs for each step in the multi-risk chain. This allowed the user to get a sense of the tool, even if the individual buttons were not yet functional. This mock-up proved particularly useful for discussing the planned features, getting feedback, and collecting requests for changes.

(3 and 4) In the following, we conducted the feedback process along the two *functional versions* of the tool, i.e. versions *V1.0* and *V2.0* (Fig. 2). Methods for assessing the feedback and requirements ranged from collaborative workshops including practical hands-on sessions to questionnaires and market research.

A set of guiding questions was developed which cover a broader spectrum of aspects regarding (1) information content, (2) user interface and (3) usability and applicability to evaluate how scientific research can be made applicable through a practical tool for the assessment of multi-risk scenarios. We collected the responses to these questions via a questionnaire, which are provided to the participants during workshops (*quick assessment*). While evaluating the tool either during practical hands-on sessions in workshops or without guidance over a certain period after these workshops an additional questionnaire (*detailed assessment*) s used. When formulating the questionnaires for the different development steps of the tool, we aim to maintain key questions throughout the entire evaluation period. Other questions have been changed or replaced in subsequent versions of the questionnaire as they were evaluated as no longer relevant. The questionnaires covered up to 48 specific questions as well as three open questions to describe the overall satisfaction with the tool (see the Supplement). Information about the personal profile, i.e., the work area and function / role of the respondents was gathered while ensuring the data protection rights of individuals.

### 2.2.4 Scenario-based system development using distributed web services

We aimed for an approach that is applicable and adaptable to different multi-risk situations, geographic areas and scales. With this objective in mind, various system architectures can be considered. Here, we decided to create a system based on a decentralized service-oriented architecture (SOA) using distributed web services. Among other factors we selected this approach because of the following three reasons:

(1) Web services can be combined to form a chain representing different multi-risk situations leading to modularity, flexibility, and scalability,

(2) exchange of models / data between institutions are facilitated as data do not need to be handed over ensuring that expertise remains with the experts, and

(3) the data and models are up-to-date as they remain at specialized institutions.

A key element of this system approach is the use of independent web services which allow to visualize the results coming from various models from research (Sect. 2.2.2). We designed a tool which consists of

(i) distributed web (processing) services,

(ii) a workflow control (orchestration and caching) that links the web services into value chains to map complex multi-risk scenarios, and

(iii) a graphical user interface (GUI) that allows users to interactively run various scenarios (Fig. 3).

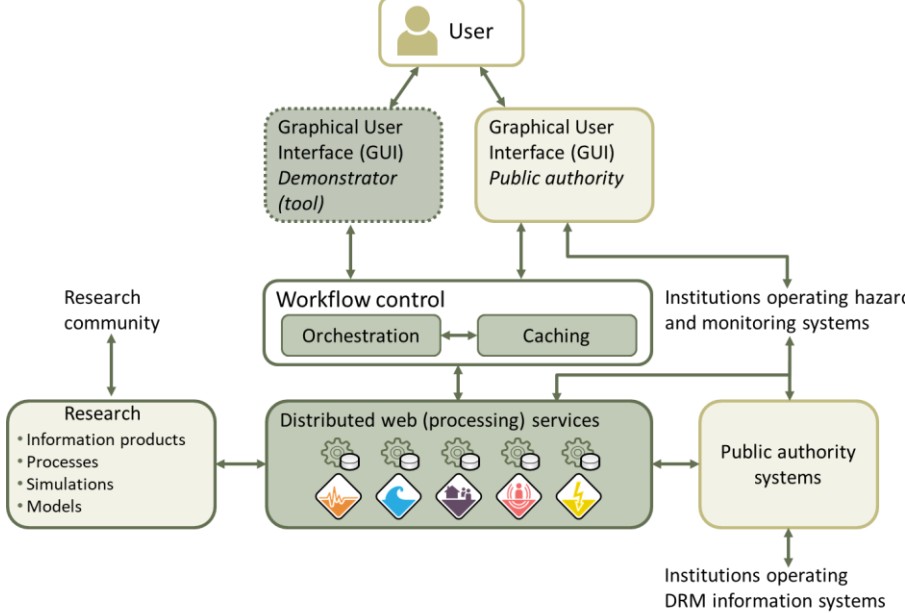

**Figure 3.** Elements and actors involved in the design and development of the multi-risk information system. The dotted line of the GUI of the demonstrator (tool) indicates its provisional character, serving for demonstration purposes only.

The interaction between the loosely coupled web services is achieved using Web Processing Service (WPS) interface standard directives published of the Open Geospatial Consortium (OGC; WPS, 2018). WPS are implemented in a flexible and scalable architecture based on Docker containers that encapsulate the running processes (Brinckmann et al., 2020). The interoperability between the different services is ensured by thorough harmonization of input and output formats and the use of on-the-fly converters. Dedicated WPS create simulations of intensity maps for specific hazards, either on the fly (e.g., for earthquake ground motion simulation) or by querying a list of pre-simulated events (e.g., for tsunami inundation maps) (Pittore et al., 2020).

For the graphical user interface (frontend) we created a web-based application (accessible via a web browser) with the aim (1) to allow users to specify the inputs to a model, trigger its execution and display the results of the models; (2) to chain a set of models into a scenario that represents the multitude of processes describing a complex risk situation (e.g., earthquake causing damage in

buildings and triggering a tsunami); and (3) to facilitate the user's exploration of the range of impacts that one or more natural hazards may have. In addition to the control layer, which orchestrates the various web services, browser caching and WebGL-rendering were introduced as another cross-cutting functionality of the tool to speed up the display of large amounts of data in the browser. To address different user needs, the GUI was split in an expert and non-expert viewer. The expert viewer ('demonstrator') allows individual setting and configurations of model parameters and outputs, whereas the non-expert viewer ('demonstrator light') runs with predefined parameters and a simplified visualization of results. The underlying web services are identical. Additionally, the 'demonstrator light' provides three modes which allows side-by-side comparison of two scenarios. The 'demonstrator light' provides three different modes for the user to select from:

(1) Analysis of one multi-risk scenario,

(2) comparison of two different scenarios within one multi-risk story two (e.g., earthquakes of different magnitudes), and

(3) analysis of different time steps within a multi-risk scenario.

For the developments, both backend (web services) and frontend (graphical user interface), we aimed for open source. This allows others to not only use this software but also to replicate the tool and to develop it further.

## 3 Results and experiences using the example of Lima Metropolitan area, Peru

### 3.1 Study area

Peru is highly exposed to natural hazards such as earthquakes, tsunamis, floods, mass movements (e.g., landslides, avalanches), strong winds, heavy rains, fires and low temperatures (INDECI, 2020). The implementation of our approach is shown for the example of Lima Metropolitan area, Peru. Together with the adjacent port city of Callao, the capital city of Peru, Lima, has nearly 11 million inhabitants representing approximately one-third of Peru's total population (INEI, 2022) (Fig. 4). This region is threatened by strong earthquakes and tsunamis originated from the Andean subduction zone, one of the longest continuous subduction zones on earth (cf., Rodríguez et al., 2020). In the past, the capital city of Peru was hit by significant earthquakes causing tsunami run-ups over 24 meters, e.g. in 1586 (Mw 8.1), and 1746 (Mw 8.6) (Kulikov et al., 2005; Olarte et al., 2008). Notably, a major part of the road network runs along the tsunami-prone coast. The same applies for the main port of Callao and the nearby Jorge Chavez International Airport (CENEPRED, 2017). Lima Metropolitan area is further exposed to critical infrastructures, such as water supply, wastewater disposal, IT and telecommunication, or electricity whereas a failure of electricity supply has the quickest negative impact on other sectors (cf., Greiving et al., 2021).

The approach is demonstrated for Lima Metropolitan area which is composed of five sectors (INEI, 2022), i.e. Lima Norte (8 districts), Lima Sur (11 districts), Lima Este (9 districts), Central Lima (15 districts) and Callao (7 districts) (Fig. 4). The multi-risk story (see Sect. 3.2) including its cascading impacts is applied for this particular case study area.

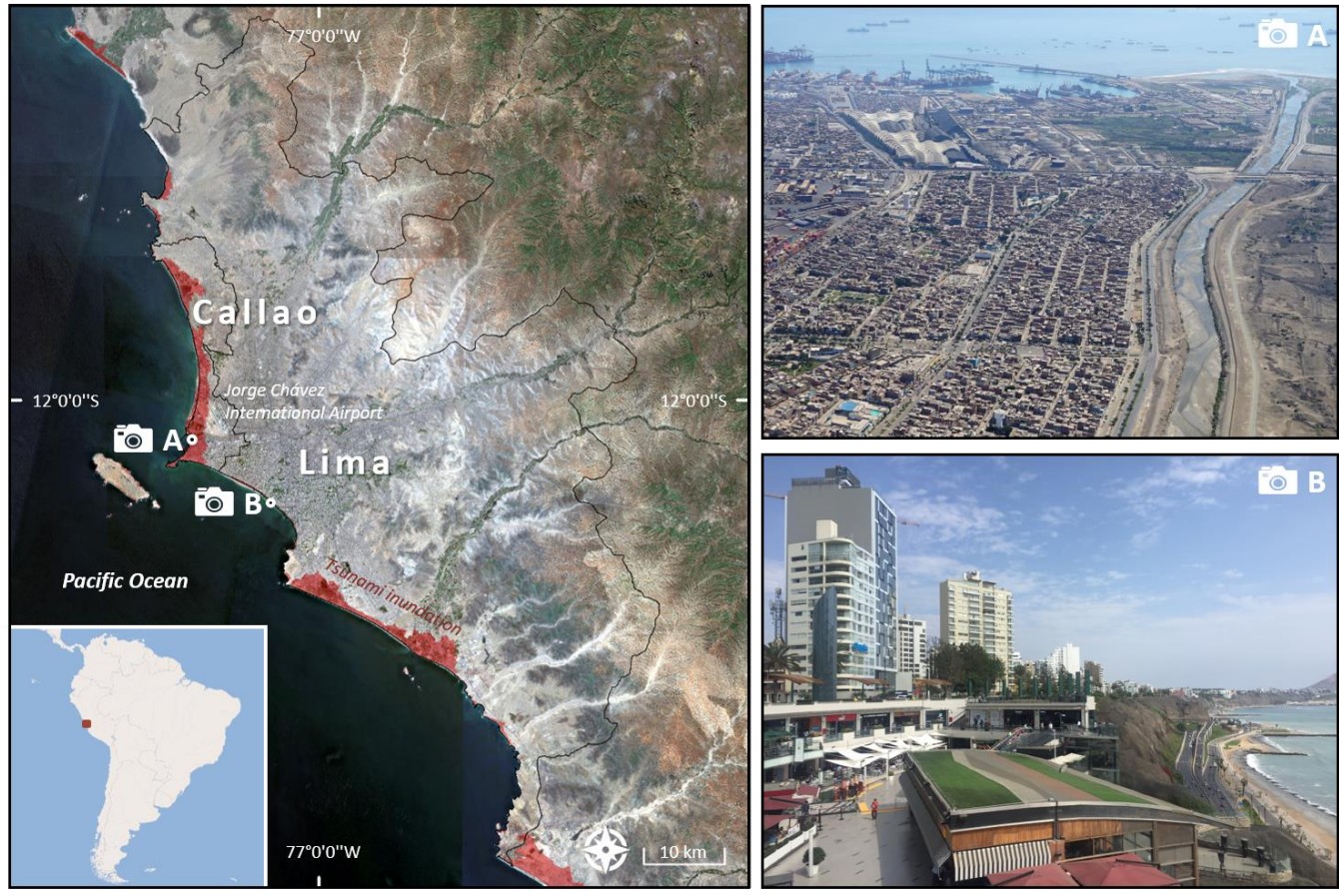

**Figure 4.** Multispectral satellite image (left) of the study area of Lima Metropolitan area, Peru, with administrative boundaries (black outlines). Additionally, an estimated tsunami inundation exemplified for a Mw 8.9 earthquake (generated with TsunAWI; Harig et al., 2008) is shown. Coastal features such as the harbour in Callao (upper right) and the coastal highway in Miraflores (lower right) are displayed. Photographs taken by (A) Torsten Riedlinger (2018) and (B) Elisabeth Schoepfer (2019). Map data © Copernicus Sentinel data 2023, processed by ESA; © DLR, EOC Basemap Map Service 2023; © Instituto Geográfico Nacional, Limites Distritales 2022.

### 3.2 Story-based concept design

Following the story-based concept design (Sect. 2.2.1) we characterized the various elements composing the multi-risk situation which was defined in consultations with Peruvian stakeholders (workshop held in Lima on 20 April 2018; see Sect. 3.4). During this consultation process, stakeholders recommended to consider the reference (worst-case) scenario of an 8.8 Mw earthquake off coast of Lima Metropolitan area as documented by INDECI (2017), when defining the following story: "*Strong shaking occurs in Lima Metropolitan area, Peru, during the day time. There are severe damages on buildings and infrastructure, many people are directly affected by building collapses. As the earthquake has the potential to trigger a tsunami, a tsunami warning is issued and evacuation to safe areas is announced. Coastal roads and roads to highlands become progressively congested. In the following a first tsunami wave impacts the coast and starts inundating parts of the harbour area in Callao. Because of the numerous building collapses, city roads become less suitable for prompt evacuation.*"

For this defined story, multiple scenarios including historical, observed as well as stochastically earthquakes, were made available. Each earthquake scenario serves as a trigger for the defined multi-risk chain resulting in different cascading impacts. A flow chart (Fig. 5) was created conceptualizing the main logic, its components and information flows of the multi-risk story.

A database of historical, observed and stochastically distributed earthquakes with different locations and magnitudes was developed and made available via a web service. As each of these individual earthquakes serves as a trigger for the defined multi-risk chain resulting in cascading impacts of different degrees, the user can analyse scenarios of varying severity by choosing a specific earthquake from the database. We applied a similar approach for a multi-risk story on volcanic activities with compound hazards, i.e. ashfall and lahars, with damage on buildings and impact on the power network in a case study for the volcano

Cotopaxi in Ecuador (see Sect. 4, vii). Here, too, users were able to analyse several scenarios of varying severities based on different VEI values (cf. Gómez Zapata et al., 2021a).

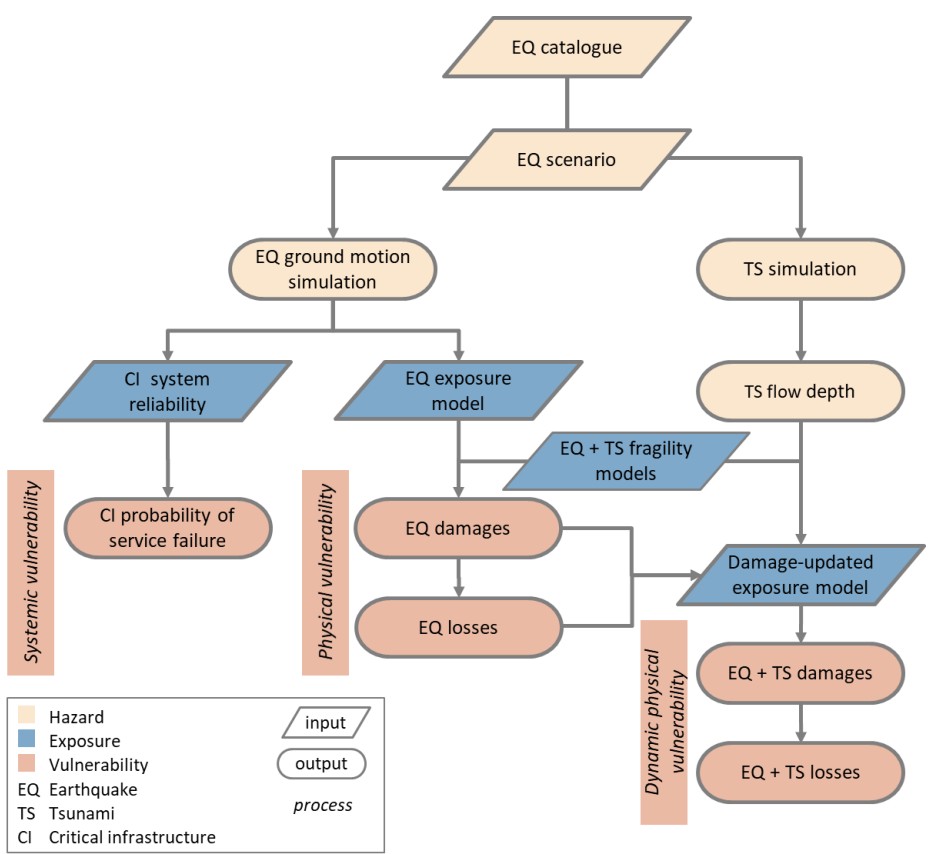

**Figure 5.** Flowchart of the multi-risk story for an earthquake / tsunami (Harig and Rakowsky, 2021) event affecting housing and the critical
infrastructure power grid.

### 3.3 Demonstrator for a multi-risk information system

### 3.3.1 Web services and workflow control

Each step in the flowchart (Fig. 5) is represented by one or more models offered as web services with corresponding digital object identifiers (DOI). In the multi-risk story exemplified in the study area of Lima Metropolitan area, the starting point for the hazard

is an earthquake catalogue (Pittore et al., 2021a) where the user can choose between different scenarios, including historical and observed (compiled by project by the Global Earthquake Model (GEM) Foundation in the framework of the SARA project (i.e., GEM Secretariat, 2015; SARA, 2016a; 2016b; CERESIS, 1995; Tavera et al., 2001; Leyton et al., 2009) and stochastically earthquakes (i.e., followed the approach outlined in Aristizábal et al., 2018). A single event within the catalogue form an earthquake scenario that is compatible with the probabilistic seismic hazard model proposed for the study area. It is described

through basic parameters, such as the epicenter location (longitude, latitude) together with hypocenter depth (in kilometres), moment magnitude (Mw), and rake, dip, and strike angles, that together are used to model finite earthquake ruptures using some openly available tools of the OpenQuake Engine (Pagani et al., 2014; Silva et al., 2014). The user can select one of these available earthquakes which in the following triggers the subsequent web service. To do so, we have decided to simulate its correspondent seismic ground motion fields through the adoption of suitable GMPEs (ground motion prediction equations) for the specific

tectonic context of the subduction inter-face tectonic regime. These spatially distributed seismic ground motion fields are simulated for the selected earthquake scenario. They are generated through the Shakyground web service (Weatherill et al., 2021), that for case of Lima Metropolitan area, uses the Montalva et al. (2017) GMPE in terms of expected accelerations (i.e., peak ground acceleration (PGA) and spectral periods (SA) 0.3, and 1.0 seconds). This web service was constructed based on the QuakeML data formats (Schorlemmer et al., 2011) and the OpenQuake Engine (Pagani et al., 2014; Silva et al., 2014). Examples

of these scenario-based ground motion fields are available in Gómez Zapata et al. (2021c). For those earthquakes which can potentially trigger a tsunami, another web service is introduced which provides access to pre-calculated numerical tsunami simulations. The simulations were generated using the physical generation and propagation model TsunAWI (Harig et al., 2008), which accounts for a triangular mesh with variable resolution as proposed by Harig et al. (2020). The size of the tsunami is related to the magnitude of the selected earthquake. Generally, larger earthquakes result in larger values of the wave amplitude at the

coast and broader inundation area. However, the relation is rather complex, since we account for the vertical displacement of the coastal area due to the earthquake, which might affect the inundation, and additionally, the run-up process is highly nonlinear. Based on the earthquake catalogue, a database of tsunami scenarios with earthquake sources offshore Peru was calculated. In case of the historic earthquake from 1746, we account for uncertainties by incorporating several scenarios covering a range of fault parameters for the source area as suggested by Jimenez et al. (2013). The available outputs including the maximum tsunami

amplitude, arrival times and tsunami inundation depth are displayed (Rakowsky et al., 2013; Androsov et al., 2024; Harig et al., 2024). Some of these scenario-based tsunami inundation maps are available in Harig and Rakowsky (2021), respectively.

In order to assess the exposed elements of interest (e.g. residential buildings), exposure models are constructed. They provide information on the location, spatial aggregation and typologies of the residential building stock of Lima Metropolitan area (Yepes-

Estrada et al., 2017). Each building typology has associated a fragility function (Villar-Vega et al., 2017) for both hazard-
vulnerability schemes (earthquake and tsunami), as documented in Gómez Zapata et al. (2021b). The demonstrator is able to
serve these exposure and fragility models through the scripts Assetmaster and Modelprop (Pittore et al., 2021b), which are used
as two web services. To assess the damage states of the residential buildings and losses (in terms of repairing costs of the
corresponding building class in US Dollars) after the occurrence of the selected earthquake the so-called damage exposure update
(web) service DEUS is triggered (Brinckmann et al., 2021). Using an updated exposure model that includes earthquake-induced
damages, and simulations of tsunami inundation depth as inputs, once again the DEUS web services is initiated to approximate
the expected cumulative damage and disaggregate the losses per hazard event (Gómez Zapata et al., 2023). This methodology
makes use of inter-scheme damage compatibility matrices, that can be consulted in Gómez Zapata et al. (2022c); and a set of
state-dependent tsunami fragility functions (Gómez Zapata et al., 2022d), that for the case of Lima Metropolitan area were
constructed after having modified the analytically derived ones originally proposed in Medina (2019).

Finally, the user can also receive information on the vulnerability of the power network showing a spatially distributed probability
of service disruption in the affected area (Rosero-Velásquez et al., 2022; Rosero-Velásquez, 2024). That probability is computed
based on a Monte Carlo simulation of cascading failures within the network, using the algorithm presented in Crucitti et al. (2004)
and Hernandez-Fajardo and Dueñas-Osorio (2013).

Table 3 provides detailed information on the system components (web services) with input and output information including
corresponding references. This set of web services documents the multi-risk sequence as visualised in Fig. 5. Interaction with the
web services is achieved using the Web Processing Service (WPS) interface standard guidelines published by the Open Geospatial
Consortium (OGC; WPS, 2018).

**Table 3.** System components (web services) with details on input data/model, source and output for the multi-risk story for an earthquake /
tsunami event affecting housing and the critical infrastructure power grid. EQ = Earthquake; TS = Tsunami; CI = Critical infrastructure.

| Web service | Input data/model and source | Output data/model |
| --- | --- | --- |
| EQ catalogue 'Quakeledger' (Pittore et al., 2021a) | - Earthquake catalogues as compiled by the SARA project for subduction events (Pagani et al., 2021).<br>- Filter parameters: depth, magnitude, and a geographic area that is defined by a bounding box upon user request. | - List of earthquakes for subduction interface that matches the filter criteria defined by the user. |
| EQ ground motion simulation 'Shakyground' (Weatherill et al., 2021) | - Earthquake source parameters (hypocentral location, depth, and strike, dip and rake angles).<br>- OpenQuake Hazard Library (Pagani et al., 2014) to generate finite fault ruptures as a function of their source properties.<br>- Ground motion prediction equation (GMPE) for subduction interface (e.g., Montalva et al., 2017).<br>- Gridded values of shear wave velocities for the uppermost 30 m depth (Vs30). For Lima, the dataset of Ceferino et al. (2018) was used. It compiles slope-based Vs30 (Allen and Wald, 2007), and the seismic microzonation for Lima defined by Aguilar et al. (2013). | - The demonstrator displays the ground motion fields of mean acceleration values for the target intensities (i.e., peak ground acceleration (PGA)). They are forecasted at each site of the Vs30 grid by the selected GMPE.<br>- Additionally, 1000 realisations of ground motion fields with uncorrelated and cross-correlated residuals for PGA and spectral accelerations at 0.3 and 1.0 seconds for six earthquake scenarios (Mw 8.5 - 9.0) are reported in the repository of Gómez Zapata et al. (2021d). |

| | | |
|---|---|---|
| EQ exposure model 'Assetmaster' (Pittore et al., 2021b) | - Official census dataset at the block level (INEI, 2017), which contains a few attributes for dwellings.<br>- 'Mapping schemes' that relate census attributes, dwellings-to-buildings fractions, and seismic-oriented building classes (GEM, 2014).<br>- Seismic-oriented residential building classes as defined by the SARA project, and their inferred replacement costs (Zschau et al., 2017).<br>- Focus maps that spatially combine tsunami inundation and population, which to generate exposure aggregation areas (Gómez Zapata et al., 2021e). | - Exposure model for residential buildings for earthquake risk applications reported in the repository of Gómez Zapata et al. (2021f).<br>- They are GEOJSON files that contain the building counts per type spatially aggregated at the block-level, and on CVT-based (Central Voronoi Tessellations) geocells. The metadata of these exposure files match the metadata of the fragility files served by 'Modelprop'. |
| TS precomputed simulations for each associated EQ using TsunAWI (Harig et al., 2008, Harig et al., 2024) | - Bathymetry by General Bathymetric Chart of the Oceans (GEBCO), GEBCO 08 Grid, 1km resolution, http://www.gebco.net.<br>- Coastal topography by Shuttle Radar Topography Mission (SRTM) SRTM, 30m resolution, https://www2.jpl.nasa.gov/srtm.<br>- Digital elevation model by TanDEM-X (Krieger et al., 2007), 12m resolution, https://www.dlr.de/en/research-and-transfer/projects-and-missions/tandem-x. | - Maximum tsunami amplitude (in meters).<br>- Arrival time (in minutes).<br>- Maximum tsunami inundation depth (in meters) (Harig and Rakowsky, 2021; Harig et al., 2024). |
| TS fragility model 'Modelprop' (Pittore et al., 2021b) | - Building fragility functions for seismic ground-shaking (Villar-Vega et al., 2017).<br>- Two types of tsunami fragility functions for buildings: analytical (Medina, 2019; Medina et al., 2019), and empirical (Suppasri et al., 2013). | - The fragility functions are expressed JSON files. Their metadata matches the exposure models served by "Assetmaster", and the earthquake and tsunami intensity measures of interest for subsequent risk assessment. |
| EQ 1st run of the software DEUS (Damage-Exposure-Update-Service) (Brinckmann et al., 2021) | - Ground motion fields (PGA; SA 0.3; SA 1.0 seconds) served by 'Shakyground'.<br>- Seismic-oriented exposure model for residential buildings served by 'Assetmaster'.<br>- Seismic fragility functions served by 'Modelprop' (state-independent). | - Spatial distribution of EQ damage in the form of a damage-updated exposure model. The damage scale of EQ is used.<br>- Spatial distribution of direct EQ losses (replacement costs in USD). Example outputs are compiled in Gómez Zapata et al. (2021c). |
| EQ + TS 2nd run of the software DEUS (Brinckmann et al., 2021) | - Raster files of TS inundation depth per EQ scenario, precomputed by TsunAWI.<br>- Damage-updated exposure model (containing EQ damage) served by 'Assetmaster'.<br>- - Building inter-scheme conversion matrices. They express the probabilistic compatibility between the EQ building classes and the TS ones. They were generated through the taxonomic disaggregation method of Gómez Zapata et al., (2022d). The script is available in Gómez Zapata et al., (2021d).<br>- Damage inter-scheme conversion matrices. They express the probabilistic compatibility between the EQ damage states and the TS ones. They were generated through the method of Gómez Zapata et al. (2023). The script is available in Gómez Zapata et al. (2022a).<br>- State-dependent tsunami fragility functions served by 'Modelprop'. They are generated by modifying the functions of Medina (2019). The script and files are available in Gómez Zapata et al. (2022d). | - Spatial distribution of EQ+TS damage in the form of a damage-updated exposure model. The damage scale of TS is used.<br>- Spatial distribution of direct EQ+TS losses (replacement costs in USD). Example outputs are compiled in Gómez Zapata et al. (2021c). |
| CI system reliability (Rosero-Velásquez, 2020; 2024) | - Ground motion fields served by 'Shakyground'. | - Probability of service failure (in percentages). |

- Seismic fragility functions for power network facilities (e.g., substations and power plants), based on HAZUS (FEMA, 2003).
- The power network topology and information were obtained (publicly available or upon request) by OSINERGMIN (2019) and COES (2019), and were adapted to the web service as shown in Merscher (2020).
- The calculation of the output is based on a network model for simulating cascading failures (Crucitti et al., 2004; Hernández-Fajardo et al., 2013).

### 3.3.2 Graphical user interface (GUI)

A graphical user interface (GUI) allows the user to independently explore the different risk scenarios making use of the above-mentioned web services. The designed GUI is available for the expert (Fig. 6) and non-expert user (Fig. 7). For the expert view ('demonstrator') the main display is divided into three areas: the map window in the centre, the configuration wizard that controls each web service on the left, and the results panel on the right (Fig. 6). In the configuration wizard, the user is guided through the multi-risk story where he can select different parameters according to his specific interests. In the layer control panel, the user can examine and view the processed results and gets more information about the outputs (e.g., legends, detailed descriptions). To maintain a solid overview, only the parameters relevant for the currently selected step are highlighted as active which enables intuitive control. In this way, the user does not lose track of the current step in the multi-risk chain, even with a long and complex multi-risk story. In the non-expert view ('demonstrator light') the user can select between three different modes (Fig. 7-9). The viewer shows a reduced configuration wizard including abstracted versions of the results (mode 1). The split-screen allows the side-by-side comparison of two selected scenarios (mode 2) or the exploration of different steps within one scenario (mode 3).

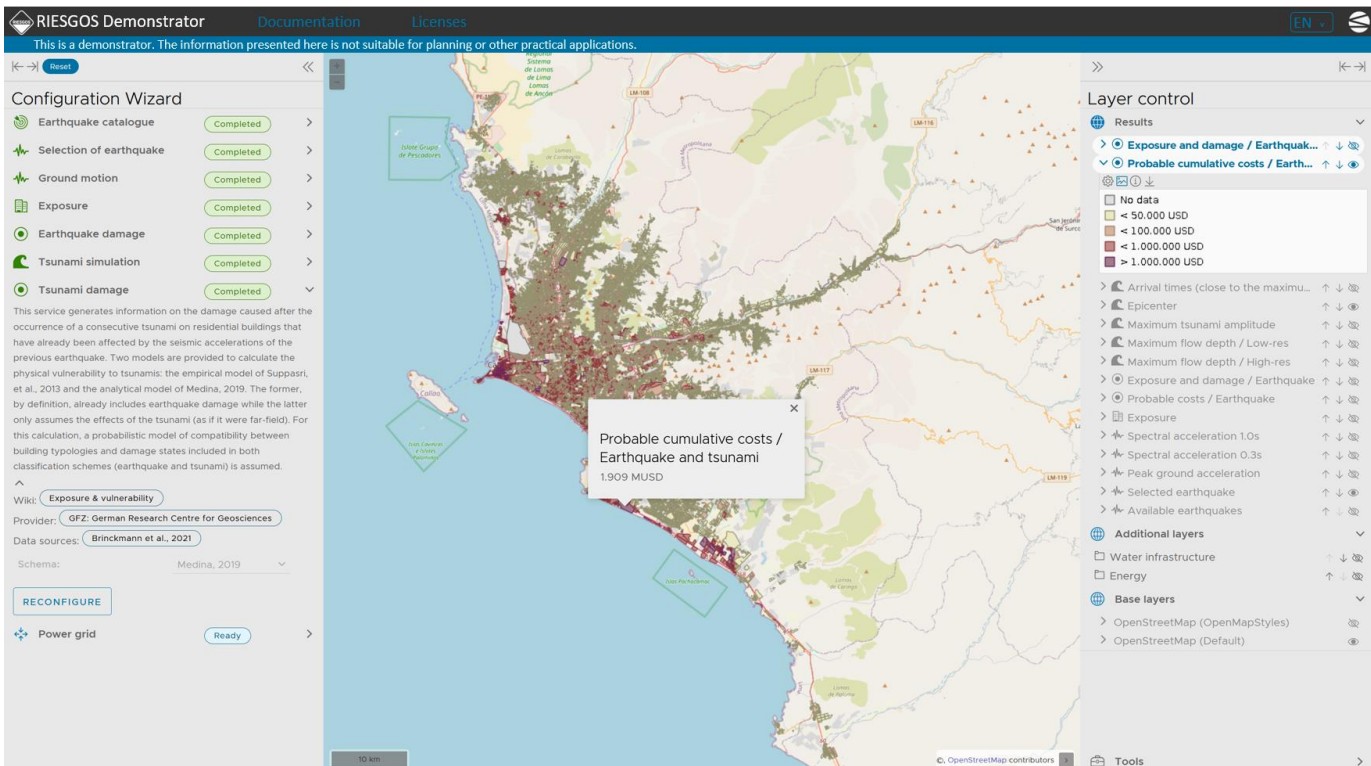

**Figure 6.** Graphical User Interface (GUI) of the information system (demonstrator) exemplified for the study area of Lima Metropolitan area, Peru (as of November 2022). The main screen is divided into three main display areas, i.e. the central map window, the configuration wizard for the control of each web service to the left, and the results panel to the right. The GUI is available in English and Spanish. The screenshot shows the accumulated damage (in terms of repairing costs of the corresponding building class in US Dollars) after the occurrence of the selected earthquake and tsunami. The exposed elements of interest are residential buildings within the study area. Map data © OpenStreetMap contributors and available from https://www.openstreetmap.org.


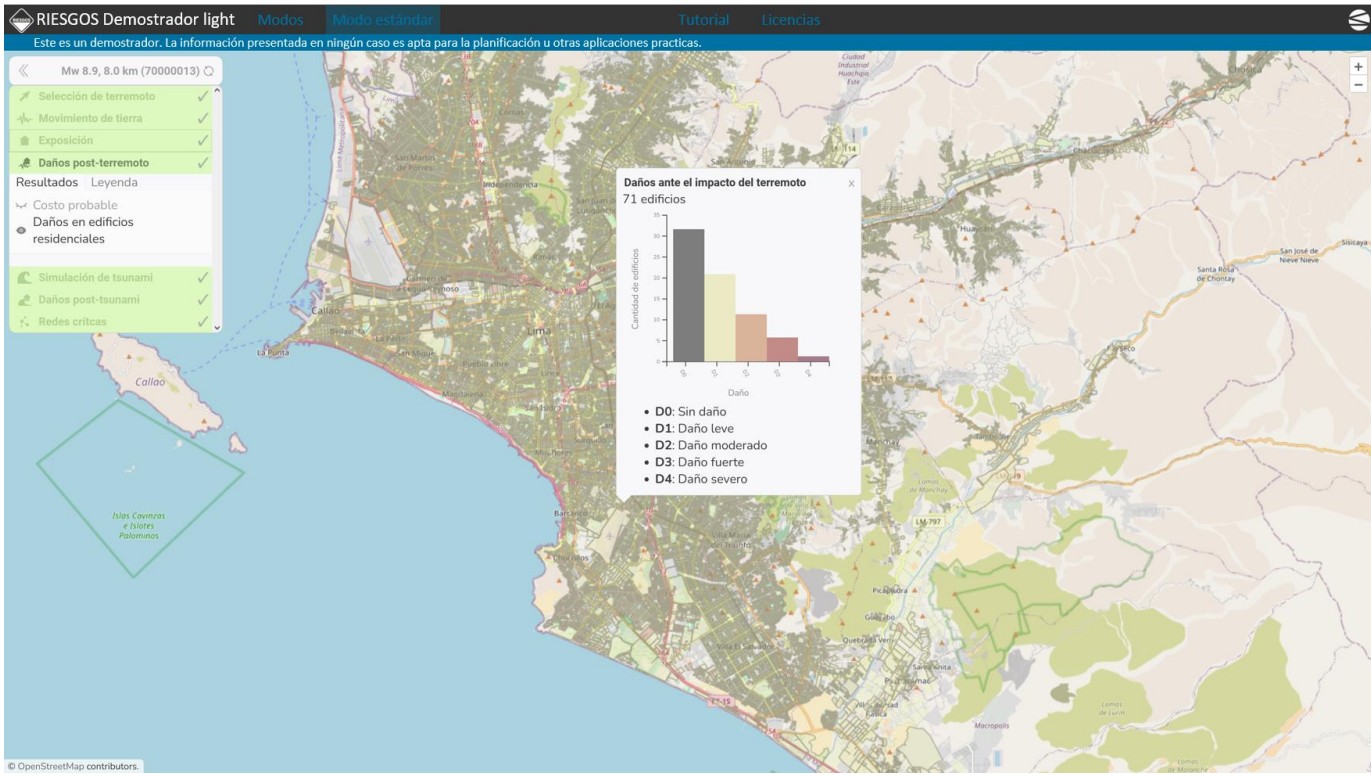

**Figure 7.** The Graphical User Interface (GUI) of the demonstrator light comes with predefined parameters and a simplified visualization of results. The underlying web services are identical to the ones used in the expert-mode of the demonstrator. The GUI of the demonstrator light is available in Spanish only. The visualized example shows the damage states of the residential buildings for a Mw 8.9 earthquake scenario. Map data © OpenStreetMap contributors and available from https://www.openstreetmap.org.


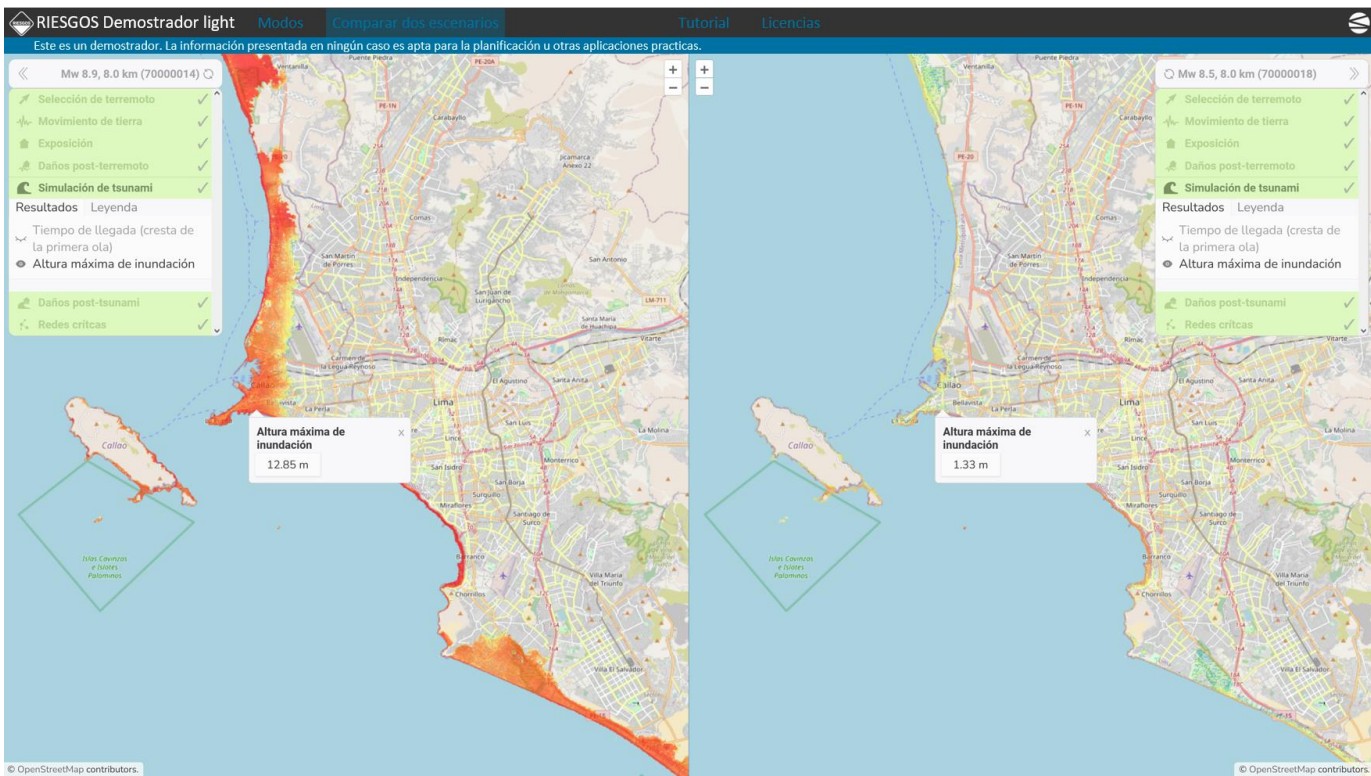

**Figure 8.** Within the demonstrator light a user can compare two different scenarios to explore the different impacts. On the left window a scenario with an earthquake of Mw 8.9 is displayed whereas on the right side a scenario with a Mw 8.5 is shown. The selected result shows the estimated tsunami inundation for each scenario. Map data © OpenStreetMap contributors and available from https://www.openstreetmap.org.


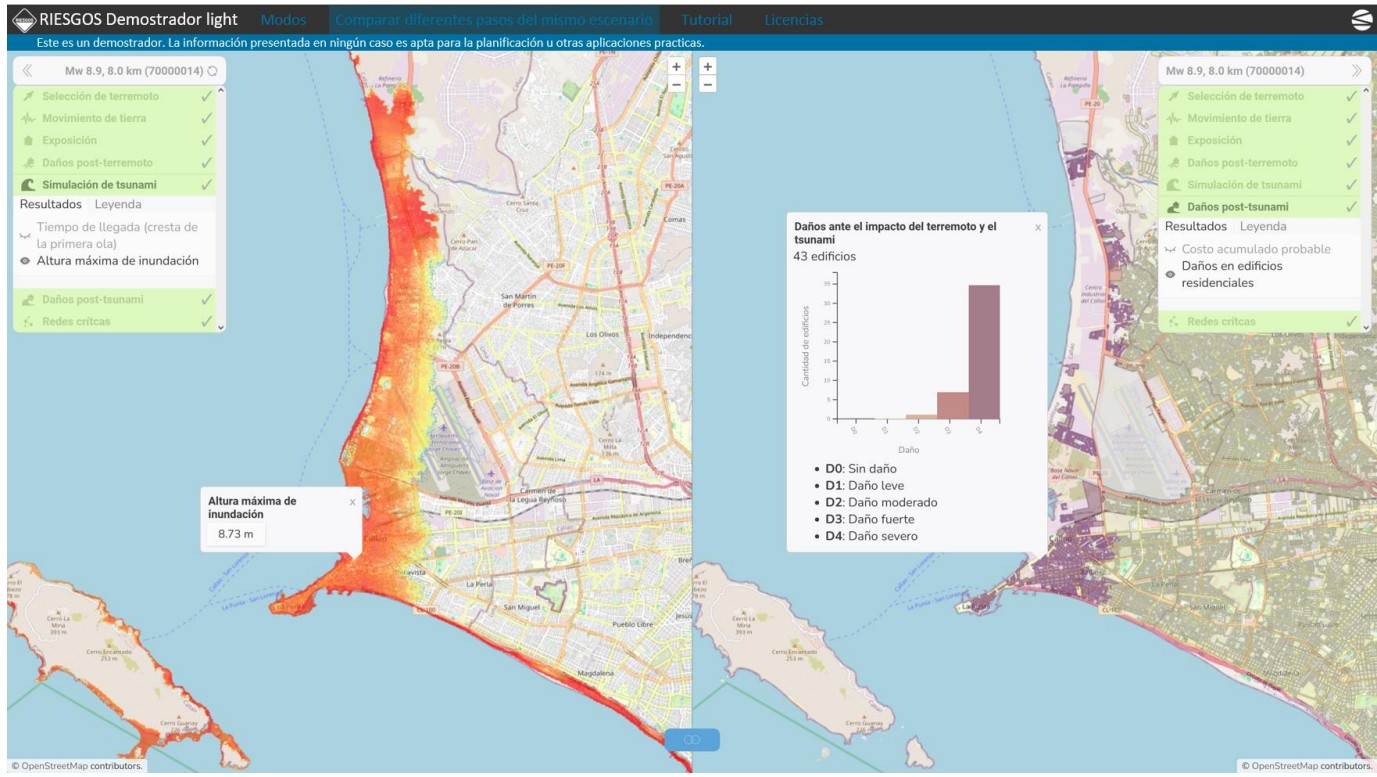

**Figure 9.** The demonstrator light offers a mode which allows a user to explore different steps within a scenario. In this example, the estimated tsunami inundation for a Mw 8.9 earthquake is shown on the left window side. On the right, the accumulated damage (in terms of repairing costs of the corresponding building class in US Dollars) after the occurrence of the selected earthquake and tsunami is visualised. Map data © OpenStreetMap contributors and available from https://www.openstreetmap.org.

The web services and the graphical user interface are published online so that the preconditions for further development into an operational system are given. Details on the code availability on GitHub - a platform for managing, versioning and sharing source code - are provided in the respective section of the paper (see Code and data availability).

### 3.4 Findings from the user perspective

Feedback from the user perspective was obtained throughout the four iterations, i.e. development stages (Fig. 2) and was mainly facilitated through joint workshops and practical hands-on sessions involving all five stakeholder groups (Sect. 2.2.3). During the overall process we respected the ethical principles and guidelines for research involving human subjects (European Commission, 2021). Informed consent was achieved by providing details about the purpose on the research and the roles of the different actors involved. Involved stakeholders were further informed how the information will be used. The participation was voluntary. Above all, we respected the confidentiality as all questionnaires were anonymous and did not allow individuals to be identified. Neither minors nor people with limited capacity were involved in our project.

The number and diversity of participants in user feedback was high throughout the process, which spanned several years. Although we tried to target the questionnaires to the same participants, there were fluctuations due to job changes and

responsibilities of the respective stakeholders. Furthermore, the responses to the questionnaires depend very much on the professional background and daily work tasks of the respondents. The following results reflect the obtained feedback from the respective participating stakeholder group representatives, but do not meet any requirements for statistical representativeness. The first step in the iterative development process was to define a *multi-risk story* describing the different elements to be analysed in the multi-risk situation (Fig. 2, iteration 1). During the joint discussion with the various stakeholders involved, a compromise had to be found between the requirements of practical DRM and planning processes on the one hand, and the technical possibilities of modelling certain processes on the other. In the end, a set of elements was agreed upon that was judged to be the most realistic and at the same time feasible in terms of data, technology and science. This discussion took place in a workshop held in Lima on 20 April 2018.

Building on this, the stakeholders were involved in the three development iterations of the tool which included a first trial version V0.1 (*mock-up*) that was not yet functional (November 2018), followed by two functional *versions* V1.0 (November 2019) and V2.0 (November 2020) (Fig. 2, iterations 2 to 4). All three versions were presented and discussed during joint workshops. For the case study in Peru, the first two workshops took place in Lima (*mock-up* V0.1: 4 December 2018, with 11 participants; *version* V1.0: 19 November 2019, with 46 participants) while the third workshop was held online due to the global pandemic travel restrictions (*version* V2.0: 9 February 2021, with 37 participants). Next to open feedback rounds, feedback was additionally collected via questionnaires. During this process, we experienced that complementary practical hands-on session (V1.0: 19 November 2019, with 16 participants; V2.0: 10 February 2021, with 12 participants) with the tool increased significantly the quality of feedback as one can document the user experience in action. The direct interaction with the participants during these hands-on sessions supported to gain a better understanding and to avoid misinterpretations of articulated requirements or feedback. In addition, these hands-on sessions allowed many suggestions for improvement regarding the practical handling of the user interface as well as the visual and descriptive presentation of the results. These included comments on the visualization of damage grades as well as losses (in US Dollars), both on colours used and number of grades. Probably the most controversial response was regarding descriptions. Some participants wanted commonly used terms, while others voted to use international standards (e.g., the Tsunami Glossary (2019) published by the Intergovernmental Oceanographic Commission). There were also opposing opinions on technical functionalities. While some participants liked to use the option of moving layers independently between levels, others were irritated by this option. From this we have learned that it is essential to always explain well the advantages and disadvantages of the embedded functions.

In the following, we further highlight the main findings related to the understandability and relevance of the information generated, and the practical applicability of the tool. Comparing the appreciation of the understandability of the information visualized in the tool over the three development stages, a steady increase can be observed, which can be attributed to the improvements made through the systematic integration of the feedback. While in year 1 (V0.1) the information displayed was rated with 36 % as moderately understandable and highly by 64 %, for the adapted version in year 2 (V1.0) users of the tool responded that the understandability was moderate (32 %) to high (59 %) and 9 % even said it was very high. In year 3 (V2.0), most of all respondents (89 % of 37 participants) agreed that the clarity of the information displayed in the demonstrator was

highly (62 %), very highly (19 %) or even totally understandable (8 %) (Fig. 10a). When asked about the reasons for lack of clarity, the main concerns were related to missing explanations of the underlying processes, concepts and variables, but also data quality seemed to play a role. The feedback was addressed by setting up a wiki to provide further details on the information presented. It also became clear that the majority of potential users have little or no experience of using more complex risk analysis tools and interpreting scientific map products. This led to the proposal to introduce two user modes, where less experienced users are guided through the analysis process using pre-set default parameters ('*basic user mode*'), while more experienced users can freely use all configuration options ('*advanced user mode*').

The increase in understandability seems to goes along with a more positive assessment of the relevance attributed to the information. In year 1 (V0.1), 18 % of the 11 participants replied that the shown information is moderately relevant whereas the majority of 73 % rated the relevance as high, and even 9 % as very high relevant. In year 2 (V1.0), already 35 % of the 46 participants said that the relevance of the information was very high, while in year 3 (V2.0) more than half (55 % of 37 participants) rated it as very highly and 31 % as even totally relevant (Fig. 10b). Stakeholders also suggested greater consideration of critical infrastructure such as gas networks, ports, bridges, water supply network, healthcare facilities and communication networks. Of particular interest were the range of possible power network system failures, the assessment of the recovery time of partial or full system functions and the minimum supply to the population.

When working with scenarios, models and data, the topic of uncertainty cannot be neglected. With version V1.0 in year 2, we asked participants if they thought it was important to visualise the uncertainty of the results shown in the demonstrator. While the researchers are aware that the data and model results included in the system are subject to epistemic and aleatory uncertainties, the results of the feedback process left the impression that the issue of uncertainty receives little attention in the practice of users involved in planning or disaster risk management. In year 2, out of all responses only 7 % of participants replied that is less important to visualize these uncertainties, 4 % rated it as moderately important, 40 % as highly and 31 % as very highly important. 18 % agreed that it is totally important to visualise uncertainties. The discussion on uncertainties during the joint workshops in year 3 seems to have increased the awareness on the topic, as 29 % of stakeholders confirmed that is totally important to visualise uncertainties, in addition with 34 % rating it as very highly important. Another 20 % rated it as moderately important and only 17 % as less important (Fig. 10c).

Regarding possible practical applicability, the question was asked how likely it is that potential users would use the tool for their practical work. For the V0.1 version presented in year 1, 18 % of the participants rated the possibility of using such a tool as moderate, 64 % as high and 18 % as totally. In year 2 (version V1.0) potential users responded that it is very less (4 %) and less (4 %) likely of using the tool in their practical work. The majority rated the likelihood of using the tool as moderate at 21 % and high with 59 %. While 8 % considered this to be very high, 4 % answered that they totally would use such a tool. The practical applicability of version V2.0 in year 3 was rated as follows. 8 % of participants said they would be moderately likely using the tool, while 39 % said they would be highly likely and 39 % very high likely would use it. Finally, 14 % of said that they totally likely would use the tool if it was available. Although there was a slight decrease in the proportion who are totally likely to use

the tool (year 1: 18%, year 3: 14%), we believe it is fair to say that the overall percentage has increased, as 39% were very likely to use the tool in year 3, while this answer was not given at all in year 1 (Fig. 10d).

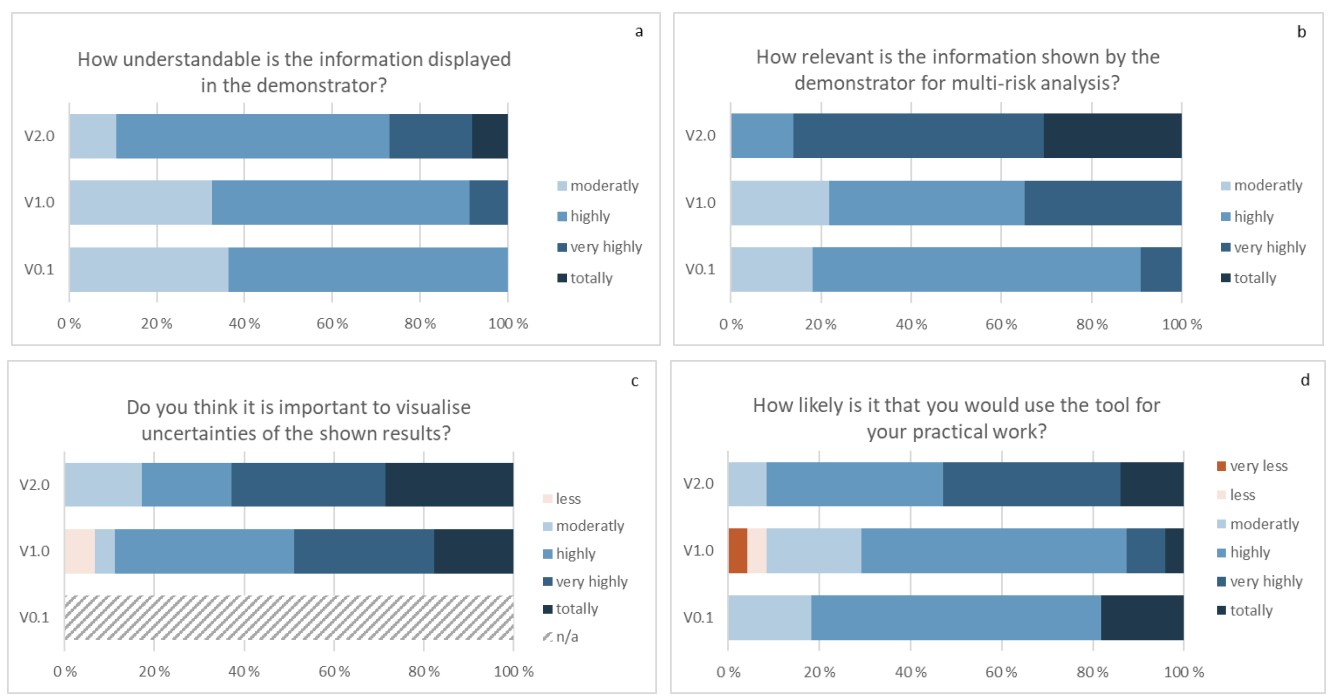

 **Figure 10.** Feedback from user perspective obtained during the three development stages (mock-up V0.1, versions V1.0 und V2.0) of the demonstrator for a multi-risk information system for years 1 (V0.1), 2 (V1.0) and 3 (V2.0). The diagrams represent four selected questions (out of a total of 45) on the information content (Fig. 10a-c) and applicability of the tool (Fig. 10d) asked to stakeholders in Lima (V0.1 was evaluated by 11 participants; V1.0 by 46 participants; V2.0 by 37 participants).

As part of the user feedback process, we discussed with the workshop participants potential fields of application. Most respondents

 saw great potential in the fields of disaster risk management, spatial planning and risk communication (Fig. 11).

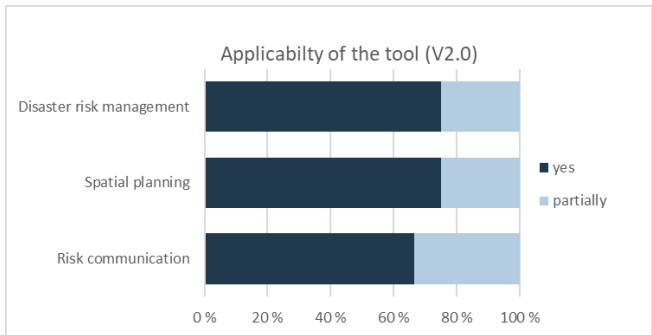

**Figure 11.** Feedback from user perspective on the potential of practical applicability obtained in a hands-on workshop with stakeholders and potential users (12 participants) in Lima during the development process of the demonstrator for a multi-risk information system in year 3 (V2.0).

 During specific user workshops, we further assessed the usefulness of the tool for local disaster risk management and spatial planning in the study area. It was found that the multi-scenario approach provided by the demonstrator has limited relevance to

current disaster risk management and spatial planning, as these processes in Peru need to be based on a fixed reference scenario with clear specifications of data and methods provided by national authorities and are already supported by existing GIS tools. However, it has also been recognised as an interactive tool for gaining a better understanding of complex risk situations. It can therefore be used as a complementary tool to existing information systems. In addition to the three topics described above, the areas of policy-making (e.g., investment planning) and disaster risk response have been identified as further fields of application. Even though the tool is having its main application field in the disaster risk reduction, users in Lima also expressed the potential to use the tool for the initial assessment of the situation in the aftermath of a disaster from complex multi-risk situations and cascading effects. In this case the architecture of the tool would have to be adapted to the requirements in the response phase after a disaster. Since it is reasonable to assume that communications and internet connections could be interrupted during a disaster, the tool would need to operate locally without depending on an internet connection. For applications in the prevention and preparedness phase, it is desirable to have a decentralized architecture like the one in the demonstrator, allowing the connection of servers that store various data that can be updated regularly.

## 4 Discussion and conclusions

In this paper, we presented one (of many possible) approach(es) to multi-risk analysis that can make a practical contribution to the implementation of global risk reduction goals. With the presented information system for multi-risk assessment, we prove that such a system can reconcile scientific research with its corresponding data and models with user requirements for describing different scenarios of a complex multi-risk situation and to support decision-making. In the following, we like to discuss various aspects, including limitations.

    i.   *Relevance and acceptance*: Users have recognized the relevance of the topic right from the beginning and have expressed a high demand. This is certainly also because the topic of multi-risk is becoming increasingly relevant in practice and that there are still few practical options available for dealing with these new challenges. Various users wanted to use the tool directly in its first version as they recognized great potential in communicating scientific results to decision-makers. With this, we emphasize to follow the recommendations on supporting the development of user-friendly systems and services as articulated in the Sendai Framework (UNISDR, 2015a, p. 14-16).

    ii.   *Story-based scenario approach*: The story-based approach enables users to simulate various scenarios in one defined multi-risk situation (*story*) and to compare the results accordingly. The multi-scenario approach may be interesting for the development of strategies to strengthen or develop resilience strategies or to check the robustness of planned or already implemented measures (e.g., with reference scenarios) under different hazard scenarios ('*stress test*') or to changing conditions. However, the multi-scenario approach has limitations in some applications, especially when there are mandatory requirements to use a predefined reference scenario for practical planning processes. This is especially the case for local DRM planning.

iii. *Complexity*: Multi-risk situations can become very complex. Obviously, models and scenarios are always incomplete as they only can approximate the complexity of real situations (see for example the risk framework introduced by Taubenböck et al. (2008) with the manifold and still incomplete indicators for operationalization). The analytical process of the interactions of elements in scenarios is furthermore confined to selected processes. For demonstration purposes, we limited ourselves to the physical elements of vulnerability (buildings, critical infrastructure). Table 3 lists the numerous and partly high-resolution input data for the relatively simple earthquake-tsunami story and this is already a minimal data set to model and approximate the situation realistically with considerable uncertainties. More high-resolution data sets can improve the modelling and reduce the uncertainties. An important factor in the evaluation of data inputs is certainly the available IT resources for processing and modelling. Economic, environmental, political, social and societal aspects of vulnerability were left out, which, however, is not implying any judgement on their relevance for assessing and understanding vulnerability. This, of course, resulted in a considerable limited representation of what would happen in a real disaster situation. This limitation was openly addressed and made transparent in the feedback process. Despite this limitation, the stakeholders still rated the potential of the tool as high, considering the results of the physical vulnerability assessment and it has stimulated them to develop new strategies for capacity building and resilience measures. Ultimately, the tool is designed in such a way that interested parties can integrate social factors of vulnerability at any time during adaptation and further development of the multi-risk story. To allow for this, we made the framework and its source code publicly available.

iv. *Uncertainties*: We presented an approach which is based on multi-risk stories and scenarios, which implies a variety of uncertainties throughout the analytical process. A first dimension of uncertainty derives from the selection of the elements to consider in the *stories* and the description of interactions among them; followed by further uncertainties, e.g., due to lack of knowledge (epistemic uncertainties) or due to the inherent variation associated with the environment under consideration (aleatory uncertainty) (e.g., Oberkampf et al., 2002). Moreover, uncertainties are interlinked along a multi-risk chain and can not only add up but ultimately reinforce each other. We call for the inclusion of mechanisms to visualize the uncertainties of the risk assessment, preferably in graphical form and without the use of technical jargon, to allow appropriate communication with users about the respective level of uncertainty.

v. *Practical applicability*: The experience throughout the development process showed that the modelling of complex multi-risk situations is challenging and subject to limitations in representing what can happen (see point iii) as well as to significant uncertainties (see point iv). This leads to the conclusion that the results generated by the tool have rather an orienting character and that the main purpose of the tool is mainly being an exploration tool to better understand complex risk situations. It should therefore be understood as an instrument that complements already existing information systems and planning tools in the different fields of applications.

vi. *Decentralized architecture*: The selected decentralized architecture certainly has advantages ranging from (1) updated information, as the data and models in the specialized institutions are usually refreshed on a regular basis, (2) modularity, flexibility, scalability of multi-risk situations to (3) easier data exchange between institutions as data remain at their

point of origin / host. However, despite the use of international standards such as the geospatial WPS defined by the OGC, the integration of new web services into the tool requires adaptations of the underlying orchestration structure. Thus, the (re-)combination of web services to form a new multi-risk chain calls for in-depth knowledge. We do see the potential of the approach for other multi-risk stories, e.g., landslides after an earthquake, failure of drinking water infrastructure, evacuation of the affected population, but recommend to analyse in advance the transfer efforts.

vii. *Transferability and scalability:* The approach was presented for an earthquake-tsunami multi-risk story for the Lima Metropolitan area in Peru. Regarding the transferability to another region, we report that we have applied the approach in two further case studies. In the coastal area of Greater Valparaíso, Chile (cf. Gómez Zapata et al., 2021d; Gómez Zapata et al., 2022a), the multi-risk story was like Lima Metropolitan area focusing on the earthquake and tsunami cascade affecting housing and the critical infrastructure power grid. In Ecuador the approach has also been adapted for compound hazards (two hazard events happening in parallel) analysing the impacts of ash fall and lahar around the volcano Cotopaxi in Ecuador (cf. Gómez Zapata et al., 2021a). Similar to the Peru study case, a feedback process with four iterations involving comparable stakeholder groups was implemented. The use of similar questionnaires in the three country studies helped to compare results from the feedback. The results showed that comments on the main features of the tool were consistent across the study cases. Of course, there were specific points to consider in the individual countries, for example, regarding the colours used to display the results or the damage classes, as participating countries use different damage categories and colour codes. To ensure the transferability and scalability of our approach, the tool was designed from the outset to be adaptable to all types of complex multi-risk stories (see point iii on 'Complexity') at different scales and to accommodate national or local preferences regarding damage categories or colour codes.

viii. *Operational system*: Users showed strong interest in the presented tool. However, the transfer from a demonstrator system to an operational service requires further efforts along with a clear commitment and solid institutional embedding. As we have chosen a decentralized service-oriented architecture (SOA) with distributed web services for the demonstrator also the individual web services (see Tab. 3) can be integrated in already existing information systems. The interaction with the web services is achieved using Web Processing Service (WPS) interface standard guidelines published by the Open Geospatial Consortium (OGC; WPS, 2018) and are openly documented. Interoperability is achieved by a thorough harmonization of input and output formats and the use of on-the-fly converters. Dedicated WPS create simulations of intensity maps for specific hazards on the fly (e.g. for earthquake ground motion simulation) or by querying a list of pre-simulated events (e.g. for tsunami inundation maps). We recommend a partnership between research institutions, public authorities and service providers whereas one key authority should act as the hosting institution to integrate the tool or individual web services. The integration process itself requires profound knowledge both, in the models and IT programming (both backend services and frontend development), which needs the interaction of different specialized institutions and professional support from IT experts.

ix. *Data availability and data exchange:* As experience shows, data is often available, but data exchange remains challenging. The use of web services is a promising option for the exchange of information between institutions. Data

do not need to be stored at a centralized place (and with this gets outdated), but can be updated regularly by the host. An open data policy (FAIR principles) eases this process, but calls for inter-institutional agreements and rules of procedure. Where data availability is still critical (e.g., detailed exposure information) the scientific community can support the creation of enhanced datasets. Our experience shows that users are often satisfied with rough estimates of *What-if scenarios*. In strongly application-oriented research, it is vital to find a balance between maximum accuracy and practical applicability.

x.   *Co-creation with users*: Our experience of collaboration between researchers, software developers and different potential users confirms that users' satisfaction with their involvement and the resulting system are interdependent, with the degree of user satisfaction evolving at different stages of the development process, as postulated by Bano (2017). It also confirms that involving users as a primary source of information is an effective means of capturing system requirements (Kujala, 2003). However, collaboration requires a strong engagement from all sides. We agree that the role of users in such a process must be carefully considered (Kujala, 2003) and therefore applied a moderated process which allows that user demands can be communicated to the researchers and developers without outweighing the scientific relevance. At the same time, the involved user must be aware and able to cope with trade-offs and compromises, as not all requirements may be addressed or they might not be able to benefit directly from the tool while it is still under development or in a demonstrator stage. To avoid false expectations and misunderstandings, we emphasize that transparency and clear statements are crucial throughout the user involvement process. Additionally, users (often) do not have the scientific expertise to adequately describe the individual processes in a multi-risk chain. Since the approach is based on the description of a multi-risk story, this story must always be defined in a joint dialog between users, researchers and software developers. In our experience, much of the mutual learning took place during face-to-face interaction rather than digitally. With this in mind, the design of such collaboration must be critically balanced against the quite justified demand for more cost-efficient methods of capturing implicit user needs and requirements in real product development contexts (Kujala 2003).

Further lessons learnt and recommendations for action are given in Schoepfer et al. (2024).

In conclusion, we have demonstrated that the tool can calculate and visualize the cumulative effects of successive hazard events. Despite some limitations, in particular regarding already standardized planning processes and the exploratory nature of the tool, users see great potential for different fields of application and a high expectation was expressed, especially from the user side in the local pilot area, that the developed tool would be available and applicable locally. Based on these findings, it appears reasonable that the research community continues working with users on the ground. Further research in the field of multi-risk assessment is certainly needed, among others, to improve the physical vulnerability assessment of various hazards. The standardization of damage scales into a transversal one across hazards will be an important aspect for the scientific community to address. Complementarily, the derivation of state-dependent analytical fragility also deserves more research attention to be optimised in the future through more refined approaches. We also recommend to further work to integrate the social vulnerability. Here, it could be of importance to investigate whether and how the social vulnerability of certain demographic groups differs in

terms of their response to future crises. Our findings also support the call to science to contribute to an evidence-based policy. After all, the future will tell us how much such a tool can help in planning for catastrophic events and what, in the end, can technologically not be forecasted but is simply fate.

## Code and data availability

Repositories of the projects RIESGOS and RIESGOS 2.0 are provided as open source code on GitHub at
725 https://github.com/riesgos and https://github.com/gfzriesgos (last access: 20 May 2024). DOI-referenced data from the RIESGOS and RIESGOS 2.0 project are hosted at GFZ German Research Centre for Geosciences at: https://dataservices.gfz-potsdam.de/portal/?q=riesgos* (last access: 20 May 2024) and at the Technical University Munich (TUM) at https://mediatum.ub.tum.de/1735865 (last access: 20 May 2024).

## Supplement

The questionnaires are provided as a supplement to this manuscript.

## Author contributions

The paper was conceptualized by ES, JL, TR, HS, GS and HT. ES coordinated the effort and prepared the original draft of the manuscript with major contributions from JL, HS, TR, and HT. JCGZ, HRV, SH and CG provided details on the study, data analysis and modelling. Technical details on software development were provided by ML and NB. Details on user involvement
and feedback analysis was performed by HS with major contributions of CDL and ES. All authors contributed on previous versions of the manuscript and approved the final version.

## Competing interests

ES, JL, TR and HT are guest editors of the journal. The other authors declare that they have no conflict of interest. The funders had no role in project design, data collection and analysis, decision to publish, or preparation of the manuscript.

**Acknowledgements**

The authors would like to thank the BMBF for its CLIENT II initiative and corresponding funding. We wish to address our gratitude to Project Management Jülich (PtJ) for the excellent supervision. The efforts of the RIESGOS and RIESGOS 2.0 projects partners (DLR, GFZ, AWI, TUM, 52°North, geomer, SLU, DIALOGIK, EOMAP, plan+risk consult) are highly appreciated and gratefully acknowledged. We thank the associated project partners (AHK Chile, AHK Ecuador, AHK Peru, GIZ, UNOOSA/UN-

SPIDER, UNESCO, Munich RE). We gratefully acknowledge the collaboration and support of various institutions in South America, in Chile CIGIDEN, SENAPRED, SENAPRED Región de Valparaíso, SHOA, and Itrend, in Ecuador SENESCYT, SNGRE, IG-EPN, ESPE, and GAD Cotopaxi, and in Peru CENEPRED, CEPLAN, INDECI, PREDES, and MVCS. Last not least, we want to express our sincere appreciation and recognition to all the individuals for their contribution to the research projects RIESGOS and RIESGOS 2.0.

**Financial support**

This research was funded by the German Federal Ministry of Education and Research (BMBF) as part of the initiative 'CLIENT II - International partnerships for sustainable innovations' within the framework programme 'Research for Sustainable Development (FONA)' under grant number 03G0876A-J (project RIESGOS) and grant number 03G0905A-H (project RIESGOS 2.0).

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
