# Peer review of "Between global risk reduction goals, scientific-technical capabilities and local realities: a modular approach for user-centric multi-risk assessment"

_Natural Hazards and Earth System Sciences, 2023_

## Referee Comment (RC1)

**Between global risk reduction goals, scientific-technical capabilities and
local realities: a novel modular approach for multi-risk assessment**

This manuscript presents a framework for implementing multi-risk assessments in practice. The approach is demonstrated for Lima, Peru, in the case of a tsunami following an earthquake.
The authors also provide a helpful overview of the motivation for shifting practical disaster risk reduction to a multi-risk framing, in the introductory section. The content is timely and would be of interest to readers of the journal. However, there are several comments provided below that I think the authors should address before the manuscript can be deemed publishable in my opinion.

**Main comments:**
1. Novelty: The authors claim to present a new conceptual approach to multi-risk assessment. But (despite what is implied by line 156), all the tools used for conducting the fundamental risk calculations have been developed in previous studies. Furthermore, the end-to-end calculations conducted in this study do not represent an advancement over the numerous frameworks for multi-risk assessment that have already been proposed in the literature (and that are referred to in the manuscript itself, for example around line 90). I think the authors should frame the novelty of the approach more accurately in terms of its practical relevance.
2. Scope: Related to the previous comment, the approach has only been demonstrated in the context of a very narrow definition of multi-risk assessment (i.e., one set of interacting hazards where one hazard triggers the other and for which there are well established models that capture the underlying interactions at the hazard and impact levels). Section 2.2.3 seems to suggest that, despite the decentralised architecture of the system, its design is inherently dependent on the multi-risk story selected. Point vi of the conclusions seems to confirm my doubts about the generalisability of this approach to other contexts. (Furthermore, is the approach limited to hazards that interact through triggering?) I think the authors need to provide a more honest description of the limited scope of this study near the start of the manuscript. This is merely a first (straightforward and somewhat simplified) demonstration of a practical approach for facilitating user-centred multi-risk assessment. Furthermore, I believe the manuscript could benefit from a discussion about the challenges associated with expanding or enhancing this type of system for more complex contexts, e.g., involving more than two hazards and/or where there is less well-established means of capturing their interactions.
3. User input: The user-oriented design of the approach is a welcome feature. However, despite its numerous advantages, there are some "dangers" associated with allowing user input in this type of system. For instance, stakeholders may not be sufficiently educated to appropriate hazard stories, particularly in the context of climate change. A comment on the potential downsides or caveats associated with user involvement should be added to the manuscript, in my opinion.
4. Case Study: This could benefit from a few more details.
    a. It seems that the multi-risk story was pre-defined in the case study (i.e., taken from INDECI, 2017), which is not compatible with the user-centered workflow presented in Section 2.2.3.
    b. How is the size of the tsunami related to the magnitude of the earthquake selected and how is the uncertainty in this size accounted for?
    c. What are the outputs (risk metrics) shown? Are all metrics disaggregated per hazard event? Do they account for cascading impacts (as described in the last two paragraphs of Section 2.2.2)? Was there consultation with the end users on the types of risk metrics to be shown in the system? The conclusion mentions that the platform can be used to compare the results of different stories, but the ability to do this (i.e., show multiple sets of results side by side) is not made clear in the case study description.
    d. I think the manuscript could benefit from more figures of the system, particularly the GUI.

e.  The spatial extent of the case study needs to be described, particularly in the context of cascading impacts (see comment 4d).

5.  Introduction: Despite its strengths, I think the introduction section is a bit disorganised. I think that some of Section 1.3 should be moved forward to Section 1.1, such that all content that provides a general motivation for risk management is contained within one section.

6.  Questionnaire and user feedback: The link between the results shown in Section 3.4 and the questions in the questionnaire needs to be clearer (I cannot find any of the questions mentioned in Figure 7 in the questionnaire questions provided in the supplementary material). The supplementary material should provide all questions, and the results for all questions should be provided in the main text (at least in summary form).

**Minor comments:**

1.  Line 180: end users are mentioned here as a stakeholder category but it is not yet clear why they would be considered a separate category in themselves - any of the other stakeholder categories listed here could also be a potential end user of this type of system. I see that the end users are described in more detail in line 281; this explanation should be moved forward to line 180 for clarity. However, the situation is further confused in the conclusions section (point viii) and Figure 2, where stakeholder groups are described as "user groups".

2.  Figure 5: A reader may look at this figure and question why an EQ catalogue is an input if we are dealing with a specific earthquake scenario. I think it should be more clearly described in the flowchart that the EQ catalogue is used to choose an earthquake scenario for which the ground motion is simulated (the earthquake scenario itself is currently missing from the diagram).

3.  Line 475: I do not believe that a value of 55% could be described as an "overwhelming majority"

4.  Line 485: it seems that the practical usability of the tool actually decreases over time – e.g., 14% said they are totally likely to use the tool in year 3 versus 18% in year 1. Furthermore, Figure 7d does match with the description of these results provided in the text; it is mentioned that 64% rated the possibility of using the tool as highly likely in year 1, but there is no highly likely colour marked on the bottom bar of fig 7d.

---

## Referee Comment (RC2)

This is a review of "**Between global risk reduction goals, scientific-technical capabilities and local realities: a novel modular approach for multi-risk assessment**" NHESS-2023-142 by Schoepfer et al.

I thank the authors for their very interesting manuscript on framing multi-risk assessments in the context of a case study in Peru.

I believe that this paper will be an excellent addition to the growing literature on multi-risk assessment, as long as it does not overstate what has been proposed and done, recognises the inherent biases and limitations involved with any such analysis, and considers a bit more strongly the praciticioner stakeholder who might use the methodology proposed (or parts of it).

Below are a series of comments, in no particular order of importance. Although some are slightly critical (the nature of doing a review), most are aimed at making the manuscript more useable and useful by practitioner stakeholders and others who might want to take your learnings and apply them to another region.

1. Title.
    a. The paper is much more about your **case study in Peru, so I would expect that to be in the title**.
    b. You use the word '**novel**'. Is this really novel? All the elements have been done previously. I believe the approach and paper are well worth while, just be careful about overstating the originality of what you are doing.
2. Abstract. **This is a bit high level and more a motivation rather than an actual (with metrics such as 'how many' and 'of what') summary of the paper.** I suggest you rethink a bit the abstract, and consider more how it is really a summary of the paper.
3. Introduction.
    a. The introduction does a nice job of bringing in some of the literature, but **I believe there are other major papers** out there that have put into context multi-hazards, multi-risk and multi-impact in the context of natural hazards. Please do a relatively rapid review to ensure you have captured the majority of papers out there that have put into context multi-hazard/risk/impact.
    b. **I did not find it easy to read the introduction due to all the definitions and quotes**. Perhaps consider for the definitions using tables or bullet points so that it is not huge chunks of text with lots and lots of quotes. I've seen half a dozen 'reviews' of the past literature on multi-hazards and multi-risk, and the most useful ones I have seen (from a practical perspective) are those that have tables, figures with timelines, ideas broken out into bullet points, etc. I understand that you do not want to do a complete review of the literature—that is fine, but perhaps one or two table with your key quotes to reduce the text? For example, much of Section 1.3 could be supplemented by a table. Many of the quotes in Section 1.1 could be in a table and then referred to. The studies given in 1.2 would be ideally put in a table, with a few headers to pull out salient parts of the studies, and then discussed in the text.
4. Conceptual Approach.
    a. This is broadly fine, within the limitations of what is presented and has a couple of nice summary figures, but I give a few comments below
    b. **General: Use of the word and approach to vulnerability**. A key part of risk, as you acknowledge, is vulnerability. The word vulnerability comes up 14 times in the manuscript (many of these are part of direct quotes), which is appropriate, but at no place do you define vulnerability (although do mention once physical vs. social vulnerability). For me, a key part of multi-risk (vs. multi-hazard) analyses, is the incorporation of both physical and social vulnerability. I would like to see a more solid defining of vulnerability either in the intro or conceptual approach, along with strengths and limitations of including physical/social vulnerability into multi-risk assessment in terms of data, equations, etc. either when vulnerability is first mentioned or in the discussion. It was not until I got to line 536 that I felt you acknowledged that physical vulnerability only was included and not social vulnerability, and this needs to be acknowledged much earlier.

c. Again, like the intro, I found **there was a lot of text to go through in the conceptual approach**, to get to the practical 'how is this being done' parts. Might you break some of the text into bullet points or numbers, to make it easier to read? I'm thinking of the practitioner (who you have aimed part of this paper at) who wants to know what to do, how to do it, and limitations.

d. Riesgos and Riesgos 2 are mentioned on line 148 (the only place in the text) and then on the data and code availability section. This code seems essential for a practitioner to operationalize the approach suggested here in a pracitical way (and which you do a test case study with peru). **I would suggest you have 1-2 paragraphs outlining more about Riesgos Code Availability and Use (or refer the reader explicitly to the places they can read about how to use it) with text both in this section and then again in the next section (Peru Case Study)** so that they can better understand the theory going into practice, or more importantly, how would they begin to implement the learnings from this paper if they were interested.

5. Peru Case study
   a. Some **very nice figures and flowcharts**, but please **reevaluate the white text in Figure 5** (not easy to read on my PDF), and where possible **enlarge font size on figures**.
   b. Broadly I was fine on the approach taken. It does get at a number of interesting aspects of multi-risk (although not social vulnerability).
   c. Please state somewhere the **ethical procedures** you went through before working with the human participants.
   d. Be careful of **typos.** Lines 374-375. Earthquakes appears twice
   e. Use of **Tables**: This section might benefit by **an additional table summarizing the data used, their sources, key parameters, and any comments such as regarding uncertainty**.
   f. I found the user groups were interesting, but I would like to see
      i. **a much better definition of the user groups** given in Figure 2 (which I assume were then used in Peru) and
      ii. some **idea of the user group numbers involved and where they were located**—in other words, why were they representative.
      iii. I also am **not a fan of the word 'end users'** as everyone in the research community, NGOs, etc., are end users. This is why (see above) I'd like a much better defining of who was actually involved. You have a couple lines on this in in 280-282, but then when we get to Section 3 you do not refer back to this discussion, and it should be more in-depth.
      iv. In all of the reporting of the results you state things like "18% of the the users"—I assume this means that you have now put all the users together into one big group. **Remind us in a few strategic place 'how many'. So 18% of the ### users.**

6. Discussion and conclusions.
   a. I found the basic ideas in the discussion and conclusions good, but felt it was rather short and did not bring us back to the overall literature of what others have done. Please relate many of **your key points back to the existing literature**.
   b. The approach relies heavily on the **availability of detailed data** (e.g., about hazards, vulnerabilities, and exposures). In regions where such data may be lacking or outdated, the application of the methodology could be challenging. Might you be able to acknowledge **more these limitations and suggest potential solutions or workarounds**?
   c. The paper focuses on a specific case study area, and while this demonstrates the practical application of the methodology, there is **limited discussion on its scalability and adaptability to other regions** with different risk profiles and socio-economic contexts.
   d. The discussion could be strengthened by a more **explicit identification of gaps in the current approach**. This would not only highlight areas for improvement but also encourage further research in the field of multi-risk assessment.
   e. **Actionable Recommendations**: Make the recommendations actionable by providing clear, specific steps that can be taken by researchers, practitioners, or policymakers.

For instance, instead of broadly stating the need for further research, specify the types of studies or methodologies that could address existing gaps.

    f. **Highlighting Implications for Policy and Practice**: Explicitly articulate the implications of your findings for disaster risk management policy and practice. This could include suggesting changes to existing frameworks or identifying new areas for policy development.

7. Overall.

    a. While the innovative methodology is a strength, its complexity could be a barrier to its widespread adoption. **The text could benefit from a more simplified explanation in places or additional step-by-step guides that could make the approach more accessible to practitioners who may not have a strong technical background**.

    b. Overall there is a high level of writing, **but tending towards VERY long paragraphs, which often could be broken up into two, or better use of bullet points**.

    c. This will most likely go through copy editing, but there are places where text could be improved. Long sentences are often used where they could be broken up into two or shortned. Some examples (there are many) include.

- **Original Lines 19-21:** "The complex relationships between multiple and consecutive natural hazards exposed population and built environment result in a variety of cascading effects which if are often not considered appropriately by decision makers can result to inadequate or even misleading risk management strategies."
- **Suggested Revision in two sentences:** "Complex interactions among multiple and consecutive natural hazards, the exposed population, and the built environment can lead to cascading effects. If not accurately considered, these can lead decision-makers to implement inadequate or misleading risk management strategies."
- **Original Lines 27-29:** "Based on recent scientific and technical capabilities we developed a tool through an iterative participative approach which has allowed users to explore various scenarios of multiple hazards cascading effects and their impacts."
- **Suggested Revision in two sentences:** "Leveraging the latest scientific and technical advancements, we developed a tool via a participatory iterative process. This tool enables users to explore various scenarios, including the cascading effects of multiple hazards and their impacts."
- **Original**: "In addition to immediate crisis management and rapid response during and after a disaster, disaster preparedness is becoming increasingly important."
- **Suggested Revision:** "Beyond immediate crisis management and rapid response, disaster preparedness is growing in importance."
- **Original**: "The shift from managing disasters to managing risk is articulated in the Sendai Framework for Disaster Risk Reduction 2015-2030, which was adopted at the Third UN World Conference in Sendai, Japan, on March 18, 2015."
- **Suggested Revision:** "The transition from disaster management to risk management is emphasized in the 2015-2030 Sendai Framework for Disaster Risk Reduction, adopted at the Third UN World Conference in Sendai, Japan on 18 March 2015."
- **Original**: "An increasing number of people worldwide are exposed to natural hazards, particularly in poorly planned urbanisations, where effective prevention and risk management can save lives and reduce all kinds of losses."
- **Suggested Revision**: "More people globally face natural hazards, especially in poorly planned urban areas where effective prevention and risk management could save lives and minimize losses."
- **Original**: "For instance, in the context of seismic hazard, information on the possible earthquakes that can hit a region in the future needs to be available. For that aim, existing earthquake catalogues are gathered."
- **Suggested Revision**: "For example, in seismic hazard assessment, future earthquake risks require access to existing earthquake catalogues."
- **Original**: "However, the design of information systems or tools that are capable of analytically exploring multi-hazard risk situations and, in particular, dynamically updating the damage on exposed elements due to various hazards with cascading effects remain challenging."
- **Suggested Revision**: "Designing information systems or tools to dynamically analyze multi-hazard risks and dynamically update exposed element damages from cascading hazard effects presents significant challenges."

---

## Author Comment (AC1)

**AUTHORS' RESPONSE TO THE REVIEWER#1 COMMENTS**

"Between global risk reduction goals, scientific-technical capabilities and local realities: a novel modular approach for multi-risk assessment" by Schoepfer et al.

**RC1: 'Comment on nhess-2023-142', Anonymous Referee #1**

This manuscript presents a framework for implementing multi-risk assessments in practice. The approach is demonstrated for Lima, Peru, in the case of a tsunami following an earthquake. The authors also provide a helpful overview of the motivation for shifting practical disaster risk reduction to a multi-risk framing, in the introductory section. The content is timely and would be of interest to readers of the journal. However, there are several comments provided below that I think the authors should address before the manuscript can be deemed publishable in my opinion.

We thank you very much for the recognition of the paper's relevance. We would like to express our gratitude for your valuable comments and suggestions for improvement. We are committed to enhance the quality of our manuscript based on your comments and aim to carefully consider all concerns and incorporate the suggestions in an improved version of the manuscript. Please find our comment-by-comment feedback in the following. The lines indicated correspond to the PrePrint: https://nhess.copernicus.org/preprints/nhess-2023-142/nhess-2023-142.pdf

**Main comments:**

1. Novelty: The authors claim to present a new conceptual approach to multi-risk assessment. But (despite what is implied by line 156), all the tools used for conducting the fundamental risk calculations have been developed in previous studies. Furthermore, the end-to-end calculations conducted in this study do not represent an advancement over the numerous frameworks for multi-risk assessment that have already been proposed in the literature (and that are referred to in the manuscript itself, for example around line 90). I think the authors should frame the novelty of the approach more accurately in terms of its practical relevance.

Thank you very much for this comment. We highly appreciate your feedback about the practical relevance.

We agree that individual results of the paper have already been published, in particular the ones entitled as "elements of risks" (see section 2.2.2; e.g., exposure and vulnerability modelling as published by co-author Gómez Zapata et al.). In this paper our focus was to present the overall conceptual approach (which has not been published in such details yet) and not on diving into details of research already published. With this, we feel that it is valid to name the approach as novel.

As you kindly pointed out, the approach is in particular relevant due to its practical relevance and the user-centred design. The tool was developed in close cooperation and consultations with users in an iterative form. In doing so, we paid attention to involve the users in crucial stages of the project, such as the story design. We integrate existing local knowledge in what the potential users are already used to work with (e.g., colour schemes, visualizations of results, etc.). Considering this local knowledge and additional feedback asked by questionnaires and observed during hands-on sessions, we constantly implemented improvements (e.g., side-by-side scenario comparison).

We will take care that in a revised version of the manuscript, the novelty of the approach is described accordingly, such as:

Line 146: *"Considering the aforementioned guidelines and strategies in the context of disaster risk reduction (DRR) and disaster risk management (DRM) as well as the outlined research needs, we*

*present a generic framework developed within the research projects RIESGOS and its successor RIESGOS 2.0 (Schoepfer et al., 2018). The projects focused on the development of innovative scientific methods for the assessment of multi-risk situations with the aim of designing an approach that meets the needs of users at the local level. In addition to the German team coming from various disciplines, the project collaborated with a variety of research institutions and public authorities in Chile, Peru and Ecuador. This collaboration, both with users and local actors, is one key aspect while framing the overall approach towards its practical applicability. The conceptualization of this overall approach is visualized in Fig. 1. We argue that the starting point of our conceptual approach is a context and stakeholder analysis (Sect. 2.1) to understand the organizational environment and underlying structures. Later, we present a framework to design a multi-risk information system (Sect. 2.2). We selected a story-based concept that allows the description of a specific multi-risk situation and its representation through multiple scenarios (Sect. 2.2.1). As input, the elements of risk (hazard, exposure, and vulnerability) and their impacts on critical infrastructure are  considered in terms of their potential implementation (Sect. 2.2.2). During these two steps, we involved users in the process from the beginning to ensure that the designed tool their requirements and needs (Sect. 2.2.3). For the demonstrator we chose a decentralized system architecture approach built on distributed web services, with a graphical user interface as the frontend (Sect. 2.2.4). It has to be noted that during the course of the projects individual results have been published and are cited accordingly. In this paper, we put particular focus on the user involvement and feedback showing the practical relevance of the overall approach and the designed tool. We are convinced that such a user-oriented approach for exploring, describing and quantifying different What-if scenarios can constitute a valuable and user-accepted tool for understanding complex multi-risk situations and to prepare for such situations."*

2. Scope: Related to the previous comment, the approach has only been demonstrated in the context of a very narrow definition of multi-risk assessment (i.e., one set of interacting hazards where one hazard triggers the other and for which there are well established models that capture the underlying interactions at the hazard and impact levels). Section 2.2.3 seems to suggest that, despite the decentralised architecture of the system, its design is inherently dependent on the multi-risk story selected. Point vi of the conclusions seems to confirm my doubts about the generalisability of this approach to other contexts. (Furthermore, is the approach limited to hazards that interact through triggering?) I think the authors need to provide a more honest description of the limited scope of this study near the start of the manuscript. This is merely a first (straightforward and somewhat simplified) demonstration of a practical approach for facilitating user-centred multi-risk assessment. Furthermore, I believe the manuscript could benefit from a discussion about the challenges associated with expanding or enhancing this type of system for more complex contexts, e.g., involving more than two hazards and/or where there is less well-established means of capturing their interactions.

Thank you for raising these issues. We agree that in this paper the approach is presented for two hazards (with one triggering hazard, i.e. earthquake, and one triggered one, i.e. tsunami). We have also tested our approach also for a situation on volcanic activities with compound hazards, i.e. ashfall and lahars, with damage on buildings and impact on the power network. The case study was located in Ecuador (Cotopaxi volcano). We have focused on the description of one case study in the manuscript, but we will add another paragraph to briefly mention the capabilities of the approach, such as:

*"It should be noted that we tested the approach also for a multi-risk story on volcanic activities with compound hazards, i.e. ashfall and lahars, with damage on buildings and impact on the power network. The case study was located in the area of the volcano Cotopaxi in Ecuador (cf. Gómez Zapata et al., 2021a)."*

We can confirm that our approach is dependent on the chosen multi-risk story as it was altogether decided with the local actors. The conditional probabilities between hazards have not been considered (multi-hazard risk). However, the approach can be extended for such conditional probabilities before the analysis of the dynamic vulnerability. We would like to mention that we deliberately chose the multi-risk approach using a defined story with multiple scenarios. This was mainly due to the joint work with users for whom this approach is easier to understand and to follow (and who are not part of the scientific community) than working with a probabilistic approach.

We will add a paragraph in Section 2.2.1 in the revised version, e.g.:

*"In summary, the selection of the multi-risk story is of crucial importance. It is the basis of our designed multi-risk approach. It is important to note that conditional probabilities between hazards are not currently considered (multi-hazard risks)."*

Furthermore, we will rework the discussion on the decentralized architecture and add another discussion point regarding the transferability of the approach, such as:

Line 554: *"vi. Decentralized architecture: The selected decentralized architecture certainly has advantages ranging from (1) updated information, as the data and models in the specialized institutions are usually refreshed on a regular basis, (2) modularity, flexibility, scalability of multi-risk situations to (3) easier data exchange between institutions as data remain at their point of origin / host. However, despite the use of international standards such as the geospatial WPS defined by the OGC, the integration of new web services into the tool requires adaptations of the underlying orchestration structure. Thus, the (re-)combination of web services to form a new multi-risk chain calls for in-depth knowledge. We do see the potential of the approach for other multi-risk stories, e.g., landslides after an earthquake, failure of drinking water infrastructure, evacuation of the affected population, but recommend to analyse in advance the transfer efforts."*

*"Transferability: The approach was presented for a multi-risk story using two hazards where one hazard (earthquake) is used as the trigger of the second event (tsunami). The approach has been successfully tested for compound hazards (two hazard events happening in parallel). However, one should note that this is a first demonstration. The existing framework of the demonstrator tool serves the basis to be transferred to other areas of interest or adapted to more complex risk contexts (see point iii)."*

3. User input: The user-oriented design of the approach is a welcome feature. However, despite its numerous advantages, there are some "dangers" associated with allowing user input in this type of system. For instance, stakeholders may not be sufficiently educated to appropriate hazard stories, particularly in the context of climate change. A comment on the potential downsides or caveats associated with user involvement should be added to the manuscript, in my opinion.

Thank you for addressing this point on the user-oriented design of our approach. We agree that the user involvement is demanding and must be carried out by experienced professionals. We we emphasize that the process is moderated and the scientific/technical expertise is not outweighed by the user requirements. We did summarize this in the discussion point viii "Co-creation with users" (line 568), and agree that rephrasing and adding more details will enhance the topic on user involvement, such as:

Line 568: *"viii. Co-creation with users: Collaboration between researchers, software developers and different user groups definitely helps to develop a tool that is useful in practice. However, collaboration requires a strong engagement from all sides. It requires a moderated process which allows that user demands can be communicated to the researchers and developers without outweighing the scientific*

*relevance. At the same time, the involved user must be aware and able to cope with trade-offs and compromises, as not all requirements may be addressed or they might not be able to benefit directly from the tool while it is still under development or in a demonstrator stage. To avoid false expectations and misunderstandings, we emphasize that transparency and clear statements are most important throughout the user involvement process. Additionally, it is important to be aware that users (often) do not have the scientific expertise to adequately describe the individual processes in a multi-risk chain. Since the approach is based on the description of a multi-risk story, this story must always be defined in a joint dialog between users, researchers and software developers."*

We also want to point out to line 285 (section 2.2.3 User involvement) which we will further update, as follows:

*Line 285: "(1) Starting point for the approach is the definition of multi-risk stories with the users (Sect. 2.2.1). The joint discussion with the different user groups is intended to ensure the realism and relevance of the stories, thus elaborating a common starting point that will allow structured discussions throughout the design and development process of the tool in order to capture the requirements from the user's point of view. Researchers must ensure that the processes of the multi-risk story are described in a scientifically sound (and possibly abstracted) way. These serve as input, definition and enhancement of the tool and its functionalities."*

4. Case Study: This could benefit from a few more details.

a. It seems that the multi-risk story was pre-defined in the case study (i.e., taken from INDECI, 2017), which is not compatible with the user-centered workflow presented in Section 2.2.3.

Thank you for raising this issue. The multi-risk story was defined jointly with users during a workshop held in Lima on 20 April 2018, and aligned based on the users' recommendation with the description of the reference scenario used in Peru. We will rewrite section 3.2 accordingly to better describe the steps taken of the user-centered workflow:

Line 354: *"Following the story-based concept design (Sect. 2.2.1) we characterized the various elements composing the multi-risk situation which was defined in consultations with Peruvian users (workshop held in Lima on 20 April 2018; see section 3.4). During this user consultation process, users recommended to consider the  reference (worst-case) scenario of an 8.8 Mw earthquake off coast of Lima Metropolitan area as documented by INDECI (2017) when  defining the following story: "Strong shaking occurs in Lima Metropolitan area, Peru, during the day time. There are severe damages on buildings and infrastructure, many people are directly affected by building collapses. As the earthquake has the potential to trigger a tsunami, a tsunami warning is issued and evacuation to safe areas is announced. Coastal roads and roads to highlands become progressively congested. In the following a first tsunami wave impacts the coast and starts inundating parts of the harbour area in Callao. Because of the numerous building collapses, city roads become less suitable for prompt evacuation."*

*For this defined story, multiple scenarios including historical and observed and stochastically earthquakes, were made available. Each earthquake scenario serves as a trigger for the defined multi-risk chain resulting in different cascading impacts.  A flow chart (Fig. 5) was created conceptualizing the main logic, its components and information flows of the multi-risk story."*

b. How is the size of the tsunami related to the magnitude of the earthquake selected and how is the uncertainty in this size accounted for?

Thank you asking this question. The simulations of the tsunami were generated using the physical generation and propagation model TsunAWI (see below). We will add further information in the respective paragraph in Section 3.3.1, line 388 as follows:

*Line 388: "The simulations were generated using the physical generation and propagation model TsunAWI (Harig et al., 2008), which employs a triangular mesh with variable resolution as proposed by Harig et al. (2020). The size of the tsunami is related to the magnitude of the selected earthquake. Generally, larger earthquakes result in larger values of the wave amplitude at the coast and broader inundation area. However, the relation is rather complex, since we account for the vertical displacement of the coastal area due to the earthquake, which might affect the inundation, and additionally, the run-up process is highly nonlinear. Although our approach is scenario-based, we account for uncertainties with regard to the historic earthquake from 1746 which serves as basis for the simulation, by covering a range of magnitudes with simulations. The available outputs including the maximum tsunami amplitude, arrival times and tsunami inundation depth are displayed (Rakowsky et al., 2013; Androsov et al., 2023). Some of these scenario-based tsunami inundation maps are available in Harig and Rakowsky (2021), respectively."*

c. What are the outputs (risk metrics) shown? Are all metrics disaggregated per hazard event? Do they account for cascading impacts (as described in the last two paragraphs of Section 2.2.2)? Was there consultation with the end users on the types of risk metrics to be shown in the system? The conclusion mentions that the platform can be used to compare the results of different stories, but the ability to do this (i.e., show multiple sets of results side by side) is not made clear in the case study description.

Thank you for raising these questions. We structure our answer in three parts:

Outputs (risk metrics)

Our approach focuses on physical and systemic vulnerability (Fig. 5 Flowchart). Regarding the physical vulnerability, the outputs of the risk metrics are direct losses in terms of repairing costs in US Dollars. Every damage state of each fragility function assigned to the corresponding building class has a loss ratio, which is a coefficient associating the replacement value to the total cost of the building unit. This means that we are indeed able to disaggregate the losses for each hazard considered in the multi-risk sequence (details of this method can be found in Gomez Zapata et al., 2023). Dysconnectivity of the nodes that make up the critical infrastructure system (i.e. electric power networks) was the metric selected to assess systemic vulnerability. It was calculated from Monte-Carlo simulations, and the related output per selected hazard scenario are provided in terms of probability of system failure on selected areas (details can be found in Rosero-Velásquez et al., 2022a). A quantitively assessment of cascading effects (e.g., effects on drinking water, failure of telecommunications) were not considered quantitatively mainly due to the scarcity of data, but the possible effects were brought into consideration during the user involvement process. This was transparently communicated following a qualitatively approach.

We will add further information in Section 3.3.1 as follows:

*Line 393: "In order to assess the exposed elements of interest (e.g. residential buildings), exposure models are constructed. They provide information on the location, spatial aggregation and typologies of the residential building stock of Lima Metropolitan area (Yepes-Estrada et al., 2017). Each building typology has associated a fragility function (Villar-Vega et al., 2017) for both hazard-vulnerability schemes (earthquake and tsunami), as documented in Gómez Zapata et al. (2021b). The demonstrator is able to serve these exposure and fragility models through the scripts Assetmaster and Modelprop (Pittore et al., 2021b), which are used as two web services. In order to assess the damage states of the*

*residential buildings and losses (in terms of repairing costs of the corresponding building class in US Dollars) after the occurrence of the selected earthquake the so-called damage exposure update (web) service DEUS is triggered (Brinckmann et al., 2021). Using an updated exposure model that includes earthquake-induced damages, and simulations of tsunami inundation depth as inputs, once again the DEUS web services is initiated in order to approximate the expected cumulative damage and disaggregate the losses per hazard event (Gómez Zapata et al., 2023). This methodology makes use of inter-scheme damage compatibility matrices, that can be consulted in Gómez Zapata et al. (2022c); and a set of state-dependent tsunami fragility functions (Gómez Zapata et al., 2022d), that for the case of Lima Metropolitan area were constructed after having modified the analytically derived ones originally proposed in Medina (2019)."*

Consultation with the end users

As we are dealing we multi-risk risk, we need to have a risk metric that can be transversally used across the different hazards in order to compare the contribution of each hazard scenario. Because of that we used a numerical metric, i.e. the replacement costs (in US Dollars) of the building portfolio. Regarding the damage distribution a colour scheme is per-se a straightforward way in doing so and a suitable way to communicate it to the users. The implemented colour schemes were discussed and agreed upon with the users during the user involvement process. We made sure that both, the colour scheme for the damage distribution (from the different hazards) and the loss (replacement costs) are comparable and easily understandable. This was important to ensure that the outputs are accepted and understood by the users. We will update the following paragraph in Section 3.4, such as:

Line 444: *"In addition, these hands-on sessions allowed many suggestions for improvement regarding the practical handling of the user interface as well as the visual and descriptive presentation of the results. These included comments on the visualization of damage grades as well as losses (in US Dollars), both on colours used and number of grades."*

Compare the results of different stories (i.e., show multiple sets of results side by side)

Your question regarding the comparison of multiple sets of results side by side is appreciated and raise a valid point. In the manuscript we focused on the expert mode of the demonstrator allowing the full exploration possibilities of the developed tool. We are pleased to add further details and figures showing the possibility of our tool to compare different scenarios side-by-side. We plan to update section 2.2.4 and 3.3.2 with more information, such as:

Line 317: *"In order to address different user needs, the GUI was split in an expert and non-expert viewer. The expert viewer ('demonstrator') allows individual setting and configurations of model parameters and outputs, whereas the non-expert viewer ('demonstrator light') runs with predefined parameters and a simplified visualization of results. The underlying web services are identical. Additionally, the 'demonstrator light' provides three modes which allows side-by-side comparison of two scenarios. The 'demonstrator light' provides three different modes for the user to select from: (1) Analysis of one multi-risk scenario; (2) comparison of two different scenarios within one multi-risk story two (e.g., earthquakes of different magnitudes); and (3) analysis of different time steps within a multi-risk scenario."*

Line 411: *"A graphical user interface (GUI) allows the user to independently explore the different risk scenarios making use of the aforementioned web services. The designed GUI is available for the expert (Fig. 6) and non-expert user (Fig. 7). For the expert view ('demonstrator') the  main display is divided into three areas: the map window in the centre, the configuration wizard that controls each web service on the left, and the results panel on the right. In the configuration wizard, the user is guided through the multi-risk story where he can select different parameters according to his specific*

*interests. In the layer control panel, the user can examine and view the processed results and gets more information about the outputs (e.g., legends, detailed descriptions). In order to maintain a solid overview, only the parameters relevant for the currently selected step are highlighted as active which enables intuitive control. In this way, the user does not lose track of the current step in the multi-risk chain, even with a long and complex multi-risk story. In the non-expert view ('demonstrator light') the user can select between three different modes. The viewer shows a reduced configuration wizard including abstracted versions of the results. The split-screen allows the side-by-side comparison of two selected scenarios or the exploration of different steps within one scenario."*

*Accordingly, we will also include further screenshots of the GUI of the demonstrator light, for all 3 modes (as described above).*

d. I think the manuscript could benefit from more figures of the system, particularly the GUI.

Thank you for your comment. In addition to the textual updates (see comment above), we will include more screenshots of the GUI of the demonstrator.

e. The spatial extent of the case study needs to be described, particularly in the context of cascading impacts (see comment 4d).

Thank you for raising this point. The extent of the case study is Lima Metropolitan area, Peru. We will include the administrative boundaries in the map shown in Figure 4 so that the reader is well aware about the spatial extent. We will also update section 3.1 accordingly:

Line 347: *"The approach is demonstrated for Lima Metropolitan area which is composed of five sectors (INEI, 2022), i.e. Lima Norte (8 districts), Lima Sur (11 districts), Lima Este (9 districts), Central Lima (15 districts) and Callao (7 districts) (Fig. 4). The multi-risk story (see section 3.2) including its cascading impacts is applied for this particular case study area."*

5. Introduction: Despite its strengths, I think the introduction section is a bit disorganised. I think that some of Section 1.3 should be moved forward to Section 1.1, such that all content that provides a general motivation for risk management is contained within one section.

Thank you for your suggestion. We appreciate your feedback regarding the introduction section. We will improve section 1 by following your advice in moving parts of Section 1.3 to Section 1.1 and Section 1.2, where applicable. With this, Section 1.3 refers to the global risk reduction goals only.

6. Questionnaire and user feedback: The link between the results shown in Section 3.4 and the questions in the questionnaire needs to be clearer (I cannot find any of the questions mentioned in Figure 7 in the questionnaire questions provided in the supplementary material). The supplementary material should provide all questions, and the results for all questions should be provided in the main text (at least in summary form).

Thank you for raising this issue. We will update the supplement material accordingly and include the questionnaires from all three years. Regarding the link between the original questionnaires (as listed in the supplement material) and the description in Section 3.4 we like to provide additional clarification: as the versions of the questionnaires have changed slightly over time, we have taken the liberty of rephrasing the wording of the selected questions in the manuscript text (which is also caused by the translation from Spanish to English). In order to find the corresponding questions in the supplement material, we will mark them with asterisks. This will allow the reader to easily identify the questions in the additional material as shown in Figure 7.

We acknowledge the interest in the user feedback. As the questionnaires have changed over time, there are only a limited number of questions which were asked in all three years. As we focus in the

manuscript on the evolution of the demonstrator tool over time, we have selected the questions in the short questionnaire which were included in all three years. The long questionnaire, as mentioned in the manuscript, was only created for years 2 and 3. In order to keep the focus of the feedback on the evolution of the tool, we would refrain from including all the results of the questions in the text.

**Minor comments:**

1. Line 180: end users are mentioned here as a stakeholder category but it is not yet clear why they would be considered a separate category in themselves - any of the other stakeholder categories listed here could also be a potential end user of this type of system. I see that the end users are described in more detail in line 281; this explanation should be moved forward to line 180 for clarity. However, the situation is further confused in the conclusions section (point viii) an Figure 2, where stakeholder groups are described as "user groups".

Thank you for highlighting this. We will follow your advice to move the sentence in line 281 further up and rephrase it accordingly:

Line 281: *"It has to be noted that the potential users of the tool span all aforementioned stakeholders (section 2.1). A specific focus was put on the so-called end users (e.g., employees of planning and disaster risk management institutions). However, also representatives from the research community (universities and scientific research institutes), institutions operating information and monitoring systems as well as non-governmental were also involved in the user participation process."*

We will update Figure 2 ("stakeholders" instead of "user groups") and some changes in the conclusions section, point viii, such as:

Line 568: *"viii. Co-creation with users: Collaboration between researchers, software developers and  potential users  definitely helps to develop a tool that is useful in practice. However, collaboration requires a strong engagement from all sides. It requires a moderated process which allows that user demands can be communicated to the researchers and developers. At the same time, the involved user must be aware and able to cope with trade-offs and compromises, as they might not benefit directly from the tool while it is still under development or in a demonstrator stage. To avoid false expectations and misunderstandings, we emphasize that transparency and clear statements are most important throughout the user involvement process."*

When working on a revised version of the manuscript, we will further check the proper wording of stakeholder and users throughout the whole text.

2. Figure 5: A reader may look at this figure and question why an EQ catalogue is an input if we are dealing with a specific earthquake scenario. I think it should be more clearly described in the flowchart that the EQ catalogue is used to choose an earthquake scenario for which the ground motion is simulated (the earthquake scenario itself is currently missing from the diagram).

Thank you very much for raising this question. Our approach is not using one fixed scenario only. The user of the tool is able to select between various different earthquake scenarios. Those different scenarios are furthermore listed in different catalogues. For example, there is a catalogue based on observed earthquakes, i.e. a collection of historical earthquakes events which happened in the past. There is a catalogue which lists earthquake scenarios which were defined by experts containing both real and synthetic events. With this, the initial step is, as displayed in Figure 5, to select an earthquake catalogue. We will follow your advice and include the earthquake scenario as an input in the diagram. Thank you for pointing this out.

Based on your comment we will further update Section 3.2 to ensure that the reader is aware that the approach is not using one fixed scenario only.

Line 354: *"Following the story-based concept design (Sect. 2.2.1) we characterized the various elements composing the multi-risk situation. In order to ensure that the multi-risk situation is described as realistic as possible, we  took into account  the description of the reference (worst-case) scenario of an 8.8 Mw earthquake off coast of Lima Metropolitan area as documented by INDECI (2017). Based on this description and additional consultations with Peruvian stakeholders (workshop held in Lima on 20 April 2018; see section 3.4), we defined the following story: "Strong shaking occurs in Lima Metropolitan area, Peru, during the day time. There are severe damages on buildings and infrastructure, many people are directly affected by building collapses. As the earthquake has the potential to trigger a tsunami, a tsunami warning is issued and evacuation to safe areas is announced. Coastal roads and roads to highlands become progressively congested. In the following a first tsunami wave impacts the coast and starts inundating parts of the harbour area in Callao. Because of the numerous building collapses, city roads become less suitable for prompt evacuation."*

*For this defined story, multiple scenarios including historical and observed and stochastically earthquakes, were made available. Each of these individual scenarios serves as a trigger for the defined multi-risk chain resulting in different cascading impacts.*

* A flow chart (Fig. 5) was created conceptualizing the main logic, its components and information flows of the multi-risk story."*

3. Line 475: I do not believe that a value of 55% could be described as an "overwhelming majority"

Thank you for this comment. We will change the wording accordingly:

Line 469: *"In year 2 (V1.0), already 35 % of the users said that the relevance of the information was very high, while in year 3 (V2.0)  more than half (55 %) rated it as very highly and 31 % as even totally relevant (Fig. 7b)."*

4. Line 485: it seems that the practical usability of the tool actually decreases over time – e.g., 14% said they are totally likely to use the tool in year 3 versus 18% in year 1. Furthermore, Figure 7d does match with the description of these results provided in the text; it is mentioned that 64% rated the possibility of using the tool as highly likely in year 1, but there is no highly likely colour marked on the bottom bar of fig 7d.

Thank you for raising these issues. First of all, we apologize for the mistake in the text. Yes, you are totally correct. The 64% in year 1 refers to 'highly' likely and not to 'very highly' likely. The figure is correct, the mistake in the text will be corrected accordingly.

With this, we can further justify our interpretation of the increase of the practical usability of the tool (and not a decrease over time). Yes, it is correct that in year 1, 18% said they are 'totally' likely to use the tool versus only 14% in year 3. However, in year 3, 39% replied that they 'very high' likely would use the tool, whereas in year 1 no one replied that they are 'very high' likely to use the tool, but only 'highly' likely. With this, we do believe that it is valid to state that there is an increase regarding the feedback on the practical usability of the tool. We will update the paragraph accordingly:

Line 484: *"With regard to possible practical applicability, the question was asked how likely it is that users would use the tool for their practical work. For the V0.1 version presented in year 1, 18 % of the users rated the possibility of using such a tool as moderate, 64 % as  high and 18 % as totally. In year 2 (version V1.0) users responded that it is very less (4 %) and less (4 %) likely of using the tool in their practical work. The majority of users rated the likelihood of using the tool as moderate at 21 %*

*and high with 59 %. While 8 % of users considered this to be very high, 4 % answered that they totally would use such a tool. The practical applicability of version V2.0 in year 3*  *was rated as follows. 8 % of users said they would be moderately likely using the tool, while 39 % said they would be highly likely and 39 % very high likely would use it. Finally, 14 % of users said that they totally likely would use the tool if it was available. Even though there was a slight decrease in the totally likelihood of using the tool (year 1: 18%, year 3: 14%), we believe that is valid to state that the overall likelihood has increased, as 39% were very likely to use the tool in year 3, while this answer was not given at all in year 1 (Fig. 7d)."*

---

## Author Comment (AC2)

**AUTHORS' RESPONSE TO THE REVIEWER#2 COMMENTS**

"Between global risk reduction goals, scientific-technical capabilities and local realities: a novel modular approach for multi-risk assessment" by Schoepfer et al.

**RC2: 'Comment on nhess-2023-142', Anonymous Referee #2**

I thank the authors for their very interesting manuscript on framing multi-risk assessments in the context of a case study in Peru.

I believe that this paper will be an excellent addition to the growing literature on multi-risk assessment, as long as it does not overstate what has been proposed and done, recognises the inherent biases and limitations involved with any such analysis, and considers a bit more strongly the pracitioner stakeholder who might use the methodology proposed (or parts of it).

Below are a series of comments, in no particular order of importance. Although some are slightly critical (the nature of doing a review), most are aimed at making the manuscript more useable and useful by practitioner stakeholders and others who might want to take your learnings and apply them to another region.

Thank you very much for your kind feedback on our manuscript. We appreciate your thorough review and suggestions for improvement. Please find our comment-by-comment feedback (answers in blue; proposed changes in the manuscript in red) as follows. The lines indicated correspond to the PrePrint: https://nhess.copernicus.org/preprints/nhess-2023-142/nhess-2023-142.pdf

1. Title.

a. The paper is much more about your **case study in Peru, so I would expect that to be in the title**.

We agree that the paper is focusing on our case study in Peru. Even though the approach is only exemplified for Lima Metropolitan area, we would prefer in not changing the title. We have also worked in case studies located in Chile and Ecuador (we will briefly report on this in a revised version of the manuscript). Since the approach can be adapted for other regions, we would opt for not including Peru in the title.

b. You use the word **'novel'**. Is this really novel? All the elements have been done previously. I believe the approach and paper are well worth while, just be careful about overstating the originality of what you are doing.

Thank you for pointing this out. Reviewer #1 had a similar comment on the word 'novel'. We agree that individual results of the paper have already been published which we cited accordingly. As the overall conceptual approach has not yet been published in a scientific paper, we felt it justified to characterise the overall approach as 'novel'. It was not our intention to overemphasize the originality. As the opinions of both reviewers are in the same direction, we have reconsidered the title and will delete the word 'novel'.

2. Abstract. This is a bit high level and more a motivation rather than an actual (with metrics such as 'how many' and 'of what') summary of the paper. I suggest you rethink a bit the abstract, and consider more how it is really a summary of the paper.

Thank you for your suggestion. We will rework the abstract carefully in a revised version of the manuscript.

3. Introduction.

a. The introduction does a nice job of bringing in some of the literature, but **I believe there are other major papers** out there that have put into context multi-hazards, multi-risk and multi-impact in the context of natural hazards. Please do a relatively rapid review to ensure you have captured the majority of papers out there that have put into context multi-hazard/risk/impact.

Thank you for your feedback on the introduction. We will include further references, such as:

De Angeli, S., Malamud, B. D., Rossi, L., Taylor, F. E., Trasforini, E., and Rudari, R.: A multi-hazard framework for spatial-temporal impact analysis, International Journal of Disaster Risk Reduction, 73, 102829, https://doi.org/10.1016/j.ijdrr.2022.102829, 2022.

Goda, K. and De Risi, R.: Future perspectives of earthquake-tsunami catastrophe modelling: From single-hazards to cascading and compounding multi-hazards, Front. Built Environ., 8, https://doi.org/10.3389/fbuil.2022.1022736, 2023.

López-Saavedra, M. and Martí, J.: Reviewing the multi-hazard concept. Application to volcanic islands, Earth-Science Reviews, 236, 104286, https://doi.org/10.1016/j.earscirev.2022.104286, 2023.

Šakić Trogrlić, R., Donovan, A., and Malamud, B. D.: Invited perspectives: Views of 350 natural hazard community members on key challenges in natural hazards research and the Sustainable Development Goals, Natural Hazards and Earth System Sciences, 22, 2771–2790, https://doi.org/10.5194/nhess-22-2771-2022, 2022.

Tilloy, A., Malamud, B. D., and Joly-Laugel, A.: A methodology for the spatiotemporal identification of compound hazards: wind and precipitation extremes in Great Britain (1979–2019), Earth System Dynamics, 13, 993–1020, https://doi.org/10.5194/esd-13-993-2022, 2022.

In this way, we hope to find a good balance between a review paper (which is not our aim) and a solid introduction to the topic in the context of our paper. As we incorporate new references in Section 1.2, we will also revise the section's structure.

b. **I did not find it easy to read the introduction due to all the definitions and quotes.** Perhaps consider for the definitions using tables or bullet points so that it is not huge chunks of text with lots and lots of quotes. I've seen half a dozen 'reviews' of the past literature on multi-hazards and multi-risk, and the most useful ones I have seen (from a practical perspective) are those that have tables, figures with timelines, ideas broken out into bullet points, etc. I understand that you do not want to do a complete review of the literature—that is fine, but perhaps one or two table with your key quotes to reduce the text? For example, much of Section 1.3 could be supplemented by a table. Many of the quotes in Section 1.1 could be in a table and then referred to. The studies given in 1.2 would be ideally put in a table, with a few headers to pull out salient parts of the studies, and then discussed in the text.

We will consider your advice and restructure the introduction. We aim to find a balance between in-text citations and the use of tables and/or bullet points. For Section 1.3 we plan to extract the current information on the global strategies from the text and list them in a table, such as:

*Table X: Overview of the key global strategies calling among others for reducing risks and damage from disasters*

| Global strategies | Excerpts and statements | References |
|---|---|---|
| 2030 Agenda for Sustainable Development; Sustainable Development Goals (SDGs) | - 17 goals for improving human society, ecological sustainability and the quality of life are aiming to contribute to the global risk reduction agenda | UNISDR, 2015a |
| | - 25 targets related to disaster risk reduction in 10 of the 17 SDGs | UNISDR, 2015b, p. 2 |
| | - Among others, the objective of reducing the in number of deaths and people affected as well as decrease of economic losses caused by disasters is addressed in goal 11: "*Make cities and human settlements inclusive, safe, resilient and sustainable*" | UNISDR, 2015a, p. 24 |
| Sendai Framework for Disaster Risk Reduction 2015-2030 | - Outlines 7 targets and 4 priorities for action to prevent new and reduce existing disaster risks | UNISDR, 2015a, p. 14 |
| | - Priority 1: Understanding of disaster risk "*Policies and practices for disaster risk management should be based on an understanding of disaster risk* | |

| | | |
|---|---|---|
| | *in all its dimensions of vulnerability, capacity, exposure of persons and assets, hazard characteristics and the environment"* | UNISDR, 2015a, p. 14 |
| | - Calls "*to promote the collection, analysis, management and use of relevant data and practical information and ensure its dissemination, taking into account the needs of different categories of users*" | UNISDR, 2015a, p. 16 |
| | - Advocates "*to support the development of local, national, regional and global user-friendly systems and services*" | |
| Paris Agreement | - International treaty on climate change | United Nations, 2015b |
| | - Calls for *"reducing vulnerability to climate change" in article 7.1* | United Nations, 2015b, p. 9 |
| | - Calls for the *"importance of averting, minimizing and addressing loss and damage associated with the adverse effects of climate change […]" in article 8.1* | United Nations, 2015b, p. 12 |
| New Urban Agenda | - Addresses various field of action and calls for strengthening resilience in the event of disasters | United Nations, 2017 |
| | - Envisages cities and human settlements that "*adopt and implement disaster risk reduction and management, reduce vulnerability, build resilience and responsiveness to natural and human-made hazards and foster mitigation of and adaptation to climate change*" | United Nations, 2017, p. 7 |

4. Conceptual Approach.

a. This is broadly fine, within the limitations of what is presented and has a couple of nice summary figures, but I give a few comments below.

b. **General: Use of the word and approach to vulnerability**. A key part of risk, as you acknowledge, is vulnerability. The word vulnerability comes up 14 times in the manuscript (many of these are part of direct quotes), which is appropriate, but at no place do you define vulnerability (although do mention once physical vs. social vulnerability). For me, a key part of multi-risk (vs. multi-hazard) analyses, is the incorporation of both physical and social vulnerability. I would like to see a more solid defining of vulnerability either in the intro or conceptual approach, along with strengths and limitations of including physical/social vulnerability into multi-risk assessment in terms of data, equations, etc. either when vulnerability is first mentioned or in the discussion. It was not until I got to line 536 that I felt you acknowledged that physical vulnerability only was included and not social vulnerability, and this needs to be acknowledged much earlier.

Thank you for raising this issue. We will indicate on right from the beginning that our approach is focusing on the physical vulnerability and systemic vulnerability only. This will be done at various places throughout the manuscript, such as:

Line 137: *"Following this introduction, Sect. 2 presents the conceptual approach to developing a scenario-based multi-risk assessment tool. With the aim of developing a demonstrator (and not a fully operational system), we focused on analysing the physical vulnerability of buildings (i.e., the likelihood that assets will be damaged or destroyed when exposed to a hazard event), and the systemic vulnerability of electrical power networks (i.e., probability of failure of interconnected systems given hazard intensities).  Sect. 3 describes the results and steps taken, including findings from the user perspective . The discussions and conclusions are outlined in Sect. 4."*

*Line 152: "The conceptualization of this overall approach is visualized in Fig. 1. We argue that the starting point of our conceptual approach is a context and stakeholder analysis (Sect. 2.1) to understand the organizational environment and underlying structures. Later, we present a framework to design a multi-risk information system (Sect. 2.2). We selected a story-based concept that allows the description of a specific multi-risk situation and its representation through multiple scenarios (Sect. 2.2.1). As input, the elements of risk (hazard, exposure, and vulnerability) and their impacts on critical infrastructure are assessed, novel scientific and technical approaches developed and considered in terms of their potential implementation (Sect. 2.2.2). It is worth noting that we devoted efforts to study the interactions at the physical and systemic vulnerability levels from cascading hazards, addressing cumulative damage and losses. During the development of the demonstrator, we involved users  from the beginning to ensure that the designed tool their requirements and needs (Sect. 2.2.3). For the demonstrator we chose a decentralized system architecture approach built on distributed web services, with a graphical user interface as the frontend (Sect. 2.2.4)."*

*Line 535: "iii. Complexity: Multi-risk situations can become very complex. Obviously, models and scenarios are always incomplete as they approximate complex real situations. The analytical process of the interactions of elements in scenarios is furthermore confined to selected processes. In our approach we focused on the physical elements of vulnerability (buildings, critical infrastructure), but neglected the economic, environmental, political, social and societal aspects of vulnerability. It is important to remember that the overall objective was to develop an approach and to demonstrate its potential. Thereby, we aimed to make the framework and its source code publicly available. With this, data restrictions and data protection issue coming along with the social vulnerability, e.g., of demographic and socioeconomic variables, had to be considered. This limitation  resulted in a considerable limited representation of what would actually happen in a real disaster situation. Nonetheless, it is worth noting that potential users have already rated the potential of the tool as high based on the physical and systemic vulnerability results. In addition, users indicated that the tool has already stimulated them to develop new strategies for capacity building and resilience measures."*

c. Again, like the intro, I found **there was a lot of text to go through in the conceptual approach**, to get to the practical 'how is this being done' parts. Might you break some of the text into bullet points or numbers, to make it easier to read? I'm thinking of the practitioner (who you have aimed part of this paper at) who wants to know what to do, how to do it, and limitations.

We are aware that there is a lot of information provided. Our main target group of this paper is the scientific community and not practitioners. With this, we aimed in providing sufficient details including a wide range of references. We will review the manuscript for possible improvements regarding the use of bullet points or numbering as suggested.

d. Riesgos and Riesgos 2 are mentioned on line 148 (the only place in the text) and then on the data and code availability section. This code seems essential for a practitioner to operationalize the approach suggested here in a pracitical way (and which you do a test case study with peru). **I would suggest you have 1-2 paragraphs outlining more about Riesgos Code Availability and Use (or refer the reader explicitly to the places they can read about how to use it) with text both in this section and then again in the next section (Peru Case Study)** so that they can better understand the theory going into practice, or more importantly, how would they begin to implement the learnings from this paper if they were interested.

Thanks for pointing this out. We will follow your advice and include additional paragraphs on the code availability in Section 2 'Conceptual Approach' and Section 3 'Results', such as:

Section 2.2.4: *"For the developments, both backend (web services) and frontend (graphical user interface), we aimed for open source. This allows others to not only use this software but also to replicate the tool and to develop it further."*

Section 3.3.2: *"The web services and the graphical user interface are published online so that the preconditions for further development into an operational system are given. Details on the code availability on GitHub - a platform for managing, versioning and sharing source code - are provided in the respective section of the paper (see Code and data availability)."*

5. Peru Case study

a. Some very nice figures and flowcharts, but please **reevaluate the white text in Figure 5** (not easy to read on my PDF), and where possible **enlarge font size on figures**.

Thank you for the positive feedback on the illustrations. We will adjust Figure 5 as suggested and check the readability (e.g., adjustments of font size where applicable) of all figures.

b. Broadly I was fine on the approach taken. It does get at a number of interesting aspects of multi-risk (although not social vulnerability).

We are pleased to read that you are generally satisfied with the approach taken. We are aware that the social vulnerability is an important part, which we unfortunately could not cover yet. For demonstration purposes of our approach, and the tool development, we focused on the physical vulnerability. For future developments, we recommend taking the social vulnerability into account. Now that we have been able to show that the approach is overall suitable, this would definitely be an important step in further increasing the added value of the tool.

c. Please state somewhere the **ethical procedures** you went through before working with the human participants.

Thanks for raising the topic of ethical procedures. We will update section 3.4 with additional information:

*"During the overall process we respected the ethical principles and guidelines for research involving human subjects (European Commission, 2021). Informed consent was achieved by providing details about the purpose on the research and the roles of the different actors involved. Involved stakeholders were further informed how the information will be used. The participation was voluntary. Above all, we respected the confidentiality as all questionnaires were anonymous and did not allow individuals to be identified. Neither minors nor people with limited capacity were involved in our project."*

*European Commission: Ethics in Social Science and Humanities. Online: https://ec.europa.eu/info/funding-tenders/opportunities/docs/2021-2027/horizon/guidance/ethics-in-social-science-and-humanities_he_en.pdf, last access: 03 May 2024, 2021.*

d. Be careful of **typos**. Lines 374-375. Earthquakes appears twice

Thanks for pointing this out. We will correct the typo accordingly and do another check of the manuscript for typos.

e. Use of **Tables**: This section might benefit by **an additional table summarizing the data used, their sources, key parameters, and any comments such as regarding uncertainty**.

We will follow your suggestion in including an additional table. At this stage we plan to add the following details in section 3.3.1 'Web services and workflow control'. Please kindly note that we will include the references (listed in this table) in a revised version manuscript.

*Table 1. System components (web services) with details on input data/model, source and output for the multi-risk story for an earthquake / tsunami event affecting housing and the critical infrastructure power grid. EQ = Earthquake; TS = Tsunami; CI = Critical infrastructure.*

| Web service | Input data/model and source | Output data/model |
| --- | --- | --- |
| EQ catalogue "Quakeledger" (Pittore et al., 2021b) | - Earthquake catalogues as compiled by the SARA project for subduction events (Pagani et al., 2021).
- Filter parameters: depth, magnitude, and a geographic area that is defined by a bounding box upon user request. | - List of earthquakes for subduction interface that matches the filter criteria defined by the user. |
| EQ ground motion simulation "Shakyground" (Weatherill et al., 2021) | - Earthquake source parameters (hypocentral location, depth, and strike, dip and rake angles).
- OpenQuake Hazard Library (Pagani et al., 2014) to generate finite fault ruptures as a function of their source properties.
- Ground motion prediction equation (GMPE) for subduction interface (e.g. Montalva et al., 2017).
- Gridded values of shear wave velocities for the uppermost 30 m depth ($Vs_{30}$). For Lima, the dataset of Ceferino et al., (2018) was used. It compiles slope-based $Vs_{30}$ (Allen and Wald, 2007), and the seismic microzonation for Lima defined by Aguilar et al., (2013). | - The Demonstrator displays the ground motion fields of mean acceleration values for the target intensities (i.e., peak ground acceleration (PGA)). They are forecasted at each site of the $Vs_{30}$ grid by the selected GMPE.
- Additionally, 1000 realisations of ground motion fields with uncorrelated and cross-correlated residuals for PGA and spectral accelerations at 0.3 and 1.0 seconds for six earthquake scenarios (Mw 8.5 - 9.0) are reported in the repository of Gómez Zapata et al., (2021c). |
| EQ exposure model "Assetmaster" (Pittore et al., 2021a) | - Official census dataset at the block level (INEI, 2017), which contains a few attributes for dwellings.
- "Mapping schemes" that relate census attributes, dwellings-to-buildings fractions, and seismic-oriented building classes (GEM, 2014).
- Seismic-oriented residential building classes as defined by the SARA project, and their inferred replacement costs (Yepes-Estrada et al., 2017).
- Focus maps that spatially combine tsunami inundation and population, which to generate exposure aggregation areas (Gómez Zapata et al., 2021a). | - Exposure model for residential buildings for earthquake risk applications reported in the repository of Gómez Zapata et al., (2021b). They are GEOJSON files that contain the building counts per type spatially aggregated at the block-level, and on CVT-based (Central Voronoi Tessellations) geocells. The metadata of these exposure files match the metadata of the fragility files served by "Modelprop". |
| Precomputed TS simulations for each associated EQ using TsunAWI (Harig et al., 2008) | - Bathymetry (GEBCO, 1km raster)[1].
- Coastal topography by SRTM[2] (raster data sets at 30m).
- TanDEM-X[3] (at 12m) (Krieger et al., 2007). | - Maximum tsunami amplitude (in meters).
- Arrival time (in minutes).
- maximum tsunami inundation depth (in meters) (Harig and Rakowsky, 2021). |
| TS fragility model "Modelprop" (Pittore et al., 2021a) | - Building fragility functions for seismic ground-shaking (Villar-Vega et al., 2017).
- Two types of tsunami fragility functions for buildings: analytical (Medina, 2019; | - The fragility functions are expressed JSON files. Their metadata matches the exposure models served by "Assetmaster", and the earthquake and tsunami |
* * *
[1] General Bathymetric Chart of the Oceans (GEBCO), GEBCO 08 Grid, http://www.gebco.net
[2] Shuttle Radar Topography Mission (SRTM), 30m resolution, https://www2.jpl.nasa.gov/srtm
[3] TerraSAR-X add-on for Digital Elevation Measurement (TanDEM-X), 12m resolution, https://www.dlr.de/en/research-and-transfer/projects-and-missions/tandem-x

| | Medina et al., 2019), and empirical (Suppasri et al., 2013). | intensity measures of interest for subsequent risk assessment. |
|---|---|---|
| 1st run of the software DEUS (Damage-Exposure-Update-Service) (Brinckmann et al., 2021) | - Ground motion fields (PGA; SA 0.3; SA 1.0 seconds) served by "Shakyground".
- Seismic-oriented exposure model for residential buildings served by "Assetmaster".
- Seismic fragility functions served by "Modelprop" (state-independent). | - Spatial distribution of EQ damage in the form of a damage-updated exposure model. The damage scale of EQ is used.
- Spatial distribution of direct EQ losses (replacement costs in USD). Example outputs are compiled in Gómez Zapata et al., (2021d). |
| 2nd run of the software DEUS (Brinckmann et al., 2021) | - Raster files of TS inundation depth per EQ scenario, precomputed by TsunAWI.
- Damage-updated exposure model (containing EQ damage) served by "Assetmaster".
- - Building inter-scheme conversion matrices. They express the probabilistic compatibility between the EQ building classes and the TS ones. They were generated through the taxonomic disaggregation method of Gómez Zapata et al., (2022a). The script is available in Gómez Zapata et al., (2021c).
- Damage inter-scheme conversion matrices. They express the probabilistic compatibility between the EQ damage states and the TS ones. They were generated through the method of Gómez Zapata et al., (2023). The script is available in Gómez Zapata et al., (2022b).
- State-dependent tsunami fragility functions served by "Modelprop". They are generated by modifying the functions of Medina, (2019). The script and files are available in Gómez Zapata et al., (2022a). | - Spatial distribution of EQ+TS damage in the form of a damage-updated exposure model. The damage scale of TS is used.
- Spatial distribution of direct EQ+TS losses (replacement costs in USD). Example outputs are compiled in Gómez Zapata et al., (2021d). |
| CI System Reliability (Rosero-Velásquez, 2020; 2024) | - Ground motion fields served by "Shakyground".
- Seismic fragility functions for power network facilities (e.g., substations and power plants), based on HAZUS (FEMA, 2003)
- The power network topology and information were obtained (publicly available or upon request) by OSINERGMIN (2019) and COES (2019), and were adapted to the web service as shown in Merscher (2020).
- The calculation of the output is based on a network model for simulating cascading failures (Crucitti et al., 2004; Hernández-Fajardo et al., 2013). | - Probability of service failure (in percentages). |

Regarding the uncertainty we can note that we identified uncertainty factors and classified them into aleatoric and epistemic uncertainty. A dedicated paper (Rosero-Velásquez et al.) is currently under preparation.

f. I found the user groups were interesting, but I would like to see

i. **a much better definition of the user groups** given in Figure 2 (which I assume were then used in Peru) and

ii. some **idea of the user group numbers involved and where they were located**—in other words, why were they representative.

iii. I also am **not a fan of the word 'end users'** as everyone in the research community, NGOs, etc., are end users. This is why (see above) I'd like a much better defining of who was actually involved. You have a couple lines on this in in 280-282, but then when we get to Section 3 you do not refer back to this discussion, and it should be more in-depth.

iv. In all of the reporting of the results you state things like "18% of the the users"—I assume this means that you have now put all the users together into one big group. **Remind us in a few strategic place 'how many'. So 18% of the ### users.**

Thank you for your feedback on the user groups. Based on your feedback as well as based on the comments provided by Reviewer#1, we recognise that the user feedback process was not sufficiently described. We also agree that the term 'end user' can be misleading. We will revise the manuscript accordingly. This starts with a better distinction of the stakeholder categories in Section 2.1, such as:

Line 181: *"A stakeholder analysis has been done to identify relevant actors involved in the DRM context, describing their roles, responsibilities, relationships, interests, and relative influence / power. Naturally, the stakeholders belong to different sectors, i.e.:  (1) research community, (2) institutions operating hazard information and monitoring systems, (3) institutions operating DRM information  systems, (4) institutions working on local and regional level in DRM contexts , and (5) institutions working on national planning level . Key stakeholders per group were identified and described in detail on different levels ranging from national and regional to local level covering their specific objectives and tasks in the working contexts.*

We also plan to rename the section headings as this should provide better clarification:

Section 2.2.3 'User involvement' => change to 'Feedback process from the user perspective'

Section 3.4 'User feedback' => change to 'Findings from the user perspective'

With this, we will also update Section 2.2.3 (including an update of Figure 2).

Line 271: *"Accordingly, we geared our approach to the needs of potential users and its practicality (cf., user-centered design; Gould and Lewis 1985; Karat, 1997)  throughout the design and development process (Fig. 2)."*

Line 278: *"The development of our multi-risk assessment tool is based on a structured and systematic feedback process from the user perspective . Thereby the goal was to target various representatives from different stakeholder groups.  The following table shows the stakeholder groups (section 2.1) and key stakeholders involved."*

*Table X: Stakeholder groups and stakeholders involved in the feedback process from the user perspective.*

| Stakeholder group | Stakeholder involved in the feedback process |
|---|---|
| Research community | - Universities
- Research institutes |
| Institutions operating hazard information and monitoring systems | - Geological institute
- Geographical institute
- Geophysical institute
- Oceanographic institute |
| Institutions operating DRM information systems | - National institutions for risk analysis, risk reduction and risk mitigation
- National civil protection agency |
| Institutions working on local and regional level in DRM contexts | - Disaster management authorities
- Municipalities
- Non-governmental organisations (NGOs) |
| Institutions working on national planning level | - National Center for Strategic Planning
- Ministry of Housing, Construction and Sanitation
- Ministry of Transport and Communications |

In the section in the results, we will provide further details for a better understanding. We will make clear that we have not differentiated between the individual stakeholder groups, but have collected the overall feedback from the users' perspective. Furthermore, we will add the number of participants for a few specific results, such as:

Line 457: *"In year 3 (V2.0), the majority of all respondents (89 % of 37 participants) agreed that the clarity of the information displayed in the demonstrator was highly (62 %), very highly (19 %) or even totally understandable (8 %) (Fig. 7a)."*

Line 468: *"In year 2 (V1.0), already 35 % of the users said that the relevance of the information was very high, while in year 3 (V2.0) an overwhelming majority (55 % of 46 participants) rated it as very highly and 31 % as even totally relevant (Fig. 7b)."*

6. Discussion and conclusions.

a. I found the basic ideas in the discussion and conclusions good, but felt it was rather short and did not bring us back to the overall literature of what others have done. Please relate many of **your key points back to the existing literature**.

Thanks for pointing this out. We will revise this section to better relate to Section 1 and existing literature. For this, we plan to add further details in the discussion section, for example:

*"Relevance and acceptance: Users have recognized the relevance of the topic right from the beginning and have expressed a high demand. This is certainly also due to the fact that the topic of multi-risk is becoming increasingly relevant in practice and that there are still few practical options available for dealing with these new challenges. Various users wanted to use the tool directly in its first version as they recognized great potential in communicating scientific results to decision-makers. With this, we emphasize to follow the recommendations on supporting the development of user-friendly systems and services as articulated in the Sendai Framework (UNISDR, 2015a, p. 14-16)."*

Regarding the state of the art, we refer to our reply to question 6d) on research needs.

b. The approach relies heavily on the **availability of detailed data** (e.g., about hazards, vulnerabilities, and exposures). In regions where such data may be lacking or outdated, the application of the methodology could be challenging. Might you be able to acknowledge **more these limitations and suggest potential solutions or workarounds**?

Data availability is in fact a controversial topic. We plan to add a paragraph on the data availability in the discussion, such as:

*"Data availability and data exchange: As experience shows, data is often available, but data exchange remains challenging. The use of web services is a promising option for the exchange of information between institutions. Data do not need to be stored at a centralized place (and with this gets outdated), but can be updated regularly by the host. An open data policy (FAIR principles) eases this process, but calls for inter-institutional agreements and rules of procedure. Where data availability is still critical (e.g., detailed exposure information) the scientific community can support the creation of enhanced datasets. At this point, it should be noted that users are often satisfied with rough estimates of 'What-if scenarios'. In strongly application-oriented research, it is important to find a balance between maximum accuracy and practical applicability."*

c. The paper focuses on a specific case study area, and while this demonstrates the practical application of the methodology, there is **limited discussion on its scalability and adaptability to other regions** with different risk profiles and socio-economic contexts.

We are pleased to take up the topic of transferability and scalability in our discussion and intend to include a paragraph, such as:

*"Transferability and scalability: The approach was presented for an earthquake-tsunami multi-risk story. Regarding the transferability to another region we can report that we could successfully adapt the approach for another case study in the coastal area of Greater Valparaíso, Chile. The approach has also been successfully tested for compound hazards (two hazard events happening in parallel). During this study, we also tested again the transferability to another region as the study area was located around the volcano Cotopaxi in Ecuador. However, one should note that this is a first demonstration. The existing framework of the demonstrator tool serves the basis to be transferred to other areas of interest or adapted to more complex risk contexts (see point iii on 'Complexity')."*

d. The discussion could be strengthened by a more **explicit identification of gaps in the current approach**. This would not only highlight areas for improvement but also encourage further research in the field of multi-risk assessment.

We are aiming to extend Section 4 'Discussion and conclusions' with further information on research needs, such as:

*"In conclusion, we have demonstrated that the tool is capable of calculating and visualizing the cumulative effects of successive hazard events. Despite some limitations, in particular with regard to already standardized planning processes and the exploratory nature of the tool, users see great potential for different fields of application and a high expectation was expressed, especially from the user side in the local pilot area, that the developed tool would be available and applicable locally. Based on these findings, it appears reasonable that the research community continues working with users on the ground. Further research in the field of multi-risk assessment is certainly needed, among others, to improve the physical vulnerability assessment of various hazards. The standardization of damage scales into a transversal one across hazards will be an important aspect for the scientific community to address. Complementarily, the derivation of state-dependent analytical fragility also deserves more research attention to be optimised in the future through more refined approaches. We also recommend*

*to further work to integrate the social vulnerability. Here, it could be of particular importance to investigate whether and how the social vulnerability of certain demographic groups differs in terms of their response to future crises. Our findings also support the call to science to contribute to an evidence-based policy. At a next step, the impact of such a system in terms of cost-benefit would be interesting to evaluate. After all, the future will tell us how much such a tool can help in planning for catastrophic events and what, in the end, can technologically not be forecasted  but is simply fate."*

e. **Actionable Recommendations**: Make the recommendations actionable by providing clear, specific steps that can be taken by researchers, practitioners, or policymakers. For instance, instead of broadly stating the need for further research, specify the types of studies or methodologies that could address existing gaps.

The original version of the manuscript was submitted on 31 July 2023. Since then, further work has been done by the team. We have created a 'Policy Brief' which documents lessons learnt and proposes selected recommendations for action. The reference of the document is as follows:

Schoepfer, E., Juzam, L., Lauterjung, J., León, C. D., Riedlinger, T., Spahn, H., and Zambrano, A. (eds.): Policy brief - Multi-risk analysis: What would happen if…? https://doi.org/10.15489/cwgicmtcja61, 2024.

We will put a brief reference to this document in Section 4 'Discussion and conclusions':

*"Further lessons learnt and recommendations for action are given in Schoepfer et al., 2024."*

Regarding the proposed studies and methodologies, we refer to our reply to question 6d).

f. **Highlighting Implications for Policy and Practice**: Explicitly articulate the implications of your findings for disaster risk management policy and practice. This could include suggesting changes to existing frameworks or identifying new areas for policy development.

Following the previous comment, we are going to refer to the 'Policy Brief' which lists specific recommendation.

7. Overall.

a. While the innovative methodology is a strength, its complexity could be a barrier to its widespread adoption. **The text could benefit from a more simplified explanation in places or additional step-by-step guides that could make the approach more accessible to practitioners who may not have a strong technical background**.

As briefly stated in our reply to Question 4c, we aimed to find a balance in providing sufficient information for researchers and developers, as well as providing details about the potential practical application. The approach cannot be implemented by practitioners themselves, but only be implemented by specialised institutions. With this, we are not addressing the practitioners with this paper. However, we hope that our additional reference to the recently published 'Policy Brief' will be of benefit to readers.

b. Overall there is a high level of writing, **but tending towards VERY long paragraphs, which often could be broken up into two, or better use of bullet points**.

Thank you for pointing this out. We will go through the manuscript and do a thorough revision of the language style.

c. This will most likely go through copy editing, but there are places where text could be improved. Long sentences are often used where they could be broken up into two or shortned. Some examples (there are many) include.

We express our sincere gratitude to you for providing these suggestions. We will update the manuscript accordingly and check for further improvements in readability.

• **Original Lines 19-21:** "The complex relationships between multiple and consecutive natural hazards exposed population and built environment result in a variety of cascading effects which if are often not considered appropriately by decision makers can result to inadequate or even misleading risk management strategies."

• **Suggested Revision in two sentences:** "Complex interactions among multiple and consecutive natural hazards, the exposed population, and the built environment can lead to cascading effects. If not accurately considered, these can lead decision-makers to implement inadequate or misleading risk management strategies."

• **Original Lines 27-29:** "Based on recent scientific and technical capabilities we developed a tool through an iterative participative approach which has allowed users to explore various scenarios of multiple hazards cascading effects and their impacts."

• **Suggested Revision in two sentences:** "Leveraging the latest scientific and technical advancements, we developed a tool via a participatory iterative process. This tool enables users to explore various scenarios, including the cascading effects of multiple hazards and their impacts."

• **Original:** "In addition to immediate crisis management and rapid response during and after a disaster, disaster preparedness is becoming increasingly important."

• **Suggested Revision:** "Beyond immediate crisis management and rapid response, disaster preparedness is growing in importance."

• **Original:** "The shift from managing disasters to managing risk is articulated in the Sendai Framework for Disaster Risk Reduction 2015-2030, which was adopted at the Third UN World Conference in Sendai, Japan, on March 18, 2015."

• **Suggested Revision:** "The transition from disaster management to risk management is emphasized in the 2015-2030 Sendai Framework for Disaster Risk Reduction, adopted at the Third UN World Conference in Sendai, Japan on 18 March 2015."

• **Original:** "An increasing number of people worldwide are exposed to natural hazards, particularly in poorly planned urbanisations, where effective prevention and risk management can save lives and reduce all kinds of losses."

• **Suggested Revision:** "More people globally face natural hazards, especially in poorly planned urban areas where effective prevention and risk management could save lives and minimize losses."

• **Original:** "For instance, in the context of seismic hazard, information on the possible earthquakes that can hit a region in the future needs to be available. For that aim, existing earthquake catalogues are gathered."

• **Suggested Revision:** "For example, in seismic hazard assessment, future earthquake risks require access to existing earthquake catalogues."

• **Original:** "However, the design of information systems or tools that are capable of analytically exploring multi-hazard risk situations and, in particular, dynamically updating the damage on exposed elements due to various hazards with cascading effects remain challenging."

• **Suggested Revision:** "Designing information systems or tools to dynamically analyze multi-hazard risks and dynamically update exposed element damages from cascading hazard effects presents significant challenges."

---

## Referee Report (RR1)

**Between global risk reduction goals, scientific-technical capabilities and realities: a modular approach for multi-risk assessment**

I thank the authors for their efforts to address my previous comments. Their responses have raised a few further queries/comments from me, which I think should be addressed before the manuscript can be published:

1. Novelty: The authors have improved the framing of the novelty of the proposed framework in terms of its practical relevance and user-centered design. But I think this framing should also be captured in the title of the manuscript, which is currently lacking any reference to user-centeredness.

2. While I appreciate the efforts of the authors to re-structure the introduction to improve its readability, it is still not clear to me why Section 1.3 deserves to be standalone, when its main aim seems identical to that of Section 1.1, i.e., providing motivation for risk management. If the authors insist on keeping it separate, then I believe its title needs to be modified given that it no longer discusses local impacts.

3. Line 240 (approximately): Some of the re-defined stakeholder categories appear to overlap. I am not sure what the difference between an institution operating a DRM information system and one that works "in DRM contexts" would be. Perhaps providing some examples as well as a rationale for the categories defined would be beneficial. Table 2 is helpful for the former, but does not come until much later in the text (it is also not clear to me whether the stakeholders listed in this table were those from the Peru case study or external to that process).

4. Line 274: If conditional probabilities of hazards are not considered, how do you ensure that the multi-risk "stories represent *realistic* multi-risk situations", as stated in line 265? (By the way, it does not seem at least entirely correct to state that conditional probabilities of hazards are not considered, given that the size of the tsunami in the case study is related to the magnitude of the considered earthquake).

5. Line 482: "we account for uncertainties with regard to the historic earthquake from 1746 which serves as basis for the simulation, by covering a range of magnitudes with simulations".. I am not sure I follow this statement. Does the word "magnitudes" used here refer to the earthquakes? If that is the case, I thought only one magnitude is considered (rather than multiple), given that the authors state in line 463 that the scenario is "a single event".

6. I think the results of the Ecuador application need to be elaborated more, explaining more specifically how the transferability to another region was tested. In particular, it would be relevant to explain how stakeholders were involved in that application and what challenges arose (e.g., around the design of the interface, any modifications that were suggested for the workflow). Furthermore, the authors state that "we can report that we could successfully adapt the approach for another case study in the coastal area of Greater Valparaiso, Chile" but provide no evidence to support this statement (it is not even clear what hazards are considered for this additional case study), which is not acceptable. The statement should either be removed or the case study described in sufficient detail to support the claims made about it.

7. It is surprising that the risk metrics output from the tool are not designed in collaboration with stakeholders, given their importance to the decision-making processes that the tool strives to facilitate. Furthermore, the metrics used are rather narrow in scope; repair costs may not fairly capture the effects of hazards on low-income populations, for instance (e.g., see https://www.nature.com/articles/s41893-

020-0508-7). I think the narrow range of metrics used and the lack of stakeholder involvement in their design might be considered a notable limitation of the framework and an aspect that should be improved in future work. I therefore believe it warrants some discussion in the conclusions.

Some minor technical comments:

1. Line 30: I think the phrase "tool through an iterative participative approach" was more a more appropriate description than the term "method"
2. Line 405: There are two sentences here that both state the demonstrator light version provides three different modes. Also, there is a typo in the statement: "comparison of two different scenarios within one multi-risk story two.." (same typo exists in line 529)
3. Line 568: please specify what you mean by "users"
4. Line 628: "totally likelihood" does not make sense .. rephrase using something along the lines of "proportion who are totally likely to use the tool"
5. Line 656: this sentence does not read correctly. I think you should rephrase the start of it along the lines of: "In this paper, we presented one (of many possible) approach(es) to multi-risk analysis that can make a practical contribution to…"
6. Figure 8: "right ride" is a typo in the caption

---

## Referee Report (RR2)

I thank the authors for their further efforts to respond to my comments, and I believe they have largely addressed my remaining concerns. I have just two outstanding minor suggestions:

1. In the caption of Table 2, you should make it clear that the stakeholders shown in the table for each group are representative of the Peruvian context, which is the study area to be discussed later in the manuscript. (Otherwise, I think the reader might be confused as to why there is a sudden focus on Peru).
2. As I mentioned in the previous review, the conditional probabilities of hazards are somehow being accounted for given that the tsunami and the earthquake are not simulated independently (i.e., the same fault parameters are used for both, and the size of the tsunami is related to the magnitude of the earthquake). So, I suggest removing the sentence about conditional probabilities not being accounted for (line 240 of the current manuscript).

- Please note that I also noticed the Jimenez et al. (2013) paper is missing from the reference section.

---

## Author Response (AR2)

**RC2: AUTHORS' RESPONSE TO THE REVIEWERS COMMENTS**

**nhess-2023-142**: "Between global risk reduction goals, scientific-technical capabilities and local realities: a novel modular approach for multi-risk assessment" by Schoepfer et al.

- page 1-5: Authors' response to the Reviewer#1 comments
- page 6-12: Authors' response to the Reviewer#2 comments

Line numbers refer to the track change version (nhess-2023-142-manuscript-version3_ATC.pdf).

**Anonymous Referee #1, RC2: 'Comment on nhess-2023-142'**

I thank the authors for their efforts to address my previous comments. Their responses have raised a few further queries/comments from me, which I think should be addressed before the manuscript can be published:

1. Novelty: The authors have improved the framing of the novelty of the proposed framework in terms of its practical relevance and user-centered design. But I think this framing should also be captured in the title of the manuscript, which is currently lacking any reference to user-centeredness.

*Thank you for your feedback and suggestion. We changed the title to "Between global risk reduction goals, scientific-technical capabilities and local realities: a modular approach for user-centric multi-risk assessment".*

2. While I appreciate the efforts of the authors to re-structure the introduction to improve its readability, it is still not clear to me why Section 1.3 deserves to be standalone, when its main aim seems identical to that of Section 1.1, i.e., providing motivation for risk management. If the authors insist on keeping it separate, then I believe its title needs to be modified given that it no longer discusses local impacts.

*We have considered your comment and changed the section title from "From global risk reduction goals to local impacts" to "From global risk reduction goals to local solutions". We aim not to merge Section 1.3 with Section 1.1 as we want to dedicate a specific section to the global strategies highlighting the importance of aiming for practical solutions.*

3. Line 240 (approximately): Some of the re-defined stakeholder categories appear to overlap. I am not sure what the difference between an institution operating a DRM information system and one that works "in DRM contexts" would be. Perhaps providing some examples as well as a rationale for the categories defined would be beneficial. Table 2 is helpful for the former, but does not come until much later in the text (it is also not clear to me whether the stakeholders listed in this table were those from the Peru case study or external to that process).

*Thank you very much for this comment. Based on your comment, we now moved Table 2 from its current position to Line 244 and add "Peruvian" to the heading of Table 2:*

*Line 205: "Table 2. Peruvian stakeholder groups and the stakeholders involved in the feedback process from the user perspective."*

4. Line 274: If conditional probabilities of hazards are not considered, how do you ensure that the multi-risk "stories represent realistic multi-risk situations", as stated in line 265? (By the way, it does not seem at least entirely correct to state that conditional probabilities of hazards are not considered, given that the size of the tsunami in the case study is related to the magnitude of the considered earthquake).

The term "realistic multi-risk situation" refers to the fact that we consider a physically sound realization of a multi-hazard event. Both the estimates of earthquake and tsunami impact are based on identical fault parameters describing the earthquake which results in damages due to ground acceleration, but is also the source of the tsunami leading to coastal inundation, which is simulated at resolutions high enough for realistic estimates of the flooding extent. The actual amount of "realism" with respect to historic events depends on the correctness of fault parameters and representation of the bottom relief and roughness, as it is already stated in the text (see e.g. in line 235: "*The quantitative models in the individual scenarios do not necessarily represent the entire complexity of a story. To which degree a story agrees with realistic circumstances depends on the modelling capabilities as well as on the availability of (geo-)data.*"). We clarified the description of the story-based concept.

Line 225: *"With these objectives in mind, we followed the concept of story and scenarios in order to understand and describe possible multi-risk situations (e.g., Jarke et al., 1998; Sutcliffe, 2003). With the term story we refer to a "narrative description of a situation, defining the specific involved hazards, cascading effects and impacts, looking at a specific area of interest". These stories represent realistic multi-risk situations with cascading effects. We ensure physically sound settings of the multi-hazard situation by performing all calculations with identical fault parameters for both earthquake and tsunami simulations. However, although a story is based on physical drivers – i.e., natural hazards, – it is not limited to their description alone."*

5. Line 482: "we account for uncertainties with regard to the historic earthquake from 1746 which serves as basis for the simulation, by covering a range of magnitudes with simulations".. I am not sure I follow this statement. Does the word "magnitudes" used here refer to the earthquakes? If that is the case, I thought only one magnitude is considered (rather than multiple), given that the authors state in line 463 that the scenario is "a single event".

Thank you for pointing this out, indeed, a scenario represents a single event, thus magnitude as well as all fault parameters are fixed. The mentioned sentence is somewhat misleading and refers to the coverage of several magnitudes by different scenarios in the database. Since the correct magnitude of the event in 1746 cannot be specified exactly, we covered a range from Mw 8.5 to Mw 9.0 by different scenarios and different extents as suggested by Jimenez et al. (2013) and other sources. These realizations and corresponding impacts may be investigated and compared as separate scenarios in the Demonstrator. We modified the corresponding statement to clarify the meaning. It reads now as follows:

Line 445: *"For those earthquakes which can potentially trigger a tsunami, another web service is introduced which provides access to pre-calculated numerical tsunami simulations. The simulations were generated using the physical generation and propagation model TsunAWI (Harig et al., 2008), which accounts for a triangular mesh with variable resolution as proposed by Harig et al. (2020). The size of the tsunami is related to the magnitude of the selected earthquake. Generally, larger earthquakes result in larger values of the wave amplitude at the coast and broader inundation area. However, the relation is rather complex, since we account for the vertical displacement of the coastal area due to the earthquake, which might affect the inundation, and additionally, the run-up process is highly nonlinear. Based on the earthquake catalogue, a database of tsunami scenarios with earthquake sources offshore Peru was calculated. In case of the historic earthquake from 1746, we account for uncertainties by incorporating several scenarios covering a range of fault parameters for the source area as suggested by Jimenez et al. (2013). The available outputs including the maximum tsunami amplitude, arrival times and tsunami inundation depth are displayed (Rakowsky et al., 2013; Androsov et al., 2024; Harig et al., 2024). Some of these scenario-based tsunami inundation maps are available in Harig and Rakowsky (2021), respectively."*

6. I think the results of the Ecuador application need to be elaborated more, explaining more specifically how the transferability to another region was tested. In particular, it would be relevant to explain how stakeholders were involved in that application and what challenges arose (e.g., around the design of the interface, any modifications that were suggested for the workflow). Furthermore, the authors state that "we can report that we could successfully adapt the approach for another case study in the coastal area of Greater Valparaiso, Chile" but provide no evidence to support this statement (it is not even clear what hazards are considered for this additional case study), which is not acceptable. The statement should either be removed or the case study described in sufficient detail to support the claims made about it.

Thank you for pointing this out. We included more information in Section 3.2 and Section 4 regarding the case studies in Ecuador and Chile (see discussion point "*vii. Transferability and scalability*").

Line 409: *"For this defined story, multiple scenarios including historical, observed* **as well as** *stochastic earthquakes, were made available. Each earthquake scenario serves as a trigger for the defined multi-risk chain resulting in different cascading impacts. A flow chart (Fig. 5) was created conceptualizing the main logic, its components and information flows of the multi-risk story.* **A database of historical, observed and stochastically distributed earthquakes with different locations and magnitudes was developed and made available via a web service. As each of these individual earthquakes serves as a trigger for the defined multi-risk chain resulting in cascading impacts of different degrees, the user can analyse scenarios of varying severity by choosing a specific earthquake from the database. We applied a similar approach** *for a multi-risk story on volcanic activities with compound hazards, i.e. ashfall and lahars, with damage on buildings and impact on the power network* **in a case study** *for the volcano Cotopaxi in Ecuador* **(see Sect. 4, vii). Here, too, users were able to analyse a number of scenarios of varying severities based on different VEI values** *(cf. Gómez Zapata et al., 2021a)."*

Line 703: *"vii. Transferability and scalability: The approach was presented for an earthquake-tsunami multi-risk story* **for the Lima Metropolitan area in Peru.** *Regarding the transferability to another region,* *we report that we* **have applied** *the approach* **in two further** *case studies. In the coastal area of Greater Valparaíso, Chile (cf. Gómez Zapata et al., 2021d; Gómez Zapata et al., 2022a),* **the multi-risk story was like Lima Metropolitan area focusing on the earthquake and tsunami cascade affecting housing and the critical infrastructure power grid. In Ecuador** *the approach has also been* **adapted** *for compound hazards (two hazard events happening in parallel)* **analysing the impacts of ash fall and lahar around the volcano Cotopaxi in Ecuador** *(cf. Gómez Zapata et al., 2021a). Similar to the Peru study case, a feedback process with four iterations involving comparable stakeholder groups was implemented. The use of similar questionnaires in the three country studies helped to compare results from the feedback. The results showed that comments on the main features of the tool were consistent across the study cases. Of course, there were specific points to consider in the individual countries, for example, regarding the colours used to display the results or the damage classes, as participating countries use different damage categories and colour codes. To ensure the transferability and scalability of our approach, the tool was designed from the outset to be adaptable to all types of complex multi-risk stories (see point iii on 'Complexity') at different scales and to accommodate national or local preferences regarding damage categories or colour codes."*

7. It is surprising that the risk metrics output from the tool are not designed in collaboration with stakeholders, given their importance to the decision-making processes that the tool strives to facilitate. Furthermore, the metrics used are rather narrow in scope; repair costs may not fairly capture the effects of hazards on lowincome populations, for instance (e.g., see https://www.nature.com/articles/s41893-020-0508-7). I think the narrow range of metrics used and the lack of stakeholder involvement in their design might be considered a notable limitation of the

framework and an aspect that should be improved in future work. I therefore believe it warrants some discussion in the conclusions.

We have updated Section 4 "Discussion and Conclusions", not only on the discussion regarding the used risk metrics, but also other topics. We covered your concerns regarding the used risk metrics in the discussion point iii on "*Complexity*":

Line 656: „*iii. Complexity: Multi-risk situations can become very complex. Obviously, models and scenarios are always incomplete as they* only can approximate the complexity of *real situations (see for example the risk framework introduced by Taubenböck et al. (2008) with the manifold and still incomplete indicators for operationalization). The analytical process of the interactions of elements in scenarios is furthermore confined to selected processes. For demonstration purposes, we limited ourselves to the physical elements of vulnerability (buildings, critical infrastructure). Table 3 lists the numerous and partly high-resolution input data for the relatively simple earthquake-tsunami story. This is already a minimal data set to model and approximate the situation realistically with considerable uncertainties. More high-resolution data sets can improve the modelling and reduce the uncertainties. An important factor in the evaluation of data inputs is certainly the available IT resources for processing and modelling. Economic, environmental, political, social and societal aspects of vulnerability were left out, which, however, is not implying any judgement on their relevance for assessing and understanding vulnerability. This, of course,* resulted in a considerable limited representation of what would actually happen in a real disaster situation. *This limitation was openly addressed and made transparent in the feedback process. Despite this limitation, the stakeholders still rated the potential of the tool as high, considering the results of the physical vulnerability assessment and it has* stimulated them to develop new strategies for capacity building and resilience measures. *Ultimately, the tool is designed in such a way that interested parties can integrate social factors of vulnerability at any time during adaptation and further development of the multi-risk story. To allow for this, we made the framework and its source code publicly available.*"

Some minor technical comments:

1. Line 30: I think the phrase "tool through an iterative participative approach" was more a more appropriate description than the term "method"

Thank you for the suggestion. We updated the abstracted and corrected the sentence accordingly.

2. Line 405: There are two sentences here that both state the demonstrator light version provides three different modes. Also, there is a typo in the statement: "comparison of two different scenarios within one multi-risk story two.." (same typo exists in line 529)

We apologize the duplication and typos. We have checked and corrected the text accordingly.

3. Line 568: please specify what you mean by "users"

We changed the sentence as follows:

Line 544: *"During the joint discussion* with the various stakeholders involved, a compromise had to be found between the requirements of *practical DRM and planning processes on the one hand, and the technical possibilities of modelling certain processes* on the other.*"*

4. Line 628: "totally likelihood" does not make sense .. rephrase using something along the lines of "proportion who are totally likely to use the tool"

Line 606: *"Although there was a slight decrease in the proportion who are totally likely to use the tool (year 1: 18%, year 3: 14%), we believe it is fair to say that the overall percentage has increased, as 39% were very likely to use the tool in year 3, while this answer was not given at all in year 1 (Fig. 10d)."*

5. Line 656: this sentence does not read correctly. I think you should rephrase the start of it along the lines of: "In this paper, we presented one (of many possible) approach(es) to multi-risk analysis that can make a practical contribution to…"

Thank you – we have amended the sentence accordingly.

6. Figure 8: "right ride" is a typo in the caption

Thanks for spotting this – we have corrected the typo.

**Anonymous Referee #2, RC2: 'Comment on nhess-2023-142'**

Thank you very much to the authors for their detailed responses to my comments (Anonymous Referee 2) and those of Anonymous Referee 1.

Your replies were clear and you let us know what you changed as a result in the manuscript. Many replies though were slightly confusing as you used future tense ('we will do….' 'we will add…') rather than telling us what you in fact did. I think this was the result of taking the reply in the response to our comments online, rather than the current stage of submitting the revised manuscript. However, the intent was mostly clear (although at times confusing, and I had to keep going back to the track change manuscript).

Yes, after getting your reviewer comments, we answered them with future tense, as we only updated the manuscript after the editor allowed us to re-submit a revised version of the manuscript. We are sorry that this brought some confusion.

In terms of replies, I think that you have made definite improvements, but at times these were superficial changes of a few words here or there, other times more substantive (e.g., for formatting, or adding in paragraphs here or there); more could have been done to substantively change the results rather than a bit of wordsmithing. As a result, I recommend slightly more than 'minor' changes to bring this to an appropriate level for publication. Ultimately, the manuscript will be a good addition to the literature, but taking time now to make the manuscript a bit more robust and in-depth, will make it that much more useful to the community.

I glanced over the Referee 1 comments and replies and will let them reply to those, although did feel, that at times you were sidestepping some of the substance of what was being asked. You answered many of my concerns, but some (will be mentioned below) could have been more substantive changes made.

Thank you for taking again for the time to provide this comprehensive review. We tried to further improve the paper based on your comments, as well as the comments of reviewer #1.

Abstract. I repeat what I said before. Although you have changed the manuscript you did not understand what I said. "This is a bit high level and more a motivation rather than an actual (with metrics such as 'how many' and 'of what') summary of the paper. I suggest you rethink a bit the abstract and consider more how it is a summary of the paper."

An example of including summary numbers to better help the reader might be the following (I'm not saying to use it, but it is an example after running your paper through an AI and asking for a summary):

"In our increasingly interconnected society, urbanization and the vulnerability to natural hazards, including climate change, have created complex risk scenarios. This paper introduces a modular approach for multi-risk assessment aimed at enhancing the capabilities of disaster risk managers, urban planners, and critical infrastructure operators. We developed a simulation and visualization method for various scenarios based on a decentralized system architecture using distributed web services, accessible via a user-friendly interface. Our approach is demonstrated through a case study of earthquakes and tsunamis in the Lima Metropolitan area, a megacity exposed to cascading natural hazards. The development involved a structured feedback process with ## participants over ## years, including ## [? disaster risk managers] and [## ? urban planners], who evaluated the tool's potential as a complementary analysis and visualization aid. Users reported high satisfaction, particularly appreciating the ability to simulate and compare different scenarios. The demonstrator's practical relevance and user-oriented design suggest it as a promising method for improving the understanding and preparedness for complex multi-risk situations."

Thank you very much for giving us such a detailed suggestion to improve the abstract. We considered your comments and updated the abstract as follows:

Line 18: *"Abstract. We live in a rapidly changing and globalized society. The increasing interdependence and interconnection of our economic, social and technical systems, growing urbanization and increasing vulnerability to natural hazards (including climate change) are leading to ever more complex risk situations. This paper presents a modular approach for user-centred multi-risk assessment aimed to support disaster risk managers, urban planners or critical infrastructure operators. Based on the latest scientific and technical capabilities, we developed a method that enables the simulation and visualization of a range of scenarios with different intensities. It is based on a modular and decentralized system architecture using distributed web services that are published online, accessible via a user-friendly interface. The approach is demonstrated using the example of earthquakes and tsunamis for the Lima Metropolitan area (Peru), a megacity exposed to various cascading natural hazards. The development involved a wider group of Peruvian stakeholders from research and practice in a structured, iterative and participative feedback process over a period of 2.5-years to capture the needs and requirements from the user perspective. Results from the feedback process, including 94 responses to 5 questionnaires, confirmed the high potential of the demonstrator as a complementary analysis and exploration tool. Together with the visualisation of cascading processes, the ability to simulate and compare scenarios of varying severity was considered relevant and useful for improving understanding and preparedness for complex multi-risk situations in the practical application, especially at the local level.*

Table 1. This is an excellent addition, but (a) please now discuss it in the text (even one sentence) such as "In Table 1 are shown four global strategies from YYYY to YYYY. Within these strategies, we see overall that…."

We are pleased to read that the additional table based on your original suggestion is well received. We have described the table in the text as follows:

Line 118: *"Table 1 shows four selected global strategies ranging from the 2030 Agenda for Sustainable Development (UNISDR, 2015) to the New Urban Agenda (United Nations, 2017). As part of these strategies, we see the need for a better understanding of disaster risk and the need to consider the requirements of different categories of users."*

Vulnerability. Thank you for defining this as physical and systemic. Please add a reference(s) for where you first introduce these definitions at the end of Section 1.

Thank you for this this advice. We have included a reference at the end of Section 1, see:

Line 133: *"Following this introduction, Sect. 2 presents the conceptual approach to developing a scenario-based multi-risk assessment tool. With the aim of developing a demonstrator (and not a fully operational system), we focused on analysing the physical vulnerability (e.g., Fuchs et al., 2018) of buildings (i.e., the likelihood that assets will be damaged or destroyed when exposed to a hazard event), and the systemic vulnerability (e.g., Pascale et al., 2010; Hernandez-Fajardo and Dueñas-Osorio, 2013) of electrical power networks (i.e., probability of failure of interconnected systems given hazard intensities). Sect. 3 describes the results and steps taken, including findings from the user perspective. The discussions and conclusions are outlined in Sect. 4."*

Fuchs S, Frazier T, Siebeneck L. Physical Vulnerability. In: Fuchs S, Thaler T, eds. Vulnerability and Resilience to Natural Hazards. Cambridge University Press; 2018:32-52.

Pascale, S., Sdao, F., and Sole, A.: A model for assessing the systemic vulnerability in landslide prone areas, Nat. Hazards Earth Syst. Sci., 10, 1575–1590, https://doi.org/10.5194/nhess-10-1575-2010, 2010.

Hernandez-Fajardo, I. and Dueñas-Osorio, L.: Probabilistic study of cascading failures in complex interdependent lifeline systems, Reliability Engineering & System Safety, 111, 260–272, https://doi.org/10.1016/j.ress.2012.10.012, 2013.

Social vulnerability. In section 4, under complexity, you state that you do not introduce social vulnerability because of data restrictions and data protection issues. Yes, I agree that doing a full analysis of social vulnerability might be difficult, but one can even do partial analyses based on (in this case) census data, or data that IS publically available or from other papers. Stating you could not do it is a bit of a cop-out. Just say you did not do it, and don't make justifications. As an example, when I go to "Changes in Spatial Inequality and Residential Segregation in Metropolitan Lima" there are nice maps of socio-economic and other variables by district level. I am familiar with Lima, and there are multiple (at least district-level) maps of various characteristics from the different censuses or papers. I get frustrated when people discuss 'risk' but then leave out social vulnerability—in this case, you did so, but it was a limitation and could have been added if you had wanted to (even at a course level). Acknowledge it was not part of your design, that there are metrics available to the public that could have been used (roughly) but were not, and that more detailed analysis might have data protection issues.

Thank you for bringing this to our attention. Based on your suggestions we have updated the paragraph on "iii. Complexity" in Section 4 accordingly.

Line 656: „iii. Complexity: Multi-risk situations can become very complex. Obviously, models and scenarios are always incomplete as they only can approximate the complexity of real situations (see for example the risk framework introduced by Taubenböck et al. (2008) with the manifold and still incomplete indicators for operationalization). The analytical process of the interactions of elements in scenarios is furthermore confined to selected processes. For demonstration purposes, we limited ourselves to the physical elements of vulnerability (buildings, critical infrastructure). Table 3 lists the numerous and partly high-resolution input data for the relatively simple earthquake-tsunami story. This is already a minimal data set to model and approximate the situation realistically with considerable uncertainties. More high-resolution data sets can improve the modelling and reduce the uncertainties. An important factor in the evaluation of data inputs is certainly the available IT resources for processing and modelling. Economic, environmental, political, social and societal aspects of vulnerability were left out, which, however, is not implying any judgement on their relevance for assessing and understanding vulnerability. This, of course, resulted in a considerable limited representation of what would actually happen in a real disaster situation. This limitation was openly addressed and made transparent in the feedback process. Despite this limitation, the stakeholders still rated the potential of the tool as high, considering the results of the physical vulnerability assessment and it has stimulated them to develop new strategies for capacity building and resilience measures. Ultimately, the tool is designed in such a way that interested parties can integrate social factors of vulnerability at any time during adaptation and further development of the multi-risk story. To allow for this, we made the framework and its source code publicly available."

Conceptual Approach. Previously I suggested you look again at the conceptual approach, to make it a bit more user-friendly and less dense, by breaking out some of the large amounts of text into bullet points or numbers or making it easier to read. You said that "you are aware that there is a lot of information provide. Our target group is the scientific community and not practitioners." Having a lot of information is fine, and lots of details, but again I ask if you might be able to make it easier to read. I'm part of the scientific community this is aimed at, and had to read the section a couple of times to grasp what was being proposed.

To improve readability, we have further structured the text into bullet points. Accordingly, we have changed it as follows:

Line 145: *"Considering the above-mentioned guidelines and strategies in the context of disaster risk reduction (DRR) and disaster risk management (DRM), as well as the outlined research needs, we present a conceptual approach developed within the research projects RIESGOS and its successor RIESGOS 2.0 (Schoepfer et al., 2018; Schoepfer et al., 2024). The projects focused on the development of innovative scientific methods for the assessment of multi-risk situations with the aim of designing an approach that meets the needs of users at the local level. In addition to the German team coming from various disciplines, the project collaborated with a variety of research institutions and public authorities in Chile, Peru and Ecuador. This collaboration, both with potential users and stakeholders across different levels, frames the novelty of the approach towards its practical applicability.*

*The starting point of our conceptual approach is the finding that local risk situations and the challenges for decision-makers to pursue global risk reduction goals in practice can vary across the globe. Thus, there is a gap between scientific and technical possibilities (i.e., the knowledge created by them and concrete fact-based decisions in the planning or political field). The conceptualization of this overall approach is visualized in Fig. 1.*

- *First, we conducted a context and stakeholder analysis to understand the organizational environment and underlying structures of the disaster risk governance and to identify stakeholders to engage (Sect. 2.1).*
- *A concept for a scenario-based multi-risk information system was developed (Sect. 2.2). We selected a story-based scenario concept that allows the description of a specific multi-risk situation and its representation through multiple scenarios (Sect. 2.2.1).*
- *As input for the demonstrator tool, the elements of risk (hazard, exposure, and vulnerability) and their impacts on critical infrastructure were considered in terms of their potential implementation (Sect. 2.2.2). In the process, we devoted efforts to the study of interactions at the physical and systemic vulnerability levels of cascading hazards, addressing cumulative damage and loss.*
- *During the development of the demonstrator for a multi-risk information system, we involved potential users from the beginning to ensure that the designed tool meets their requirements and needs (Sect. 2.2.3). For the demonstrator we chose a decentralized system architecture approach built on distributed web services, with a graphical user interface as the frontend (Sect. 2.2.4).*

*During the project individual results have already been published and are cited accordingly. In this paper, we aim to present the overall approach, with focus on the feedback process from the user perspective showing the practical relevance of the designed tool. We are convinced that such a user-oriented approach for exploring, describing and quantifying different What-if scenarios can constitute a valuable tool for understanding complex multi-risk situations and to prepare for such situations."*

We have also introduced some paragraphs to make the structure of the text and the line of argumentation clearer. We have done this systematically throughout section 2.

Users. Adding Table 2 was helpful. I'm still confused though how many came from which group and how many participants there were for each year. So in one place you you state 46 participants in year 3 and in another place there were 37 participants in Year 3. I can't find how many there were in Year 2 or 1, or their realistic makeup, and only twice do you mention for Year 3, and the numbers are different. I'm confused. And, this was a point I brought up before. You need to be much clearer throughout on the number of participants in every workshop, both putting this into the text and into figure captions where 'percentages' are mentioned. This is a 'red flag' for me, when an author is vague.

Thank you for noticing this error in participant counts. We have corrected the paragraph and added more information about the numbers of participants in Section 3.4, both in the text and figure captions, such as:

Line 551: *"For the case study in Peru, the first two workshops took place in Lima (mock-up V0.1: 4 December 2018, with 11 participants; version V1.0: 19 November 2019, with 46 participants) while the third workshop was held online due to the global pandemic travel restrictions (version V2.0: 9 February 2021, with 37 participants). Next to open feedback rounds, feedback was additionally collected via questionnaires. During this process, we experienced that complementary practical hands-on session (V1.0: 19 November 2019, with 16 participants; V2.0: 10 February 2021, with 12 participants) with the tool increased significantly the quality of feedback as one can document the user experience in action."*

Line 612: *"Figure 10. Feedback from user perspective obtained during the three development stages (mock-up V0.1, versions V1.0 und V2.0) of the demonstrator for a multi-risk information system for years 1 (V0.1), 2 (V1.0) and 3 (V2.0). The diagrams represent four selected questions (out of a total of 45) on the information content (Fig. 10a-c) and applicability of the tool (Fig. 10d) asked to stakeholders in Lima (V0.1 was evaluated by 11 participants; V1.0 by 46 participants; V2.0 by 37 participants)."*

Line 620: *"Figure 11. Feedback from user perspective on the potential of practical applicability obtained in a hands-on workshop with stakeholders and potential users (12 participants) in Lima during the development process of the demonstrator for a multi-risk information system in year 3 (V2.0)."*

Table 3. This is an excellent addition, but please discuss it in a few sentences (or paragraph). What should the reader be taking away from it?

Thank you for this positive feedback on Table 3. More information on the web services is described in section 3.3.1 above Table 3. We have now added as you suggested two more sentences as follows:

Line 476: *"Table 3 provides detailed information on the system components (web services) with input and output information including corresponding references. This set of web services documents the multi-risk sequence as visualised in Fig. 5. Interaction with the web services is achieved using the Web Processing Service (WPS) interface standard guidelines published by the Open Geospatial Consortium (OGC; WPS, 2018)."*

Additional Figures. Thank you for adding these additional figures on the GUI, as they helped to better understand what was being done. Double check that these will be legible though (font size) for the publication.

Thank you for pointing this out. We have improved the font size in the figures of the GUI. We have tried to find a good compromise between font size and displayed information layers as the tool is primarily optimized for online usage and not for screen prints (please see track change version).

Discussion. This has been expanded upon (thank you, although could use a bit more depth and insight from lessons learned, particularly around the practical elements of applying this in another location, data, and scalability), but you still do not relate a lot of your ideas back to the existing literature, as I suggested in my original review. In your revision, you added two in-text citations to the one already there. I realise it is not a numbers game, but it highly unusual to have a substantive discussion with so few revisions, particularly given the amount of work you have done (and as you say, this is for the scientific community). Examples of places where you might expand your discussion could be (these are examples):

• Expand on the user feedback, challenges they found, you summarised, relating it back to the wider literature, of what others have found. Are you finding similar results?

We have extended the discussion point „x. Co-creation with users" in Section 4 as follows:

Line 742: *"x. Co-creation with users: Our experience of collaboration between researchers, software developers and different potential users confirms that users' satisfaction with their involvement and*

*the resulting system are interdependent, with the degree of user satisfaction evolving at different stages of the development process, as postulated by Bano (2017). It also confirms that involving users as a primary source of information is an effective means of capturing system requirements (Kujala, 2003). However, such collaboration requires a strong engagement from all sides. We agree that the role of users in such a process must be carefully considered (Kujala, 2003) and therefore applied a moderated process, which allows that user demands can be communicated to the researchers and developers without outweighing the scientific relevance. At the same time, the involved user must be aware and able to cope with trade-offs and compromises, as not all requirements may be addressed or they might not be able to benefit directly from the tool while it is still under development or in a demonstrator stage. To avoid false expectations and misunderstandings, we emphasize that transparency and clear statements are most crucial throughout the user involvement process. Additionally, users (often) do not have the scientific expertise to adequately describe the individual processes in a multi-risk chain. Since the approach is based on the description of a multi-risk story, this story must always be defined in a joint dialog between users, researchers and software developers. In our experience, much of the mutual learning took place during face-to-face interaction rather than digitally. With this in mind, the design of such collaboration must be critically balanced against the quite justified demand for more cost-efficient methods of capturing implicit user needs and requirements in real product development contexts (Kujala 2003)."*

• Expand on what would be the minimum amount/types of data to apply this study elsewhere, commenting on limitations, and bringing this back to the wider literature.

We have expanded the discussion point "iii. Complexity" in Section 4 accordingly:

Line 656: *"iii. Complexity: Multi-risk situations can become very complex. Obviously, models and scenarios are always incomplete as they only can approximate the complexity of real situations (see for example the risk framework introduced by Taubenböck et al. (2008) with the manifold and still incomplete indicators for operationalization). The analytical process of the interactions of elements in scenarios is furthermore confined to selected processes. For demonstration purposes, we limited ourselves to the physical elements of vulnerability (buildings, critical infrastructure). Table 3 lists the numerous and partly high-resolution input data for the relatively simple earthquake-tsunami story. This is already a minimal data set to model and approximate the situation realistically with considerable uncertainties. More high-resolution data sets can improve the modelling and reduce the uncertainties. An important factor in the evaluation of data inputs is certainly the available IT resources for processing and modelling. Economic, environmental, political, social and societal aspects of vulnerability were left out, which, however, is not implying any judgement on their relevance for assessing and understanding vulnerability. This, of course, resulted in a considerable limited representation of what would actually happen in a real disaster situation. This limitation was openly addressed and made transparent in the feedback process. Despite this limitation, the stakeholders still rated the potential of the tool as high, considering the results of the physical vulnerability assessment and it has stimulated them to develop new strategies for capacity building and resilience measures. Ultimately, the tool is designed in such a way that interested parties can integrate social factors of vulnerability at any time during adaptation and further development of the multi-risk story. To allow for this, we made the framework and its source code publicly available."*

• How might your results be integrated into work being done by others. You mention it briefly in the 'operational system' but can you go further?

We further discussed point "viii. Operational system" in Section 4:

Line 721: *"viii. Operational system: Users showed strong interest in the presented tool. However, the transfer from a demonstrator system to an operational service requires further efforts along with a*

*clear commitment and solid institutional embedding. As we have chosen a decentralized service-oriented architecture (SOA) with distributed web services for the demonstrator also the individual web services (see Tab. 3) can be integrated in already existing information systems. The interaction with the web services is achieved using Web Processing Service (WPS) interface standard guidelines published by the Open Geospatial Consortium (OGC; WPS, 2018) and are openly documented. Interoperability is achieved by a thorough harmonization of input and output formats and the use of on-the-fly converters. Dedicated WPS create simulations of intensity maps for specific hazards on the fly (e.g. for earthquake ground motion simulation) or by querying a list of pre-simulated events (e.g. for tsunami inundation maps). We recommend a partnership between research institutions, public authorities and service providers whereas one key authority should act as the hosting institution to integrate the tool or individual web services. The integration process itself requires profound knowledge both, in the models and IT programming (both backend services and frontend development), which needs the interaction of different specialized institutions and professional support from IT experts."*

• There are a number of decision support tools out there now, how does this compare with them? This is what you have designed, with two references that include 'decision support tools'. Can you expand on these as decision-support tools?

The demonstrator tool developed in this project does not claim to be a decision support system. The aim was to develop a multi-risk assessment approach to help potential users such as disaster risk managers, urban planners or critical infrastructure operators to improve their ability to cope with the increasing complexity of risks. Based on the latest scientific and technical capabilities, we have developed a method that enables the simulation and visualization of a range of scenarios with different intensities. By using web services, the tool can be adapted very quickly and flexibly to changing input data (e.g. exposure data). The references to Komendantova (2014) and van Westen (2014b) listed in Section 1.2 were made in relation to user involvement and to the overall topic of multi-hazard frameworks, and not regarding decision support tools.

Minor:

• "It is worth noting" "It has to be noted" "We note" "It should be noted" "It is important to note" are used on over a dozen occasions. I generally avoid these in my own writing, and recommend you look at and remove some of these from the text (if not all of them). As a minimum, remove the 'it is important' or 'It should be', as these convey an urgency that is not needed.

Thank you for your recommendation. We have revised the manuscript accordingly and rephrased sentences where necessary.

• I suggest reconsidering the word 'tested' and definitely 'successful tested' as it often conveys a sense of 'truth' to practitioners, for whom this is intended. See "Verification, validation, and confirmation of numerical models in the earth sciences" by Naomi Oreskes et al. (1994). Perhaps instead consider replacing it with the words 'evaluated' or we 'used the approach'? For successful testing, this is difficult to ascertain how successful you really were, so I suggest other words such as 'applied'.

Thank you, we have corrected the wording in line with your advice.

• Paragraphs. You still have some very long paragraphs.

We have made another revision of the current version.

• First line, Section 4, should 'novel' be removed?

We have deleted the word 'novel'.

---

## Author Response (AR3)

**RC3: AUTHORS' RESPONSE TO THE REVIEWERS COMMENTS**

**nhess-2023-142**: "Between global risk reduction goals, scientific-technical capabilities and local realities: a novel modular approach for multi-risk assessment" by Schoepfer et al.

- Authors' response to the Reviewer#1 comments

Line numbers refer to the track change version (nhess-2023-142-manuscript-version4_ATC.pdf).

**Anonymous Referee #1, RC3: 'Comment on nhess-2023-142'**

I thank the authors for their further efforts to respond to my comments, and I believe they have largely addressed my remaining concerns. I have just two outstanding minor suggestions:

1. In the caption of Table 2, you should make it clear that the stakeholders shown in the table for each group are representative of the Peruvian context, which is the study area to be discussed later in the manuscript. (Otherwise, I think the reader might be confused as to why there is a sudden focus on Peru).

Thank you for your feedback and suggestion. In order to be more precise, we have modified the caption and have now integrated "Peruvian Stakeholders" in the second column in addition.

2. As I mentioned in the previous review, the conditional probabilities of hazards are somehow being accounted for given that the tsunami and the earthquake are not simulated independently (i.e., the same fault parameters are used for both, and the size of the tsunami is related to the magnitude of the earthquake). So, I suggest removing the sentence about conditional probabilities not being accounted for (line 240 of the current manuscript).

We have removed the sentence as suggested (line 225).

Please note that I also noticed the Jimenez et al. (2013) paper is missing from the reference section.

Thank you for pointing this out, we have now added Jimenez et al. (2013) in the reference section.

---

## Author Response (AR4)

**RC4: AUTHORS' RESPONSE TO THE EDITORIAL SUPPORT TEAM RECOMMENDATIONS**

**nhess-2023-142**: "Between global risk reduction goals, scientific-technical capabilities and local realities: a novel modular approach for multi-risk assessment" by Schoepfer et al.

- Authors' response to Editorial Support Team Recommendations in blue.

In your manuscript, please use full first names for all authors.

The Name of one co-author (Christian D. León) was corrected to "Christian León".

Please ensure that the reproduction rights for all figures have already been secured and that maps and aerials include the required copyright statements or credits as requested by the providers.

This was checked again for all figures.

**RC4: AUTHORS' RESPONSE TO THE EDITORIAL SUPPORT TEAM RECOMMENDATIONS**